

# An Overview of the Vertical Structure of the Atmospheric Boundary Layer in the Central Arctic during MOSAiC

Gina C. Jozef[1,2,3], John J. Cassano[1,2,3], Sandro Dahlke[4], Mckenzie Dice[1,2,3], Christopher J. Cox[5], Gijs de Boer[2,5,6]

5   [1]Dept. of Atmospheric and Oceanic Sciences, University of Colorado Boulder, Boulder, CO, USA
[2]Cooperative Institute for Research in Environmental Sciences, University of Colorado Boulder, Boulder, CO, USA
[3]National Snow and Ice Data Center, University of Colorado Boulder, Boulder, CO, USA
[4]Alfred Wegener Institute Helmholtz Centre for Polar and Marine Research, Potsdam, Germany
[5]NOAA Physical Sciences Laboratory, Boulder, CO, USA
10   [6]Integrated Remote and In Situ Sensing, University of Colorado Boulder, Boulder, CO, USA

*Correspondence to:* Gina Jozef (gina.jozef@colorado.edu)

**Abstract.** Observations collected during the Multidisciplinary drifting Observatory for the Study of Arctic Climate (MOSAiC) provide an annual cycle of the vertical thermodynamic and kinematic structure of the atmospheric boundary layer (ABL) in the central Arctic. A self-organizing map (SOM) analysis conducted using radiosonde observations shows a range in the Arctic ABL vertical structure from very shallow and stable, with a strong surface-based virtual potential temperature ($\theta_v$) inversion, to deep and near-neutral, with a weak elevated $\theta_v$ inversion. Profile observations from the DataHawk2 uncrewed aircraft system between 23 March and 26 July 2020 largely sampled the same profile structures, which can be further analyzed to provide unique insight into the turbulent characteristics of the ABL. The patterns identified by the SOM allowed for the derivation of criteria to categorize stability within and just above the ABL, which reveals that the Arctic ABL is stable and near-neutral with similar frequencies. In conjunction with observations from additional measurement platforms, including a 10 m meteorological tower, ceilometer, and microwave radiometer, the radiosonde observations provide insight into the relationships between atmospheric stability and a variety of atmospheric thermodynamic and kinematic features. The average ABL height was found to be 150 m, and ABL height increases with decreasing stability. A low-level jet was observed in 76% of the radiosondes, with an average height of 401 m and an average speed of 11.5 m s$^{-1}$. At least one temperature inversion below 5 km was observed in 99.7% of the radiosondes, with an average base height of 260 m and an average intensity of 4.8 °C. The only cases without a temperature inversion were those with weak stability aloft. Clouds were observed within the 30 minutes preceding radiosonde launch 64% of the time. These were typically low clouds, and high clouds largely coincide with a stable ABL. The amount of atmospheric moisture present increases with decreasing stability.

## 1 Introduction

The atmospheric boundary layer (ABL) is the turbulent lowest part of the atmosphere that is directly influenced by the earth's surface (Stull, 1988; Marsik et al., 1995). Its structure dictates the transfer of energy, moisture, and momentum between the Earth's surface and the overlying atmosphere (Brooks et al., 2017). A lack of detailed understanding of ABL structure over Arctic sea ice results from a historical shortage of the necessary in situ 
measurements. This study utilizes newly available high temporal and vertical spatial resolution atmospheric



observations from the Multidisciplinary drifting Observatory for the Study of Arctic Climate (MOSAiC; Shupe et al. 2020) to analyze the lower atmospheric structure over the full annual cycle in the central Arctic, focusing on statistical quantities relating to the ABL, low-level jets, temperature inversions, and atmospheric moisture.

Previous studies have concluded that the Arctic ABL is typically stably stratified, having minimal turbulence, with height less than 200 m, and often less than 50 m (Esau and Sorokina, 2010). However, the ABL over the Arctic Ocean can also be well-mixed (Esau and Sorokina, 2010; Persson et al., 2002). In summer, surface turbulent heat fluxes are generally quite small (Brooks et al., 2017) because the surface temperature is locked to the melting point of ice, so turbulent fluxes do not respond directly to changes in surface radiative forcing (Brooks et al. 2017; Persson 2012). Instead, turbulent fluxes are largely controlled by horizontal advection, either as a response to clouds (Brooks et al.,
2017) or when air flows from over open water to over sea ice (Tjernström, 2005), which has been observed to produce a shallow stable ABL (Cheng-Ying et al., 2011).

One source of ABL turbulence is a low-level jet (LLJ), which is a local maximum in the wind speed profile below 1.5 km (Tuononen et al., 2015) that is at least 2 m s$^{-1}$ greater than wind speed minima above and below (Stull, 1988). LLJs are common in the central Arctic and produce strong wind shear in the layer below the jet core, causing turbulent
mixing in the ABL (Banta, 2008) and subsequently may lead to weakening of stability. LLJs have been shown to primarily form under conditions of baroclinicity (either near the ice edge (Brümmer & Thiemann, 2002) or due to the passing of a transient cyclone (Jakobson et al., 2013)), or inertial oscillations (which can be induced when stable stratification returns after the passing of a storm (Andreas et al., 2010a; Jakobson et al., 2013)). A previous study conducted on LLJs in the central Arctic between 25 April to 31 August of 2007 found a mean LLJ core speed of 7.1
m s$^{-1}$, LLJ core altitude typically between 100 and 500 m, and LLJ depth usually between 400 and 600 m (Jakobson et al., 2013). Jakobson et al. (2013) also found the faster LLJs to have the jet core located inside the ABL. A similar analysis of LLJs during MOSAiC found LLJs to be present more than 40% of the time, with typical height below 400 m and speed between 6 and 14 m s$^{-1}$ (Lopez-Garcia et al., 2022).

Surface-based and low-level temperature inversions (TIs) are also common in the central Arctic, and are correlated to
ABL stability. TIs have been shown to contribute to Arctic amplification (Serreze and Francis, 2006; Serreze and Barry, 2011; Bintanja et al., 2011; Lesins et al., 2012; Gilson et al., 2018; Previdi et al., 2021) by dynamically decoupling the surface from the free atmosphere, so that surface heat flux perturbations cannot easily spread through the troposphere, and warming is concentrated near the surface (Lesins et al., 2012). Near-surface TIs also impact Arctic aerosol characteristics including the destruction of boundary layer ozone at the onset of polar sunrise and the
transport of Arctic haze (Kahl, 1990), and contribute to the formation of fog during Arctic summer (Gilson et al., 2018). In Arctic winter, TIs form due to longwave cooling of the surface under predominantly clear-sky conditions while in Arctic summer, TIs typically originate from advection and subsidence processes, though ice and snow melt can also contribute to the formation of surface-based TIs (Gilson et al., 2018). Previous studies conducted over Greenland and along the Alaskan Arctic found there to be at least one TI 85-99.2% of the time, which had base height
up to about 300m, depth between 225 and 850 m, and intensity of 1.6 to 11 °C (Gilson et al., 2018; Kahl, 1990).



Arctic ABL structure is highly dependent on atmospheric moisture, particularly clouds containing liquid water, which have a warming influence on the surface most of the year when compared to clear-sky conditions (Brooks et al., 2017; Shupe and Intrieri, 2004). In the Arctic winter, a lack of open water evaporation leads to persistent periods of clear

skies or thin high clouds, though low-level clouds are possible during stormy conditions (Brooks et al. 2017; Persson et al., 2002). In the Arctic summer, the ABL is often capped with low level stratiform clouds (Tjernström et al., 2012), which form as a result of ample moisture available during the melt season. Observations of clouds over Ny-Ålesund, Svalbard reveal that about 40% of the lowest cloud base heights are located between 0.5 and 1 km AGL, with clouds observed 75% of the time (Asutosh et al., 2021), and cloud observations from six Arctic observatories found an annual

cloud occurrence of 58 – 83% (dominated by low clouds), with the highest and lowest cloud frequencies in fall and winter, respectively (Shupe et al., 2011a; Shupe et al., 2011b).

The atmospheric features discussed above all interact with the surface energy budget, which impacts sea ice thickness and extent. Thus, to properly represent the central Arctic in weather and climate models, the frequency and

characteristics of these features must be documented. While previous work does reveal some important information about the Arctic ABL features and processes, most in situ observations have either been brief, located near the coast, or have only included measurements of a subset of important atmospheric features. Particularly lacking have been observations of atmospheric properties during the winter, as few previous field campaigns have gathered wintertime Arctic observations (e.g., the Surface Heat Budget of the Arctic Ocean (SHEBA) project; Uttal et al., 2002). MOSAiC

obtained the necessary data from the central Arctic ice pack, between September 2019 and October 2020, to analyze atmospheric thermodynamic and kinematic features related to the ABL above the sea ice pack, from deep in the pack ice to near the marginal ice zone.

The questions guiding this study are as follows: what are the different ABL structures and stability regimes present in

the central Arctic, and how do these differ by season? Additionally, what are the relationships between ABL stability and other relevant atmospheric features, including LLJs, TIs and atmospheric moisture? We hypothesize that each of the previously stated atmospheric features are prevalent throughout an entire year in the central Arctic, and ABL stability directly relates to the characteristics of these features.

To determine ABL structure and identify important thermodynamic and kinematic features in the Arctic ABL, we

primarily use profile data from radiosondes launched at least four times per day throughout the MOSAiC year. A self-organizing map (SOM) analysis is conducted with the radiosonde profiles to objectively identify the different ABL structures that occur in the central Arctic, and their relative frequencies during the MOSAiC year. Additionally, profiles collected by the DataHawk2 (DH2) uncrewed aircraft system (UAS) from March – July of 2020 (de Boer et al., 2022) are mapped to the SOM to identify which ABL structures were sampled by the DH2, as the DH2

observations can be further evaluated for a higher resolution perspective of these structures. For each radiosonde and UAS profile, we determine ABL stability. From the radiosondes, we identify additional characteristics of the ABL, LLJs and TIs. This is supplemented with information about near-surface atmospheric state and atmospheric moisture



from additional measurement platforms discussed in Sect. 2.1. Then, we provide statistics on the various atmospheric characteristics as a function of the stability regime. The results provide an annual cycle of the central Arctic ABL

structure and relevant features.

## 2 Methods

### 2.1 Observational data from MOSAiC

Data used in this study were collected during MOSAiC, a year-long icebreaker-based expedition lasting from September 2019 through October 2020, in which the Research Vessel *Polarstern* (Alfred-Wegener-Institut Helmholtz-

Zentrum für Polar- und Meeresforschung, 2017) was frozen into the central Arctic Ocean sea ice pack, and was set to drift passively across the central Arctic for the entire year. However, between 17 May and 18 June, between 31 July and 21 August, and between 21 September and 1 October 2020, it was necessary for the *Polarstern* to travel under its own power. During the MOSAiC year, many measurements were taken to observe the atmosphere (Shupe et al. 2022), sea ice (Nicolaus et al. 2022), and ocean (Rabe et al. 2022), with the result being the most comprehensive observations

of the central Arctic climate system to date. These measurements span all seasons, as well as both far from and close to the sea ice edge, as the *Polarstern* essentially followed one ice floe for its annual life cycle (only relocating to a new ice floe for the final two months of the expedition).

For this study, we primarily use profile data from the balloon-borne Vaisala RS41 radiosondes, which were launched from the helicopter deck of the *Polarstern* (~12 m above sea level) at least four times per day (every 6 hours), typically

at 05:00, 11:00, 17:00, and 23:00 UTC (Maturilli et al., 2021). We use the level 2 radiosonde product (Maturilli et al., 2021) for this analysis, as the level 2 radiosonde data are found to be more reliable in the lower troposphere than the level 3 radiosonde data (Maturilli et al., 2022). Figure 1 shows the location of each radiosonde launch throughout the MOSAiC year. From the radiosondes, we utilize measurements of temperature, pressure, relative humidity, and wind speed and direction, as well as derived measurements of virtual potential temperature ($\theta_v$) and absolute humidity up

to 5 km. The radiosondes ascend at a rate of approximately 5 m s$^{-1}$, sampling with a frequency of 1 Hz, which results in measurements about every 5 m throughout the ascent. Additional profile data comes from the DH2 (Jozef et al., 2021), a battery powered fixed-wing UAS (1.1 m wingspan, 1.8 kg weight, 40 min endurance; Hamilton et al. (2022)), which was operated between 23 March and 26 July 2020 (de Boer et al., 2022). From the DH2, we utilize measurements of temperature, pressure, and relative humidity, as well as derived measurements of $\theta_v$ up to 1 km in

clear-sky conditions, or cloud base height if lower than 1 km. The locations of DH2 profiles are also marked in Fig. 1.





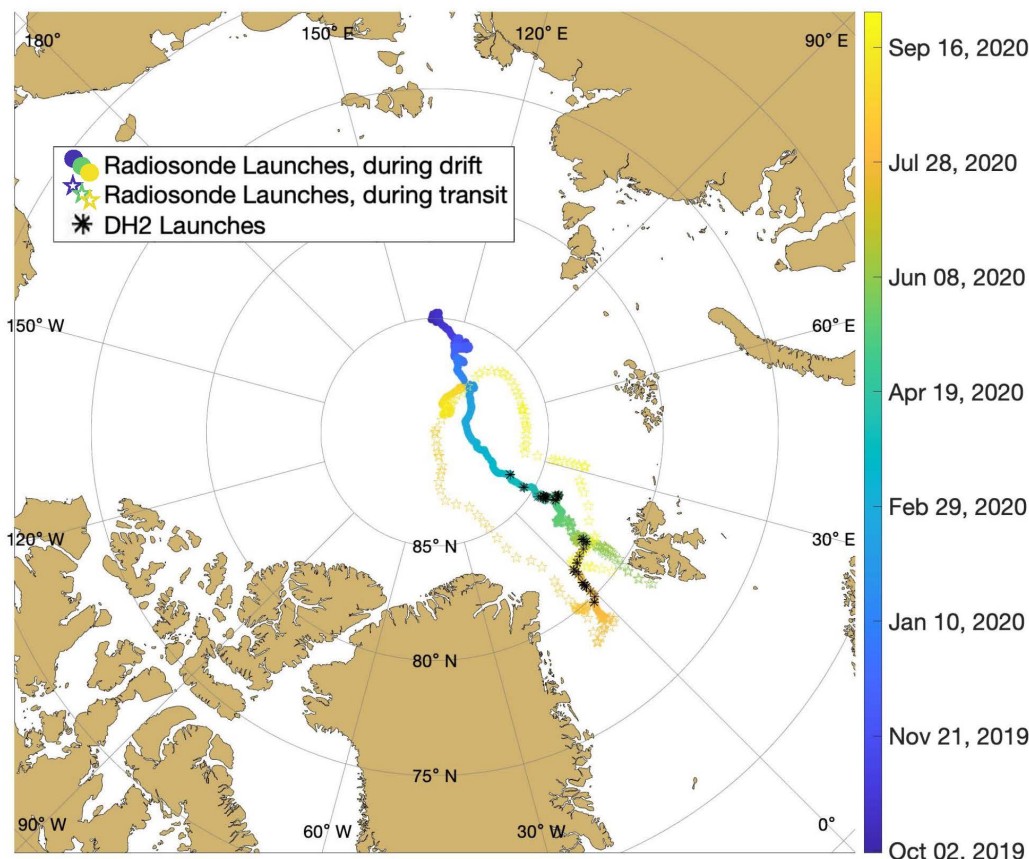

**Figure 1.** Map of the central Arctic showing the location of each radiosonde launch, color coded by date. Circular symbols indicate when the *Polarstern* was passively drifting, and star symbols indicate when the *Polarstern* was travelling under its own power. Black stars indicate the locations of DataHawk2 flights.

In addition to the profile data provided by the radiosondes and DH2, we utilize observations from a few other measurement platforms which add to the overall picture of the ABL at the time of each radiosonde launch. Atmospheric observations of wind speed at 2 m above the surface, as well as a derivation of bulk friction velocity ($u_*$), come from a 10 m meteorological tower (hereafter called the met tower; Cox et al., 2023) located on the sea ice near the *Polarstern* (Cox et al., submitted), and provide information about near-surface turbulence at the time of each radiosonde launch.

Information on cloud cover comes from a Vaisala Ceilometer CL31 (ARM user facility, 2019a), which measures atmospheric backscatter and cloud base height (CBH), and allows us to determine the altitude and frequency of clouds at and before radiosonde launch. Additionally, precipitable water vapor (PWV) and liquid water path (LWP) come from the MWRRET Value-added Product (ARM user facility, 2019b) which derives PWV and LWP from ARM 2-channel microwave radiometer measured brightness temperatures. PWV and LWP derivations and uncertainties are



discussed in Turner et al. (2007) and Cadeddu et al. (2103) respectively. Both the ceilometer and microwave radiometer were located on the P-deck of the *Polarstern* (depicted in Fig. 3 of Shupe et al. 2022), which is approximately 20 m above sea level, and could occasionally be above a layer of shallow fog. Table 1 lists the instrument name and uncertainty for each of the observational variables used in this study.


**Table 1:** Instrument name and uncertainty for each variable used in this study.

| Platform | Variable | Instrumentation | Uncertainty |
|---|---|---|---|
| **Radiosonde** | Pressure | Vaisala RS41-SGP | 1.0 hPa (> 100 hPa), 0.6 hPa (< 100 hPa) |
| | Temperature | | 0.3 °C (< 16 km) 0.4 °C (> 16 km) |
| | Relative Humidity | | 4 % |
| | Wind speed | | 0.15 m s$^{-1}$ |
| | Wind Direction | | 2 ° |
| **DataHawk2** | Pressure | Vaisala RSS421 | 0.4 hPa |
| | Temperature | | 0.1 °C |
| | Relative Humidity | | 2 % |
| **Met tower** | 2 m wind speed | Metek uSonic-Cage MP sonic anemometer | 0.3 m s$^{-1}$ |
| | Bulk friction velocity ($u_*$) | Derived from Vaisala HMT337/PTU307, Metek uSonic-3 Cage MP, and ARM's Eppley Precision Infrared Radiometer (following Andreas et al. (2010b)) | 4.4 % (estimated random error, 10 min) and 6% (bias) |
| **Ceilometer** | Cloud base height | Vaisala CL31 | 5 m |
| **Microwave radiometer** | Precipitable water vapor | ARM 2-channel microwave radiometer | 0.5 mm |
| | Liquid water path | | 30 g m$^{-2}$ |

### 2.2 Deriving quantities from observational data

Before the radiosonde profiles were analyzed, radiosonde measurements were corrected to account for the local "heat
island" resulting from the presence of the *Polarstern.* This local source of heat resulted in the frequent occurrence of elevated temperatures near the launch point, resulting in inconsistencies in the observed temperatures in the lowermost part of the atmosphere. This phenomenon can be recognized by an artificial temperature structure indicative of a convective layer in the lowest radiosonde measurements, which we know is unlikely (Tjernström et al., 2004; Brooks et al., 2017). Thus, if this "convective layer" was present, then the lowest radiosonde measurements were visually
compared to measurements from the met tower to identify when temperature values were anomalously warm. This was identifiable when the tower measurements interpolated upward, given their observed slope, did not match up with the lowest radiosonde measurement. The first credible value of the radiosonde measurements was found when the tower measurements interpolated upward would line up with the observed radiosonde measurement, or in the case of





a temperature offset between the tower and radiosonde, would have the same slope. All data at the altitudes below this first credible value were removed. This helps in also removing faulty wind measurements that occur as a result of flow distortion around the ship (Berry et al., 2001), and the radiosonde motion induced by the initial unraveling of the string that connects the radiosonde to the balloon.

An additional disruption of the radiosonde measurements sometimes occurred because of the passage of the balloon through the ship's exhaust plume. When it was unambiguous that the radiosonde passed through the ship's plume (evident by a sharp increase and subsequent decrease in temperature, typically by ~0.5-1°C over a vertical distance of ~10-30 m, identified visually), these values were replaced by values resulting from interpolation between the closest credible values above and below the anomalous measurements, which were identified as the last point just before the increase and the first point just after the decrease in temperature values, to acquire a continuous profile of reliable temperatures. Lastly, we determined that 92% of profiles have credible measurements as low as 35 m AGL. To allow for a consistent bottom height for our ABL analysis, we only consider profiles in which there is a good measurement at 35 m. This altitude is a compromise between removing too much low altitude data or removing too many radiosonde profiles from analysis. After removing all profiles in which there is not trustworthy data as low as 35 m, we retain 1377 MOSAiC radiosonde profiles for analysis.

ABL height from each radiosonde profile was determined using a bulk Richardson number ($Ri_b$) based approach in which the top of the ABL was identified as the first altitude in which $Ri_b$ exceeds a critical value of 0.5 and remains above the critical value for at least 20 consecutive meters (Jozef et al., 2022). The methodology for calculating the $Ri_b$ profile used to identify ABL height follows that described in Jozef et al. (2022) and Jozef et al. (submitted). Once ABL height was identified, we determined the vertical gradient in virtual potential temperature ($d\theta_v/dz$) and horizontal wind speed ($dV/dz$) between 35 m and the top of the ABL for each radiosonde.

LLJs were identified from each radiosonde, where there was a maximum in the wind speed that was at least 2 m s$^{-1}$ greater than the wind speed minima above and below (Stull, 1988). As described in Tuononen et al. (2015), only situations in which both the wind speed maximum (the LLJ core) and the minimum above the core were both below 1.5 km were identified as LLJs. Following methods described in Jozef et al. (submitted), if an LLJ was found, we identified the LLJ core altitude as the altitude of the maximum in the wind speed, and the LLJ speed as the wind speed at that altitude (Jakobson et al., 2013). Additionally, we identified the LLJ top as the altitude of the minimum in the wind speed profile above the LLJ. The altitude difference between the LLJ core and top is then the LLJ depth (Jakobson et al., 2013; Jozef et al., submitted). Our analysis differs from that by Lopez-Garcia et al. (2022) as they only considered LLJs in which the jet core speed was at least 25% faster than the wind speed minimum above the jet core, whereas we do not include this criterion, and thus our analysis also includes LLJs which occur in ubiquitously high wind speed environments.

TI layers were identified using a profile of temperature gradient ($dT/dz$) for each radiosonde case. Using methods described in Jozef et al. (submitted), the $dT/dz$ profile was calculated, and a TI layer was identified where $dT/dz$ exceeds a threshold of 0.65 °C (100 m)$^{-1}$ for at least 25 m. If $dT/dz$ goes below the threshold for less than 100 m



between two TI layers, then this is all considered the same TI layer (Kahl, 1990; Gilson et al., 2018). TI depth is the
vertical distance between the TI bottom and top, and the intensity of the TI is the difference between the temperatures
at the TI bottom and top (Gilson et al., 2018).

Moisture characteristics associated with each radiosonde are identified using measurements within the 30 minutes
preceding radiosonde launch. Thus, CBH, LWP, and PWV are taken as the average within that 30 minute interval.
We use this 30 minute interval, as this is a long enough time for the presence of the cloud and atmospheric moisture
to impact atmospheric stability and structure close to the surface. Mixing ratio at ABL height is derived from the
radiosonde profile.

Any other point measurements associated with each radiosonde (2 m wind speed and $u_*$) are calculated as the average
over a period of 5 minutes before to 5 minutes after radiosonde launch, as described in Jozef et al. (submitted). The
variables described in this section will hereafter collectively be called "composite variables."

**2.3 Self-organizing map analysis**

To determine the range of lower atmospheric structures which occur in the central Arctic, as observed during
MOSAiC, we applied a self-organizing map (SOM) analysis. The SOM analysis is an artificial neural network
approach that objectively identifies a user selected number of patterns in a training data set (Kohonen, 2001). The
result of this analysis is a two-dimensional array called a SOM, in which similar patterns are grouped so that the
squared difference between the training data and the patterns present in the SOM are minimized (Dice and Cassano,
2022). In the SOM, the patterns which are most similar to each other are located adjacently in the two-dimensional
array, and conversely the most different patterns are on opposite sides of the SOM (Dice and Cassano, 2022; Cassano
et al., 2016). In this study, the SOM was trained with the $d\theta_v/dz$ profile for each of the 1377 MOSAiC radiosonde
observations, using the SOM-PAK software (http://www.cis.hut.fi/research/som-research), the details of which are
described by Kohonen et al. (1996). While we also tried training the SOM with $\theta_v$ anomaly profiles, the SOM trained
with $d\theta_v/dz$ profiles best differentiated the various stability regimes present in the observations, by distinctly
highlighting stable versus well-mixed layers.

A SOM is an unsupervised neural network algorithm, but the user must specify the size of the resulting two-
dimensional array (the number and orientation of patterns; Cassano et al., 2015) before the training begins. For this
analysis, we evaluated SOMs with size and orientation of 5x4 (20 patterns) to 7x5 (35 patterns), and in the end found
that a 6x5 SOM (30 patterns) best displayed the range of $d\theta_v/dz$ structures present in the 1377 MOSAiC profiles. The
SOM algorithm applies an iterative process to identify the 30 patterns that span the full range of profile types in the
training data. Once these 30 patterns have been defined by the algorithm, each individual radiosonde profile is
"mapped" to the SOM (Dice and Cassano, 2022; Cassano et al., 2016). This is done by associating each individual
profile to a single SOM pattern by finding the SOM pattern that has the least squared difference when compared to
the radiosonde profile.



Before applying the SOM algorithm, the radiosonde profiles were interpolated to a consistent vertical grid of 5 m spacing between 35 m and 1000 m. The maximum altitude of 1000 m was chosen because it includes the full depth of the ABL in every case and also allows for diagnosing stability immediately above the ABL. A linear interpolation was used for temperature, relative humidity, and wind speed, and pressure was interpolated with the hypsometric equation. After interpolation, wind direction was calculated using the interpolated values of zonal and meridional wind speed. Additionally, $\theta_v$ every 5 m was calculated using the interpolated values of temperature, relative humidity, and pressure. Then, profiles of $d\theta_v/dz$ in K $(100\ m)^{-1}$ were calculated, resulting in values of $d\theta_v/dz$ at 37.5 m, 42.5 m, 47.5 m, and so on, with the last value being at 997.5 m. It is these profiles of $d\theta_v/dz$ which were used to train the SOM.

Figure 2 shows the 30 SOM patterns of $d\theta_v/dz$, as well as each individual radiosonde profile mapped to the SOM. The observations that map to a given SOM pattern are referred to as the best matching units (BMUs; Dice and Cassano, 2022) for that pattern. Using the list of BMUs, the frequency of occurrence of each pattern, annually or seasonally, can be determined to analyze the ABL structure at different times of year. Seasonal analysis in this paper is carried out by grouping observations during September, October, and November as fall; December, January, and February as winter; March, April, and May as spring; and June, July, and August as summer.

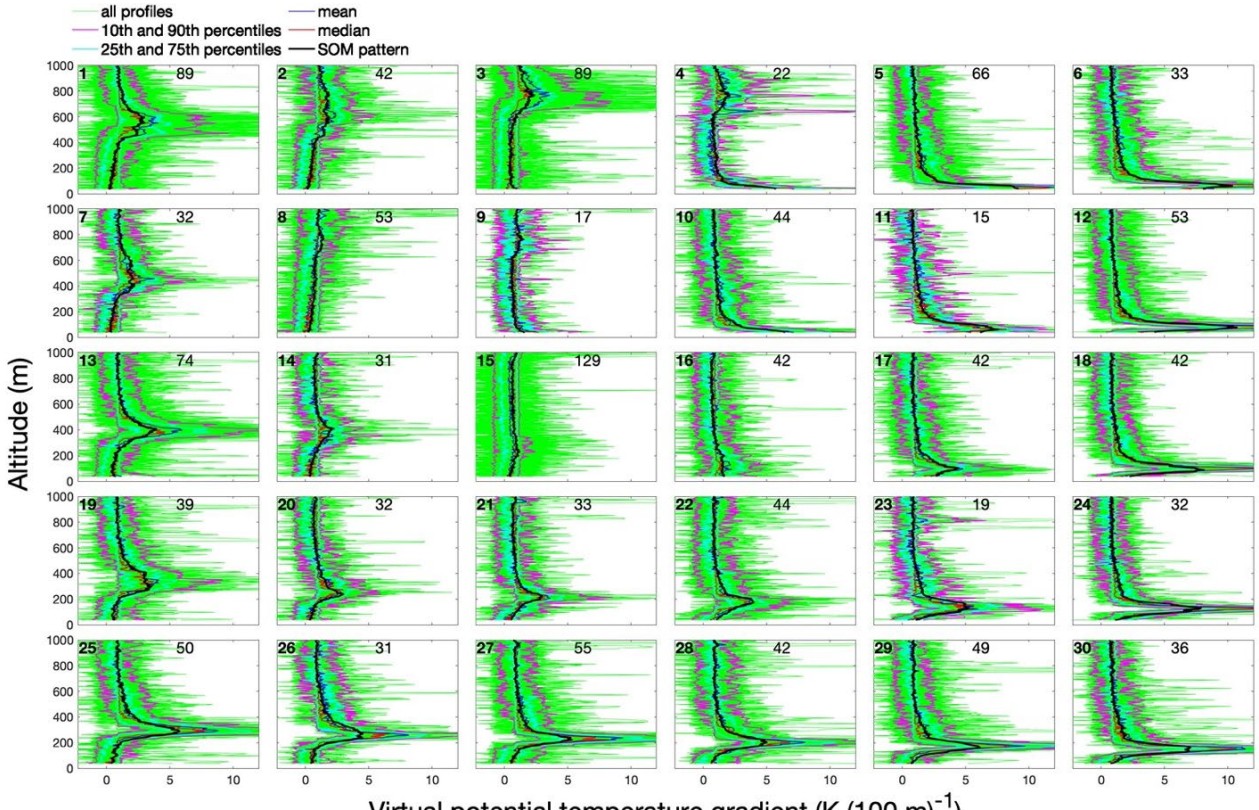

**Figure 2.** The 30 SOM patterns resulting from training with 1377 MOSAiC $d\theta_v/dz$ profiles. On each subplot, the black line is the SOM defined profile of $d\theta_v/dz$. Individual radiosonde profiles of $d\theta_v/dz$ mapped to each pattern are shown in green. The $10^{th}$ and $90^{th}$ percentiles (pink) and the $25^{th}$ and $75^{th}$ percentiles (light blue) are shown for all radiosonde data that maps to each pattern. The mean $d\theta_v/dz$ profile is shown in dark blue and the median $d\theta_v/dz$ profile is shown in red for each pattern. The bold number in the upper lefthand corner of each subplot is the number of that pattern (1 through 30), and the number in the upper center of each subplot is the number of radiosonde profiles which map to that pattern.

Next, DH2 profiles were mapped to the SOM by determining the SOM pattern with the least squared difference when compared to the DH2 profile, above 35 m. While some DH2 profiles extend to nearly 1 km altitude, many of them do not extend that high (if the CBH was lower than 1 km, or other operational limitations prevented the DH2 from making a full profile), so in some cases the DH2 did not sample stability aloft and can only be attributed to a SOM pattern based on the low altitude measurements. Mapping the DH2 profiles to the SOM reveals which ABL patterns were sampled by the DH2 to understand if the more limited sampling abilities of the DH2 were able to capture the range of ABL structures present. This analysis also documents the range of ABL depth and stability sampled by the DH2 in order to facilitate future research utilizing the unique measurements (e.g., turbulence) of the DH2.





**2.4 Stability regime analysis**

To more easily visualize the ABL structures represented by the SOM, profiles of $\theta_v$ anomaly are shown for each SOM pattern (Fig. 3). The $\theta_v$ anomalies were calculated as differences from $\theta_v$ at 1 km. Anomalies are shown to remove the large annual range of $\theta_v$ that would otherwise be present in the data while preserving the profile shape, which defines
the stability and ABL height (Stull, 1988). Thus, the average $\theta_v$ anomaly profile for all radiosondes that map to each SOM pattern is shown in Fig. 3. The 30 SOM patterns reveal the range of stability regimes present in the observations, from strongly stable to near-neutral.

Twelve stability regimes have been defined based on the near-surface stability within the ABL as well as the stability between the top of the ABL and 1 km (Table 2). These stability regime definitions are based on the range of ABL
profiles seen in the SOM (Fig. 3) and were applied to each SOM pattern (using the average of all radiosonde profiles mapped to a given SOM pattern) as well as to individual radiosonde profiles. The stability regime definitions allow us to explore the frequency, both annually and seasonally, of different stability types and how ABL, LLJ, TI and atmospheric moisture vary with stability. The stability regime definitions can be applied to datasets from other regions allowing for comparison of the MOSAiC results to ABL stability elsewhere (Dice et al., in prep).

The first step in identifying stability regime is to smooth some of the noise in the original $d\theta_v/dz$ profiles. Since our stability criteria in part depend on near-surface $d\theta_v/dz$, within the ABL, and some observations have an ABL height as low as 50 m, we first include a measurement of $d\theta_v/dz$ over 15 m between 35 m (lowest point of the profile) and 50 m. For values at and above 50 m, $d\theta_v/dz$ is calculated across 30 m intervals in steps of 5 m and attributed to the center altitude of $\Delta z$ (i.e., 35-65 m, 40-70 m, 45-75 m and so on, resulting in a $d\theta_v/dz$ profile with values at 42.5 m,
50 m, 55 m, 60 m AGL, and so on).

The thresholds used to define each stability regime are based on the range of stability patterns identified in the SOM (Fig. 2) as well as a similar SOM-based analysis of ABL profiles in Antarctica (Dice et al., in prep). Table 2 shows the thresholds associated with each stability regime, and how they are applied. The first step for stability regime identification is to identify the near-surface stability using the $d\theta_v/dz$ value at 42.5 m, as this value is representative
of stability within the ABL. The possible near-surface regimes are strongly stable (SS), moderately stable (MS), weakly stable (WS) and near-neutral (NN). To differentiate between stable cases (SS, MS, or WS) and near-neutral cases (NN), we use a threshold of $0.5 \, \text{K} \, (100 \, \text{m})^{-1}$, where if $d\theta_v/dz$ below 50 m is less than the threshold, it is considered NN, and if it is greater than or equal to the threshold, it is stable. This threshold was chosen, as it equates to the threshold of 0.2 K over 40 m used to discern a stable versus neutral ABL in Jozef et al. (2022). Additional thresholds
were derived to differentiate SS, MS, and WS. While a range of thresholds were tested, the ones listed in Table 2 were determined to best discern meaningful differences in near-surface $\theta_v$ inversion strength for both the MOSAiC data presented here as well as radiosonde profiles at several sites in Antarctica (work currently in preparation for publication).



The second step for stability regime identification is only applied to cases with a near-surface regime of WS or NN
and is carried out to differentiate weakly stable or near-neutral cases (both considered relatively well-mixed) that are
very shallow, from those that are deeper. We make this distinction because we hypothesize that there are different
processes that would lead to a shallow versus deep well-mixed layer. Thus, if ABL height is less than 125 m, we
consider this a very shallow mixed (VSM) case. This threshold of 125 m was chosen, as there is a cluster of SOM
patterns with near-surface regime of WS or NN that have ABL height less than 125 m, and a jump in height before
the next cluster of SOM patterns with ABL height above 125 m.

Lastly, stability aloft is determined. This step is only applied to VSM, WS, and NN cases, as we only address stability
aloft if it is more stable than the near-surface stability regime. For SS and MS cases, the profile is at its most stable
near the surface, and transitions to the free atmosphere above the ABL, so stability aloft does not provide additional
information. Using the maximum in the $d\theta_v/dz$ profile above the ABL, but below 1 km, the same thresholds as were
applied to identify the near-surface regime are also applied to identify stability aloft, where the options are strongly
stable aloft (SSA), moderately stable aloft (MSA), and weakly stable aloft (WSA). All of the resulting options for
stability regime are listed in Table 2. These regimes are color coded with the colors that will be used to discern each
regime for the remainder of the paper.

**Table 2:** Thresholds used to differentiate between stability regime, where the various near-surface regimes are SS
(strongly stable), MS (moderately stable), VSM (very shallow mixed), WS (weakly stable) and NN (near-neutral),
and the various stabilities aloft are SSA (strongly stable aloft), MSA (moderately stable aloft), and WSA (weakly
stable aloft).

| $d\theta_v/dz$ at 42.5 m AGL | ABL Height | Max. $d\theta_v/dz$ above ABL | Stability Regime | Abbreviation |
|---|---|---|---|---|
| $\geq 5$ K (100 m)$^{-1}$ | - | - | Strongly Stable | SS |
| $\geq 1.75$ K (100 m)$^{-1}$ $< 5$ K (100 m)$^{-1}$ | - | - | Moderately Stable | MS |
| $< 1.75$ K (100 m)$^{-1}$ | $< 125$ m | $\geq 5$ K (100 m)$^{-1}$ | Very Shallow Mixed – Strongly Stable Aloft | VSM-SSA |
| | | $\geq 1.75$ K (100 m)$^{-1}$ $< 5$ K (100 m)$^{-1}$ | Very Shallow Mixed – Moderately Stable Aloft | VSM-MSA |
| | | $< 1.75$ K (100 m)$^{-1}$ | Very Shallow Mixed – Weakly Stable Aloft | VSM-WSA |
| $\geq 0.5$ K (100 m)$^{-1}$ $< 1.75$ K (100 m)$^{-1}$ | $\geq 125$ m | $\geq 5$ K (100 m)$^{-1}$ | Weakly Stable – Strongly Stable Aloft | WS-SSA |
| | | $\geq 1.75$ K (100 m)$^{-1}$ $< 5$ K (100 m)$^{-1}$ | Weakly Stable – Moderately Stable Aloft | WS-MSA |
| | | $< 1.75$ K (100 m)$^{-1}$ | Weakly Stable | WS |
| $< 0.5$ K (100 m)$^{-1}$ | | $\geq 5$ K (100 m)$^{-1}$ | Near-Neutral – Strongly Stable Aloft | NN-SSA |
| | | $\geq 1.75$ K (100 m)$^{-1}$ $< 5$ K (100 m)$^{-1}$ | Near-Neutral – Moderately Stable Aloft | NN-MSA |
| | | $\geq 0.5$ K (100 m)$^{-1}$ $< 1.75$ K (100 m)$^{-1}$ | Near-Neutral – Weakly Stable Aloft | NN-WSA |
| | | $< 0.5$ K (100 m)$^{-1}$ | Near-Neutral | NN |





## 3 Results and discussion

### 3.1 Range of vertical ABL structure

The $\theta_v$ anomaly profiles for each SOM pattern, shown in Fig. 3, reveal the annual range of ABL stability and height in the central Arctic during the MOSAiC year. The average $d\theta_v/dz$ profile for each SOM pattern is also shown in Fig. 3, calculated as described in Sect. 2.4, along with the corresponding stability regime identified for each SOM pattern. Three stability regimes listed in Table 2 (VSM-WSA, WS, and NN) are not represented by a SOM pattern, since

VSM-WSA and WS occur infrequently, and NN was never observed in an individual profile. The SOM shows the continuum of ABL vertical structure, with each pattern having a smooth transition to those adjacent, such that similar types of ABL structure are situated in the same section of the SOM. The patterns with stronger stability are located on the right half of the SOM, with the $\theta_v$ inversion at or near the surface (SS and MS cases) in the upper right of the SOM, and the $\theta_v$ inversion becoming more elevated moving to the lower right of the SOM (VSM cases). The weaker

stability and near-neutral patterns are located on the left half of the SOM, with decreasing stability and increasing depth of the well-mixed layer, moving from the bottom left (largely WS) to the top left (largely NN) of the SOM. Thus, the ABL during MOSAiC revealed by the SOM spanned from very shallow and stable, with a strong surface-based $\theta_v$ inversion, to deep and near-neutral, with a weak elevated $\theta_v$ inversion.



**Figure 3.** Profiles of mean virtual potential temperature ($\theta_v$) anomaly (orange line, bottom x-axis) for all radiosonde profiles mapped to each SOM pattern. Profiles of mean virtual potential temperature gradient ($d\theta_v/dz$) for all radiosonde profiles (magenta line) and SOM-defined $d\theta_v/dz$ profiles (black line) (top x-axis). Vertical and horizontal black lines in each subplot indicate the various thresholds used to determine stability regime, following the "Lines Key," and the horizontal red line in each subplot is the average ABL height for that pattern. The bold number in the upper lefthand corner of each subplot is the number of that pattern (1 through 30), the number in the upper center of each subplot is the number of radiosonde profiles which map to that pattern, and the letters in the upper righthand corner of each subplot indicates that pattern's stability regime. Stability regime is also indicated by the color of the border for each subplot, following the colors given in the "Stability Regime Key".

The seasonal breakdown of SOM pattern frequency is displayed in Fig. 4, which shows, for each SOM pattern, the frequency of cases which occurred in a given season. For example, 27% of all radiosondes that map to pattern 1 occurred in the fall (Fig. 4a). Observations in the fall most heavily contribute to the SOM patterns in the center and left of the grid (patterns 2, 8, 15, 20, and 22), as greater than 40% of all profiles in each of these patterns are observations that occurred in the fall. These are largely patterns with well-mixed near-surface layer, with moderate to



strong stability aloft. Observations in the winter most heavily contribute to the SOM patterns in the far right and the bottom of the grid (patterns 5, 6, 12, 18, and 23 to 30). For each of these patterns, greater than 35% (and greater than 50% for some patterns) of observations occurred in the winter. These are largely patterns with a surface based $\theta_v$ inversion, or a shallow well-mixed layer capped by a strong $\theta_v$ inversion.

Observations in the spring are more evenly distributed among all SOM patterns than any other season, as no SOM pattern contains greater than 36% of the total observations. The least common SOM patterns for spring are in the upper right of the grid (patterns 4, 6, and 18), which all have $\theta_v$ inversion down to at least 35 m (the lowest measurement altitude). Lastly, observations in summer most heavily contribute to two SOM patterns in the upper right of the grid (patterns 4 and 17), which each contain 55% of the annual observations. Pattern 4 is particularly interesting, as there is an elevated region of enhanced stability around 600 m AGL, which must be explained by unique processes occurring primarily in summer, perhaps a low fog layer and additional elevated cloud layer. Two patterns on the left side of the SOM (7 and 21) are also dominated by summer observations, as they each contain at least 39% of the annual observations. The annual frequencies of the SOM patterns are demonstrated with Supplementary Fig. S1, which shows that annual frequencies are quite variable from one pattern to the next, with the SOM pattern with the highest frequency only accounting for 9.4% of MOSAiC observations (pattern 15). As 13 of the SOM patterns depict strong, moderate, or weak near-surface stability, this shows that the ABL in the Arctic is often stable (Esau and Sorokina, 2010). However, as 10 of the SOM patterns depict near-neutral near-surface stability (the remaining seven patterns depicted are VSM profiles), this presents a new finding that a near-neutral ABL is perhaps more prevalent than previously known, disagreeing with Esau and Sorokina (2010), which found the Arctic ABL to be stably stratified 70 – 100% of the time.



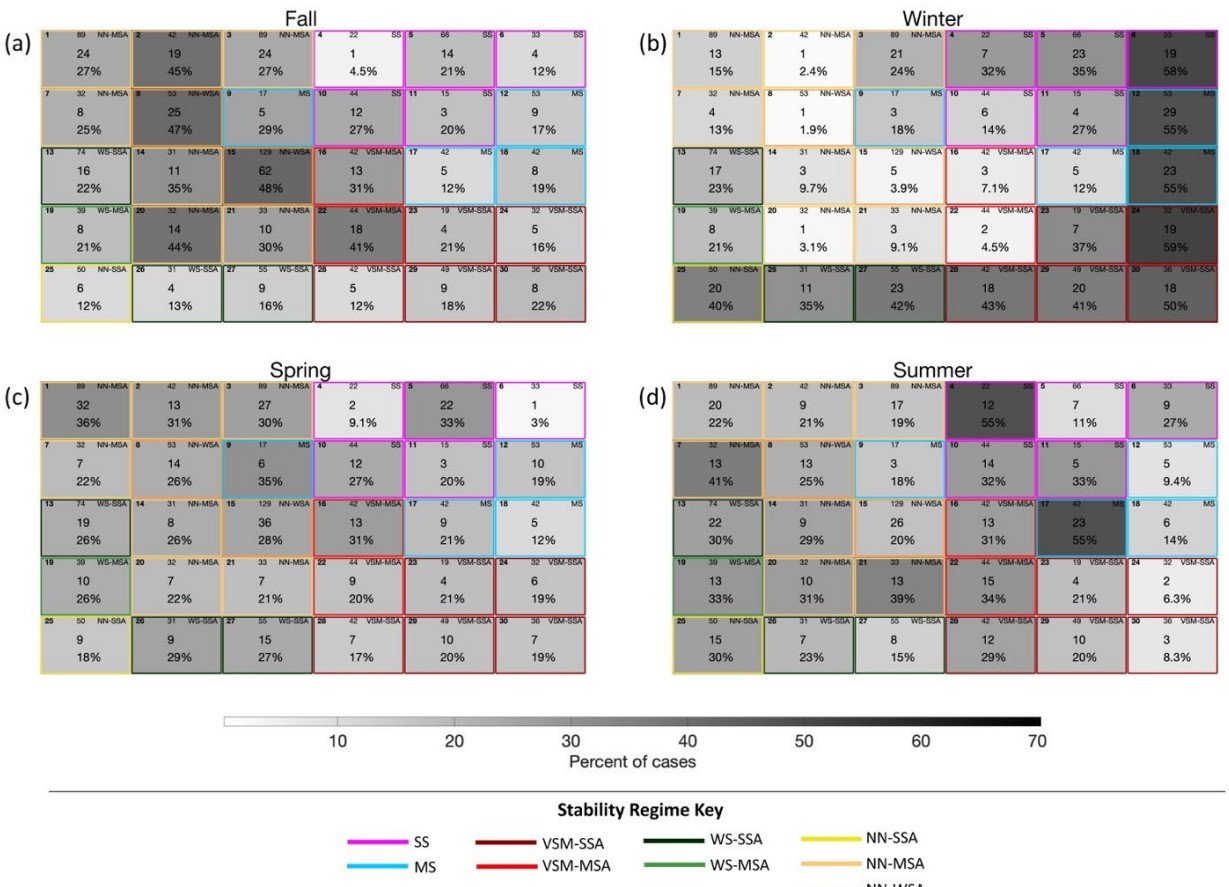

**Figure 4.** Grid plots following the same layout as the SOM indicating the number (upper middle of each subplot) and frequency (lower middle of each subplot) of radiosonde profiles that map to each pattern during (a) fall, (b) winter, (c) spring, and (d) summer. The greyscale color bar corresponds to the percent of cases in each pattern, where darker grey signifies a higher percent of cases. The bold number in the upper lefthand corner of each subplot is the number of that pattern (1 through 30), the number in the upper center of each subplot is the number of radiosonde profiles which map to that pattern, and the letters in the upper righthand corner of each subplot indicates that pattern's stability regime. Stability regime is also indicated by the color of the border for each subplot, following the colors given in the "Stability Regime Key".

DH2 profiles mapped to the SOM (Supplementary Fig. S2a) show that all but four SOM patterns (8, 19, 28, and 29) were sampled by the DH2. Thus, we have DH2 data covering nearly all the ABL vertical $\theta_v$ structures observed throughout MOSAiC. Supplementary Fig. S2b-c shows the frequencies each SOM pattern observed by the DH2, as well as by the radiosondes during the time of DH2 deployment (23 March – 26 July 2020). These results reveal that, while the DH2 did sample nearly all ABL structures represented by the SOM, the frequencies of these regimes sampled by the two measurement platforms are different, and this can likely be attributed to the DH2 sampling limitations.



To understand the influence of mechanical mixing on stability, we compare wind speed profiles and LLJ characteristics for each pattern (Fig. 5). As Fig. 5 depicts average wind speed profiles for all radiosondes in each SOM pattern, and LLJ core height varies for each case, an LLJ does not always appear in the average wind speed profile. While the LLJ height was similar across all SOM patterns (i.e., roughly 400 m AGL), the higher ABL heights of the weaker stability patterns (WS and NN; on the left side of the SOM) placed the LLJ closer to the ABL top than for the stronger stability patterns with lower ABL heights (SS, MS and VSM; on the right side of the SOM). Thus the LLJ is more closely coupled to the ABL in the weak stability cases when the LLJ core is closer to the top of the ABL. Additionally, the LLJ speeds, 2 m wind speeds, and overall wind speed profiles have greater values for the patterns on the left half of the SOM (mean LLJ speed of 12.3 m s$^{-1}$ and mean 2 m wind speed of 5.3 m s$^{-1}$), compared to the right half (mean LLJ speed of 9.7 m s$^{-1}$ and mean 2 m wind speed of 3.3 m s$^{-1}$). This supports the previous finding that faster wind speeds work to weaken stability in the ABL through mechanical generation of turbulence (Banta, 2008). Lastly, while some of the LLJ characteristics between the left and right halves of the SOM are different, the frequencies of LLJs throughout all SOM patterns are similar, with all SOM patterns showing that an LLJ was present for 67% – 84% of all observations mapped to that pattern, with a median LLJ frequency of 76%.

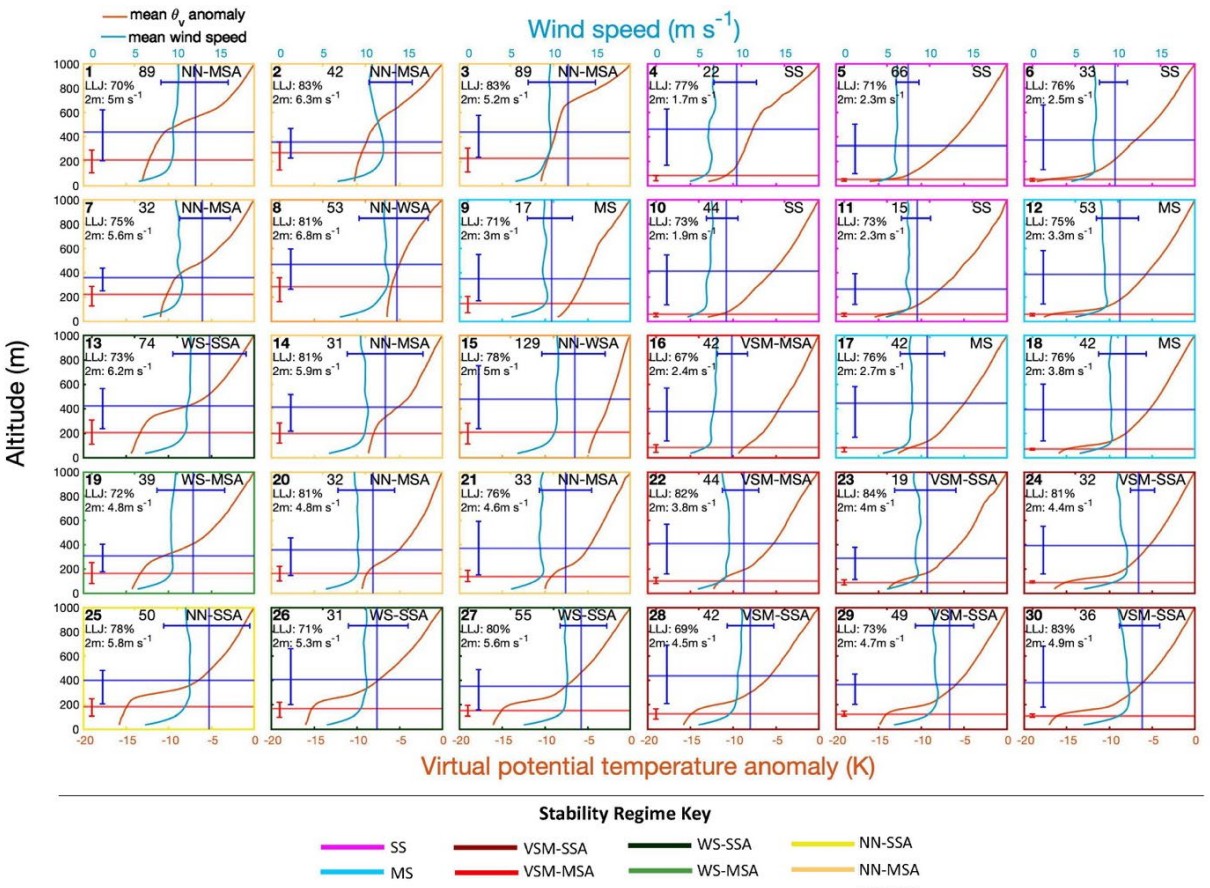

**Figure 5.** Profiles of mean virtual potential temperature ($\theta_v$) anomaly profiles (orange line, bottom x-axis) for all radiosonde profiles mapped to each SOM pattern. Profiles of mean wind speed for all radiosonde profiles (light blue line, top x-axis). The horizontal red line in each subplot is the average ABL height, with the red error bar indicating the 25th and 75th percentiles of ABL height. The horizontal blue line in each subplot is the average LLJ core height, with the vertically oriented error bar indicating the 25th and 75th percentiles of LLJ core height. The vertical blue line in each subplot is the average LLJ speed, with the horizontally oriented error bar indicating the 25th and 75th percentiles of LLJ speed. Each subplot also has written the frequency of LLJs and average 2 m wind speed. The bold number in the upper lefthand corner of each subplot is the number of that pattern (1 through 30), the number in the upper center of each subplot is the number of radiosonde profiles which map to that pattern, and the letters in the upper righthand corner of each subplot indicates that pattern's stability regime. Stability regime is also indicated by the color of the border for each subplot, following the colors given in the "Stability Regime Key".

**3.2 Frequency of stability regimes**

The analysis below transitions from the SOM-based perspective in the previous section to focus on stability regimes (as defined by Table 2) based on each individual radiosonde profile. The annual and seasonal frequency of each stability regime is shown in Fig. 6a. For the stability regime frequencies shown in Fig. 6 and subsequent figures, the regimes are organized from strongest to weakest near-surface stability going from left to right (where VSM is





considered more stable than WS due to a shallower ABL), and within a given near-surface regime, the aloft regimes are also organized such that stability decreases from left to right.

Annually, the stability regime which occurred with the highest frequency is NN-SSA followed by VSM-SSA. In decreasing order, MS, SS, and NN-MSA, also occurred with high frequency. VSM-MSA occurred with moderate frequency, and VSM-WSA, NN-WSA, and all WS regimes were relatively infrequent. The high frequency of regimes with either moderate or strong stability near the surface, or a well-mixed ABL with strong stability aloft, suggests that the central Arctic lower atmosphere trends towards being strongly stable, but sometimes the near-surface atmosphere

can become well-mixed due to the generation of turbulence. In fall, the strongest stability regimes (SS and MS) were less frequent, while NN was more frequent, which could be due to a combination of the thinner sea ice resulting in more upward heat transfer from the ocean to the atmosphere, and a higher frequency of low-level liquid-bearing clouds (Shupe et al., 2011a; Shupe et al., 2011b), which both weaken ABL stability. Of all seasons, the winter stability regime frequency distribution is most different from the annual results. Winter had a higher frequency of the strongest stability

regimes (SS, MS, and VSM-SSA), and the NN regime was dominated by NN-SSA. Thus, there is a clear dominance of stronger stability in winter compared to other seasons due to the lack of solar radiation, which allows for longwave cooling of the surface, resulting in strong stability near the surface. In spring, the relative frequencies of stability regime exactly match the pattern that is seen annually. Lastly, in summer, the relative frequencies of SS, MS, VSM-SSA, VSM-MSA, NN-SSA, and NN-MSA were similar, which suggests that the forcing mechanisms of each of these

regimes occurred with similar frequency, or that certain regimes may occur under a range of forcing mechanisms such that one stability regime doesn't heavily dominate over the others.

    When comparing the frequencies of the various stability regimes observed by the DH2 to those observed by the radiosondes during the time period in which the DH2 was deployed (Fig. 6b), we see that the DH2 observed similar relative frequencies as did the radiosondes. The primary difference is that the DH2 appears to have observed much

lower frequencies of the various near-neutral regimes with stability aloft, and a higher frequency of purely near-neutral (where the radiosonde did not observe any purely NN cases). Thus, even though the operational scope of the DH2 was more limited, compared to that of the radiosondes, the DH2 was still largely able to capture the relative frequencies of stability regimes between the end of May and the end of July. However, we also demonstrate a limitation of the DH2 observations, which is that because the DH2 was not always able to be deployed up to a full 1 km if weather or

other operational obstacles interfered, then the DH2 sometimes did not capture the stability aloft, which is why we see instances of NN observed by the DH2, but not by the radiosondes.

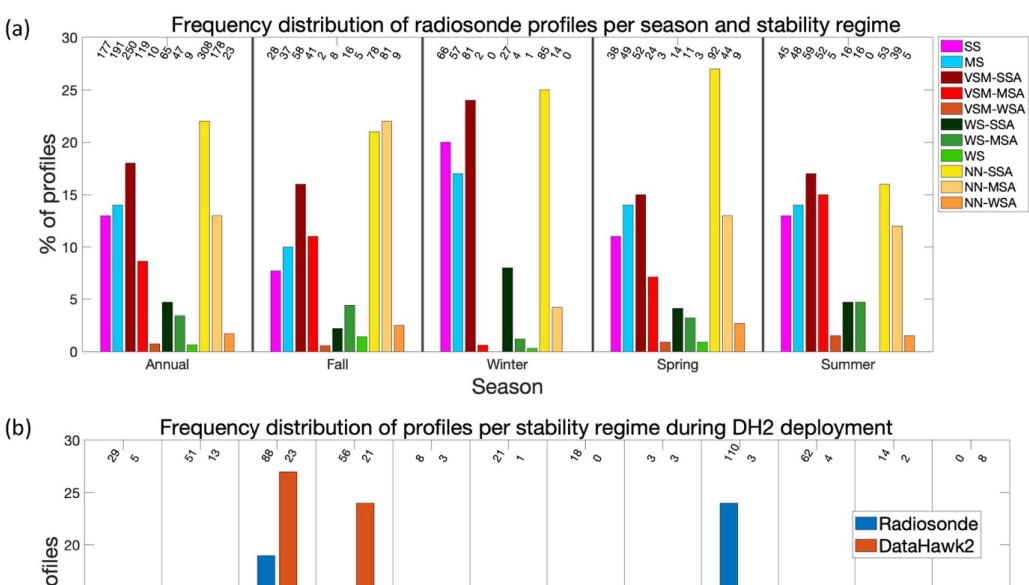

**Figure 6.** (a) Frequency distribution showing the percent of radiosonde profiles in each stability regime, annually and seasonally. For the seasonal sections, the percent shown is with respect to the total number of radiosonde profiles in that season. The numbers along the top of the plot, above each bar, indicate the total number of radiosonde profiles of that stability regime and season. (b) Frequency distribution showing the percent of radiosonde (blue) and DataHawk2 (orange) profiles in each stability regime, during the time period in which the DataHawk2 was operated (23 March – 26 July 2020), with respect to the total number of radiosonde or DataHawk2 profiles in that time period. The numbers along the top of the plot, above each bar, indicate the total number of radiosonde and DataHawk2 profiles of that stability regime during the time period the DataHawk2 was flown.

### 3.3 Variability of atmospheric state as a function of stability regime

#### 3.3.1 Atmospheric boundary layer

Here, we characterize the structure and stability of the ABL by the ABL height, change in $\theta_v$ over the depth of the ABL ($d\theta_v/dz$), horizontal wind shear over the depth of the ABL ($dV/dz$), average $Ri_b$ within the ABL, and bulk $u_*$. The annual range of values of each of these variables for each stability regime is shown in Fig. 7. Supplementary Fig. S3 indicates when there is a statistically significant difference at the 5 % significance level between the mean values of each variable between all pairs of stability regimes, using a two-tailed t-test when degrees of freedom (df) $\leq 100$, and a two-tailed z-test when df > 100.



The annual mean and median of ABL height observed (Fig. 7a) were 150 m and 118 m respectively, which agrees
with Esau and Sorokina (2010). There is an increase in ABL height as stability decreases. A marked increase in ABL
height separates the shallower SS, MS, and the VSM regimes with the deeper WS and NN regimes. This noticeable
difference is in part a product of how we define the VSM regime, which requires an ABL height of 125 m or less,
however the fact that we see this drastic increase also supports the choice of this threshold. Thus, ABL height increases
as stability weakens, supporting the well-recognized notion that the turbulent processes that weaken stability in the
ABL also work to deepen the ABL (Stull, 1988). Additionally, we find that within the near-surface regimes that have
specified stability aloft (VSM, WS, and NN), the ABL heights are similar, regardless of stability aloft, though ABL
height does slightly increase as stability aloft decreases within each of these near-surface regimes.

$d\theta_v/dz$ and $dV/dz$ within the ABL tell us about the two components of $Ri_b$, buoyancy and shear respectively, which
are the processes controlling turbulence. $d\theta_v/dz$ responds to both heating from below (or cooling from above), as well
as mechanical mixing. There is a notable difference in $d\theta_v/dz$ between SS and MS (values largely above average), and
all other regimes, with SS having the largest $d\theta_v/dz$ value, consistent with a stable ABL. These two regimes also had
the highest amounts (also largely above average) of wind shear ($dV/dz$) within the ABL (Fig. 7c), which suggests that
for these observations, static stability suppressed turbulence (though did not completely eliminate it), so the ABL
remained stable despite high amounts of wind shear. Significant differences in $d\theta_v/dz$ and $dV/dz$ between most pairs
of stability regimes (Fig. S3a) reveal that the buoyant and mechanical processes that generate or suppress turbulence
are distinct for each regime, which supports the idea that how we defined the regimes is physically meaningful.
Additionally, the $dV/dz$ results suggest that when stability is weaker, winds vary less with height due to greater mixing,
which is a common behavior of winds within the ABL (Wallace and Hobbs, 2006).

$Ri_b$ is an indication of the likelihood of turbulence to form in a laminar atmosphere, with a lower $Ri_b$ value indicative
of a greater probability of turbulence. $Ri_b$ within the ABL had higher values for SS and MS than the rest of the regimes,
which supports the expectation that there is less turbulence within a more strongly stable ABL. However,
Supplementary Fig. S3b shows that there is little significant difference between mean $Ri_b$ from the various stability
regimes. This is likely resulting from the wide range of values for SS and MS $Ri_b$, much of which overlap with the $Ri_b$
values from all other regimes. Thus, we conclude that the near-surface atmosphere is essentially always turbulent
(however minimally in the stronger stability cases), but we need to look at $u_*$ to fully understand the difference in
turbulence between the various regimes.

The Arctic environment is often characterized by turbulence. To identify turbulent periods we use bulk $u_*$, where a
greater $u_*$ value indicates more turbulence. Figure 7e shows that $u_*$ and thus turbulence increases with decreasing
stability. This is supported by Supplementary Fig. S3b, which shows that $u_*$ is largely significantly different across
regimes, except for within the various NN regimes. As $u_*$ was measured from the met tower independent from the
radiosondes, the strong trend in $u_*$ versus stability regime again shows that the stability regime criteria and definitions
applied to the radiosondes were well-chosen. The fact that $u_*$ values for SS and MS are relatively low, while wind
shear values are relatively high, supports the conclusion that static stability is largely impacting the strength of



turbulence for these regimes. Lastly, within the VSM, WS, and NN regimes, $u_*$ decreases with weakened stability
aloft, which suggests that it takes more mechanically generated turbulence to mix out the near-surface layer when
stability aloft is greater.

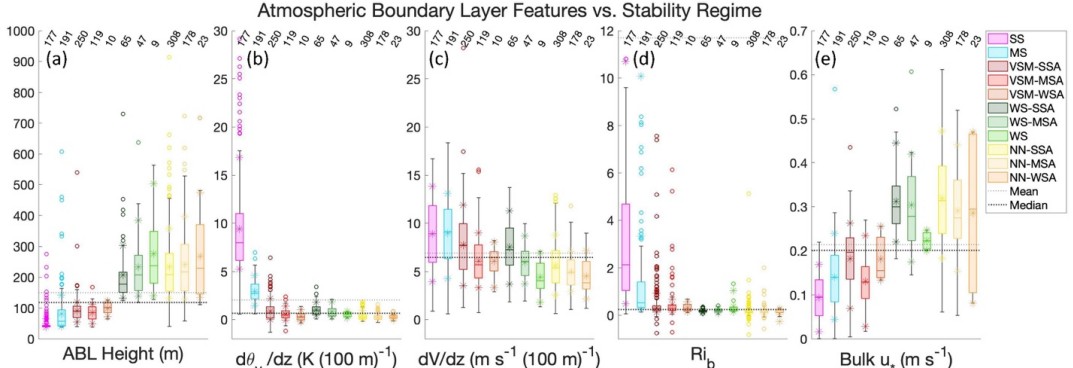

**Figure 7.** Box and whisker plots showing the range of (a) ABL height, (b) virtual potential temperature gradient over
the depth of the ABL, (c) wind speed gradient over the depth of the ABL, (d) average $Ri_b$ within the ABL, and (e)
bulk $u_*$ from the meteorological tower for each stability regime over the entire year. The center line of each box is the
median, and the outer edges of the boxes are the upper and lower quartiles. The whiskers show the range of values
within 1.5 times the interquartile range from the top or bottom of the box, and outliers are shown with hollow circles.
Asterisks are included at the mean, 10th percentile, and 90th percentile. Horizontal dotted black lines show the annual
mean (light dotted) and median (heavy dotted) values of each variable. The number of cases in each stability regime
are given along the top of the figure.

### 3.3.2 Low-level jets

LLJs are common in the Arctic, as demonstrated by Supplementary Fig. S4, which shows that annually, 76% of
radiosondes observations contained an LLJ, with a similar overall frequency when separated by season. There was a
similar frequency of LLJs for all stability regimes, of least about 70%, highlighting the fact that an LLJ can both be a
cause and an effect of stability (the only exception is the WS regime which had an annual LLJ frequency of about
45% due to the fact that some of the WS cases had high wind speeds throughout the entire profile). For a well-mixed
or weakly stable ABL, LLJs often contribute to the creation of the mechanical turbulence that mixes the ABL (Banta
et al., 2003). For more strongly stable ABLs, an LLJ can be an effect of the strong stability if the above atmosphere
becomes decoupled from the surface. Annually, the stronger stability regimes (SS, MS, and VSM) were dominated
by slow (speed ≤ 10 m s⁻¹) and moderate (10 m s⁻¹ < LLJ speed ≤ 20 m s⁻¹) LLJs, and the weaker stability regimes
(WS and NN) were dominated by moderate and fast (LLJ speed > 20 m s⁻¹) LLJs. The seasonal LLJ frequencies per
regime were largely similar to the annual frequencies. However, in winter the frequency of slow LLJs was less than
the annual results, and in summer the frequency of slow LLJs was more than the annual results, for most stability
regimes.

LLJs interact with the ABL, and thus the LLJ characteristics may help to explain the different stability regimes. Here
we tested LLJ core height, vertical distance between the LLJ core height and ABL height, and LLJ depth and speed.



The annual range of values of each of these LLJ variables for each stability regime is shown in Fig. 8 (refer to Supplementary Fig. S5 for corresponding significance testing). The annual mean and median of LLJ core height were 401 and 318 m respectively, and the annual mean and median of LLJ depth were 428 and 366 m. This LLJ core height is consistent with the findings of Jakobson et al. (2013), but the LLJ depth is slightly less than was found in Jakobson et al. (2013). Interestingly, there does not appear to be much difference in LLJ core height (Fig. 8a) or LLJ depth (Fig. 8c) for the various stability regimes, which is supported by the fact that there are no two stability regimes for which these variables are significantly different (Fig. S5a and S5b).

SS, MS, and the VSM regimes largely had LLJs which were situated above the ABL (Fig. 8b), with a mean distance of 290 to 329 m. The WS and NN regimes, which also had faster LLJs (Fig. 8d), largely had LLJs which were situated much closer to the ABL (mean distance of 73 to 214 m), and in the case of WS, had a median value of the LLJ core height being within the ABL. This finding that a faster LLJ is more likely to have its core within the ABL agrees with the findings of Jakobson et al. (2013). The greater difference between ABL height and LLJ height in the case of SS, MS, and the VSM regimes suggests decoupling between the relatively stable ABL and the LLJ, which is consistent with inertial oscillations (Jakobson et al., 2013; Blackadar 1957). The smaller difference between ABL height and LLJ height for the WS and NN regimes suggests greater coupling between the well-mixed ABL and the LLJ, so inertial oscillations are unlikely to be the formation mechanism, and rather baroclinicity is the more probable cause. The similarity in LLJ core height across the different stability regimes also supports these two different formation mechanisms.

The strongest relationship is between LLJ speed and stability regime, supported by Fig. S5b which shows a large number of regime pairs that are significantly different from each other. The annual mean and median of LLJ speed were 11.5 and 10.8 m s$^{-1}$ respectively which is greater than the average LLJ speed found in Jakobson et al. (2013). There is a step change increase in LLJ speed from SS, MS, and the VSM regimes to the WS and NN regimes, which suggests that LLJ speeds differentiate between stronger and weaker stability ABLs, where faster speeds produce weaker stability ABLs. Figure S5b shows that there is a significant difference in LLJ speed when comparing the stronger stability regimes to the weaker stability regimes, but not much significant difference between regimes with similar stability. Within each near-surface regime that has various aloft categories (VSM, WS, and NN), LLJ speed was slower for weaker stability aloft. This supports the hypothesis that when stability above the ABL is greater, more mechanical turbulence is required to mix out the near-surface layer. Alternatively, stronger stability aloft could be associated with stronger LLJs due to differences in LLJ forcing mechanisms.

The annual frequency, height, and speed of LLJs found in this study exceed those found in Lopez-Garcia et al. (2022). This is likely because faster LLJs, which typically occur at a slightly higher altitude, may not have a jet core speed that is at least 25% faster than the wind speed minimum above the LLJ core, and such cases were not considered in Lopez-Garcia et al. (2022). However, the LLJs identified here can still be important, because even if the wind speeds are fast throughout the entire profile up to 1.5 km (for example, during a storm), a LLJ can still work to produce excess turbulence in the ABL, and thus should not be ignored.

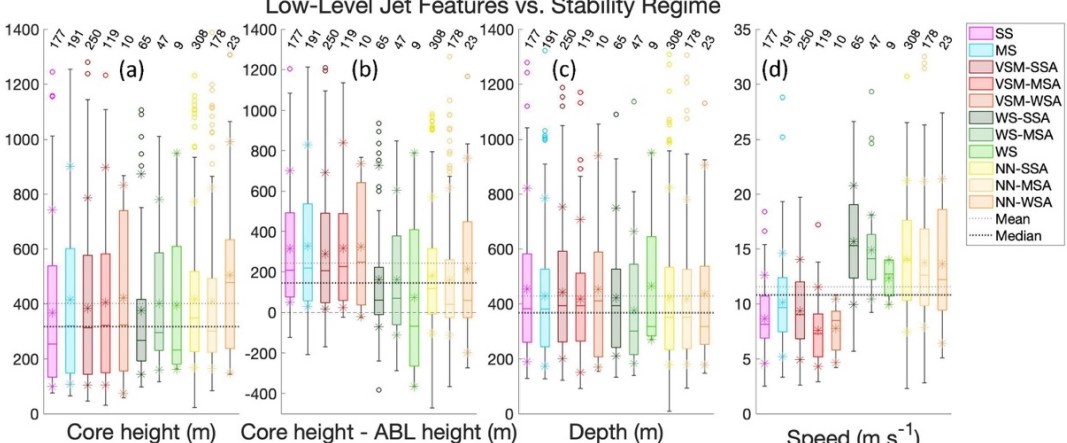

**Figure 8.** Box and whisker plots showing the range of (a) LLJ core height, (b) the difference between the LLJ core height and ABL height, where a negative value indicates an LLJ core below the ABL top, (c) LLJ depth, and (d) LLJ speed, for each stability regime over the entire year. The center line of each box is the median, and the outer ranges of the boxes are the upper and lower quartiles. The whiskers show the range of values within 1.5 times the interquartile range from the top or bottom of the box, and outliers are shown with hollow circles. Asterisks are included at the mean, 10th percentile, and 90th percentile. Horizontal dotted black lines show the annual mean (light dotted) and median (heavy dotted) values of each variable. The number of cases in each stability regime are written along the top of the figure.

### 3.3.3 Temperature inversions

TIs are also common in the central Arctic, as demonstrated by Supplementary Fig. S6, which shows that annually, 99.7% of radiosondes observed at least one TI below 5 km, which is a greater frequency than was found in Gilson et al. (2018), but roughly in agreement with the findings of Kahl (1990). The only regimes that had TIs less than 100% of the time were VSM-MSA, WS, and NN-WSA, which are the three regimes with weak stability aloft, and when a TI was present for these regimes, it was most often a weak TI (TI intensity $\leq 5$ °C), with no cases of a strong TIs (TI intensity $> 10$ °C). Annually and seasonally, only SS and MS, as well as VSM, WS, and NN with strong stability aloft had instances of strong TIs. The regimes with moderate and weak stability aloft only had instances of moderate (5 °C $<$ TI intensity $\leq 10$ °C) and weak TIs. This is expected, as our definition of stability regimes is in part dependent on TI intensity. In winter, the highest frequency of strong TIs was observed across regimes, followed by spring. In fall and summer, the TI intensity was largely weak or moderate.

TIs contribute to ABL formation, and thus the TI base height, vertical distance between the TI base height and ABL height, and TI depth and intensity may help to explain the different stability regimes. The lowest altitude TI most directly interacts with the surface, and thus the aforementioned TI characteristics are shown only for the lowest altitude TI. The annual range of values of each of these TI variables for each stability regime is shown in Fig. 9 (refer to Supplementary Fig. S7 for corresponding significance testing). The mean and median TI base height were 260 and 142 m, respectively, which agrees with the findings of Gilson et al. (2018). Increasing TI base height is correlated



with decreasing stability. When the regimes with weak stability aloft did have a TI, they had a much higher base altitude than TIs for all other regimes (Fig. 9a). As the base of a TI is often located at or near the top of the ABL, and we also saw increasing ABL height with decreased stability, this makes sense. However, the large jump in TI altitude for all regimes with weak stability aloft shows that TI altitude is not always the best method for determining ABL height, which agrees with Jozef et al. (2022). This is further supported by the difference between TI base height and ABL height (Fig. 9b) which shows that the TI base was just below the ABL height for SS, MS, and WS-SSA (mean difference of -7 m), and largely well above the ABL height for VSM-WSA, WS, and NN-WSA (mean difference of 1336 m), with the overall annual mean value across all regimes being 111 m. In the cases with weak stability aloft, the TI being well above the ABL suggests that there is a greater potential for the exchange of momentum, heat, and moisture between the ABL and the free atmosphere. The annual mean and median of TI depth were 215 and 180 m respectively which agrees more closely with Gilson et al. (2018) than Kahl (1990). TI depth (Fig. 9c) decreases with decreasing near surface stability, and within each near-surface regime (VSM, WS, and NN), as stability aloft decreases.

The annual mean and median values of TI intensity were 4.8 and 4.3 °C respectively, which is greater than the TI intensity described in Gilson et al. (2018), but similar to the results of Kahl (1990). TI intensity (Fig. 9d) was largely similar for SS, MS, VSM-SSA, WS-SSA, and NN-SSA (mean intensity of 5.3 to 6.8 °C), with the SS regime having the strongest intensity TIs (mean of 6.8 °C). Within each near-surface regime with categories for stability aloft (VSM, WS, and NN), TI intensity decreases with decreasing stability aloft, similar to the trend in TI depth, suggesting that TI intensity and depth are positively correlated. However, this relationship between TI intensity and stability regime is in part a result of how we define the stability regimes, since temperature goes into the calculation of $\theta_v$ and thus, $d\theta_v/dz$ is partly a function of TI intensity.

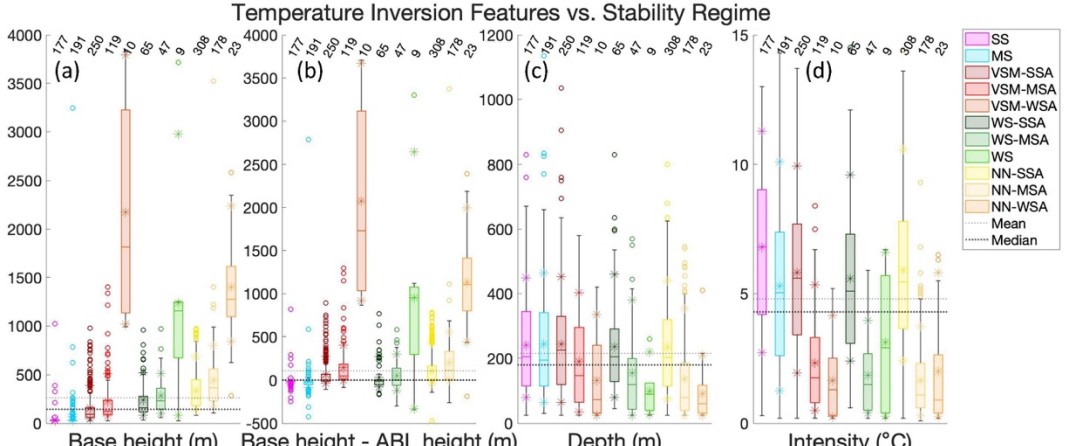

**Figure 9.** Box and whisker plots showing the range of (a) TI base height, (b) the difference between the TI base height and ABL height, where a negative value indicates a TI base below the ABL top, (c) TI depth, and (d) TI intensity for the lowest TI below 5 km, for each stability regime over the entire year. The center line of each box is the median, and the outer ranges of the boxes are the upper and lower quartiles. The whiskers show the range of values within 1.5 times the interquartile range from the top or bottom of the box, and outliers are shown with hollow circles. Asterisks are included at the mean, 10[th] percentile, and 90[th] percentile. Horizontal dotted black lines show the annual mean (light dotted) and median (heavy dotted) values of each variable. The number of cases in each stability regime are written along the top of the figure.

### 3.3.4 Atmospheric moisture

Properties of clouds and moisture can impact stability due to their radiative effect and ability to decouple below-cloud layers from the atmosphere above. Thus, cloud frequency differs by stability regime, as seen in Fig. 10a which shows the frequency distribution of whether clouds were present within 30 minutes before each radiosonde launch, depending

on stability regime and season, differentiated into the first cloud laying being a low cloud (CBH ≤ 2 km), mid-level cloud (2 km < CBH ≤ 6.1 km), or high cloud (CBH > 6.1 km). Figure 10a also shows the total frequency of clouds observed annually and seasonally, which reveals that clouds were observed to be present during 64% of radiosonde launches (a lower frequency than found in Asutosh et al. (2021) but within the frequency range found in Shupe et al. (2011b)), the majority of which were low clouds (78% of clouds observed were low clouds), with the highest seasonal

frequency of clouds during fall (78%) and the lowest seasonal frequency of clouds during spring (52%). This is likely due to the increased moisture flux from the ocean into the atmosphere, and enhanced moist air advection, which is possible when the ice is thinner and less concentrated in fall versus in spring when the ice is at its thickest. This finding agrees with Shupe et al. (2011b) on the season with highest cloud frequency, but disagrees on the season with lowest cloud frequency (Shupe et al. (2011b) found the lowest cloud frequency in winter).

Annually, SS had the lowest frequency of cases in which there were clouds within 30 minutes of radiosonde launch, followed by MS, and the WS regimes. The NN regimes had the highest frequency of cases with clouds, followed by the VSM regimes, with NN-WSA (the least stable of all regimes) having the highest percent of cases with clouds present. This suggests that the VSM and NN regimes largely occur in the presence of clouds, so are driven by the



radiative signature of clouds, whereas SS, MS, and the WS can be forced more heavily by other mechanisms. Low clouds dominate the cloud type for all regimes, but for some regimes, mid-level and high clouds were present in some cases. SS had the most instances of high clouds, with the frequency of mid-level and high clouds generally decreasing with decreased stability. When there were clouds, VSM-WSA and NN-WSA only occurred when low clouds were present. Thus, low clouds dominate for the weakest stability regimes because low clouds create more downwelling longwave radiation which favors weaker stability and may result in an ABL that is well-mixed up to the cloud base, while other regimes have mid-level and high clouds as well.

These annual patterns in cloud frequency as a function of stability are largely reflected seasonally as well, however there are some differences. In fall, there was a higher frequency of clouds for all regimes except SS. In winter, VSM-MSA, WS-MSA and WS only occurred in the presence of clouds. The latter two also had LLJs 100% of the time, which suggests that these regimes only occurred in high wind, cloudy situations, indicative of a storm, in winter. However, this may be coincidental, as the sample size of these regimes is small. In spring, the patterns are similar to the annual patterns, but there was a lower frequency of clouds for all regimes except VSM-WSA and NN-WSA. In summer, the primary difference is that there was a higher frequency of clouds for SS and MS regimes, and a lower frequency for the NN regimes. Thus, in summer, cloudy conditions were more conducive to a stronger stability, and less conducive to neutral ABL, which was not true for the rest of the year.

Some additional variables can give further insight into the influence of atmospheric moisture on stability. These are the base height of the lowest cloud layer, mixing ratio at the ABL top, LWP, and PWV. The annual range of values of each of these moisture variables for each stability regime is shown in Fig. 10b-e (refer to Supplementary Fig. S8 for corresponding significance testing).

The annual mean and median of CBH of the lowest cloud layer were 1278 and 575 m, respectively (Fig. 10b), which agrees with the findings of Asutosh et al. (2021). CBH was similar for all stability regimes, except SS, MS, and WS, which all had higher cloud bases. However, Fig. S8a shows that CBH for SS and MS regimes are significantly different than that for most other regimes, but the CBH for WS is not significantly different from the others. Thus, higher cloud base can contribute to SS and MS, but there are further contributing factors that differentiate the other regimes from each other. Mixing ratio at ABL height (Fig. 10c), LWP (Fig. 10d), and PWV (Fig. 10e) all tell a similar story in that the median values of these variables were the smallest for SS and increase with decreasing stability of the near-surface atmosphere. They also increase with decreasing stability aloft, within the different near-surface regimes. Thus, increased levels of moisture in the atmosphere can contribute both to weakening stability of the near-surface atmosphere, and the atmosphere aloft.



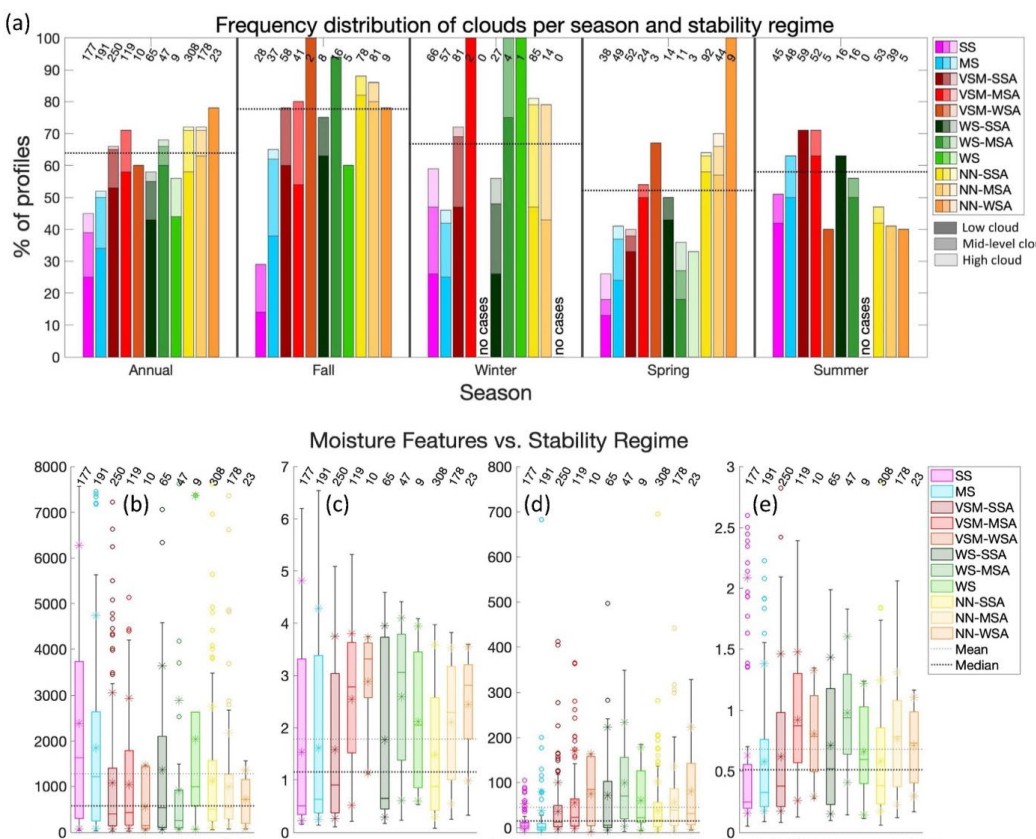

**Figure 10.** Top: (a) Annual and seasonal frequency distribution showing the percent of radiosonde profiles in each stability regime with clouds present within 30 minutes before radiosonde launch, separated into low (CBH ≤ 2 km), mid-level (2 km < CBH ≤ 6.1 km), and high (CBH > 6.1 km) clouds. For the seasonal sections, the percent shown is with respect to the total number of radiosonde profiles in that season. The numbers along the top of the plot, above each bar, indicate the total number of radiosonde profiles of that stability regime and season. The horizontal dotted black lines in each section indicate the overall frequency of clouds when considering all radiosonde observations annually, and per season. Bottom: Box and whisker plots showing the (b) first CBH, (c) mixing ratio at ABL height, (d) LWP, and (e) the PWV within 30 minutes before radiosonde launch, for each stability regime over the entire year. The center line of each box is the median, and the outer ranges of the boxes are the upper and lower quartiles. The whiskers show the range of values within 1.5 times the interquartile range from the top or bottom of the box, and outliers are shown with hollow circles. Asterisks are included at the mean, 10[th] percentile, and 90[th] percentile. Horizontal dotted black lines show the annual mean (light dotted) and median (heavy dotted) values of each variable. The number of cases in each stability regime are written along the top of the figure.

## 4 Summary and conclusions

The work presented in this paper provides an annual cycle of the vertical structure of the ABL and statistics about key thermodynamic and kinematic features of the central Arctic lower atmosphere in the context of stability regime, using data from the MOSAiC expedition. The patterns shown in Fig. 3 represent the range of $\theta_v$ structures observed over the annual cycle during MOSAiC, as revealed by the SOM, which highlights that near-surface stability spanned from



strongly stable with a surface-based temperature inversion, to near-neutral in which $\theta_v$ was approximately constant with altitude. Stability aloft, but below 1 km, was always stable, ranging from strong to weak stability. Wind speed profiles with corresponding LLJ characteristics (Fig. 5), when presented in the context of the SOM patterns reveal that LLJs core height was typically more distant from the ABL height, and thus the LLJ was likely decoupled from the ABL, for SS, MS, and VSM regimes. The WS and NN regimes correspond to faster wind speeds throughout the profile, supporting the fact that faster wind speeds work to weaken stability in the ABL through mechanical generation of turbulence. The DH2 was able to observe all but four of the SOM patterns (Fig. S2a), but the frequencies of the different patterns sampled by the DH2 differ from those sampled by the radiosondes due to the operational limitations of the DH2 (Fig. S2b-c).

When grouping the radiosonde profiles by stability regime, NN-SSA was the most frequent regime, and the least common regimes were those with weak stability aloft (VSM-WSA, WS, and NN-WSA; Fig. 6a). The prominence of a near-neutral atmosphere (37% of profiles had near-surface stability of NN) over the sea ice contradicts previous findings which state that the Arctic ABL is almost always stable (Esau and Sorokina, 2010). The high frequency of regimes with either strong or moderate stability near the surface (13% of profiles were SS and 14% were MS), or a well-mixed ABL with strong stability aloft (18% of profiles were VSM-SSA, 5% were WS-SSA, and 22% were NN-SSA), suggests that the central Arctic lower atmosphere is inclined to be strongly stable somewhere in the lowest 1 km, but the height of this strongly stable layer can become elevated, separated from the surface by a well-mixed layer, when turbulence is generated. We also found that the DH2 was largely able to capture the relative frequencies of the various stability regimes, aside from overrepresenting the amount of purely NN cases, and underrepresenting the amount of NN cases with enhanced stability aloft (Fig. 6b).

The average ABL height during MOSAiC was 150 m, and ABL height increases with decreasing stability (Fig. 7a). While strong wind shear was often present in the ABL, in many cases, the strong static stability dominated keeping the ABL stable (Fig. 7b-c). When the atmosphere aloft is very stable, there must be higher amounts of turbulence to mix out the near-surface layer, than if there is weaker stability aloft (Fig. 7e). An LLJ was observed in 76% of the radiosondes (Fig. S4), with an average height of 401 m and an average speed of 11.5 m s$^{-1}$ (Fig. 8a and 8d). For the stronger stability cases (SS, MS, and VSM), LLJs were situated an average of 311 m above the ABL (Fig. 8b), suggesting decoupling between the ABL and the LLJ, consistent with inertial oscillations. For the weaker stability cases (WS and NN), LLJs were situated an average of 172 m above the ABL (Fig. 8b), suggesting greater coupling between the ABL and the LLJ, so inertial oscillations are less likely, and rather baroclinicity is the more probable formation mechanism. These weaker stability cases had significantly faster LLJ speeds than the stronger stability cases, and thus LLJ speed is the primary LLJ characteristic driving stability.

A TI was observed in 99.7% of the radiosondes (Fig. S6), with an average base height of 260 m and an average intensity of 4.8 °C (Fig. 9a,d). In most cases of weak stability aloft, the turbulent ABL is coupled with the laminar layer above the ABL and below the TI base, allowing for the transfer of momentum, heat, and moisture between the two layers. TI base height increases as stability decreases, and TI intensity and depth both decrease as stability



decreases (Fig. 9). For some regimes (VSM-WSA, WS, and NN-WSA) there were large differences between TI base height and ABL height indicating that TI height is not always reliable as an ABL height detection metric.

Clouds were observed within 30 minutes before radiosonde launch 64% of the time (Fig. 10a). Low clouds (CBH ≤ 2 km) dominate the clouds that are observed in the central Arctic (78% of clouds observed were low clouds), with mid-level or high clouds never observed with VSM-WSA or NN-WSA regimes. The highest frequency of cloud cover within 30 minutes before radiosonde launch was observed with the NN regimes, followed by the VSM regimes. SS, followed by MS and the WS regimes, had the lowest frequency of cloud cover within 30 minutes before launch.

Weaker stability regimes are less likely to form in the presence of high clouds, whereas a stronger stability ABL can be decoupled from a high cloud layer above. CBH was on average 1278 m (Fig. 10b) but can be much higher for SS and MS regimes. The amount of atmospheric moisture generally increases with decreasing stability, so elevated moisture levels contribute to the weakening of the ABL, likely through changes in downwelling longwave radiation.

One limitation of this study is that stability regimes are based on radiosonde profiles starting at 35 m, since
measurements below this are often unreliable, so differences in stability below this height are neglected (and potentially important). A complementary paper (Jozef et al., in prep) addresses the impact of atmospheric radiative and mechanical forcings on ABL stability and how these relationships vary by season, with a focus on the peculiarities of summer processes, and thus such results are not addressed in this work. Future work will be conducted to determine how well the observed results are represented by weather and climate models. Thus, we hope that these findings serve
to help inform the improvement of parameterizations of the central Arctic in weather and climate models.

**Data availability**

The level 2 radiosonde data used in this study are available at the PANGAEA Data Publisher at https://doi.org/10.1594/PANGAEA.928656 (Maturilli et al., 2021). The DataHawk2 data are available at the National Science Foundation Arctic Data Center at https://doi.org/10.18739/A2KH0F08V (Jozef et al., 2021) as described in
de Boer et al. (2022). Meteorological tower data are available at the National Science Foundation Arctic Data Center at https://doi.org/10.18739/A2PV6B83F (Cox et al., 2023) as described in Cox et al. (submitted). Ceilometer and microwave radiometer data are available at the Department of Energy Atmospheric Radiation Measurement Data Center at http://dx.doi.org/10.5439/1181954 (ARM user facility, 2019a) and http://dx.doi.org/10.5439/1027369 (ARM user facility, 2019b) respectively, as described in Shupe et al. (2021).

**Author contributions**

SD provided the radiosonde data; CC provided the meteorological tower data; GdB and JC planned the DH2 data collection and acquired funding; GJ and JC conducted DH2 flights; GJ, JC, MD and GdB conceptualized the analysis presented in this paper; GJ analyzed the data; GJ wrote the manuscript; JC, MD, GdB, SD and CC reviewed and edited the manuscript.



**Competing interests**

The authors declare that they have no conflict of interest.

**Acknowledgments**

Data used in this paper were produced as part of RV *Polarstern* cruise AWI_PS122_00 and of the international
Multidisciplinary drifting Observatory for the Study of the Arctic Climate (MOSAiC) with the tag MOSAiC20192020.
We thank all those who contributed to MOSAiC and made this endeavor possible (Nixdorf et al., 2021). Radiosonde
data were obtained through a partnership between the leading Alfred Wegener Institute (AWI), the Atmospheric
Radiation Measurement (ARM) User Facility, a US Department of Energy (DOE) facility managed by the Biological
and Environmental Research Program, and the German Weather Service (DWD). Meteorological tower data were
obtained by the National Oceanographic and Atmospheric Administration (NOAA). Ceilometer and microwave
radiometer data were obtained by the AWI and DOE-ARM User Facility. We appreciate comments provided by an
anonymous internal reviewer at NOAA.

**Financial support**

Funding support for this analysis was provided by the National Science Foundation (award OPP 1805569, de Boer,
PI) and the National Aeronautics and Space Administration (award 80NSSC19M0194). The meteorological tower
observations were supported by the National Science Foundation OPP-1724551, by NOAA's Physical Sciences
Laboratory (PSL) (NOAA Cooperative Agreement NA22OAR4320151) and by NOAA's Global Ocean Monitoring
and Observing Program (GOMO)/Arctic Research Program (ARP) (FundRef https://doi.org/10.13039/100018302).
Additional funding and support were provided by the Department of Atmospheric and Oceanic Sciences at the
University of Colorado Boulder, the Cooperative Institute for Research in Environmental Sciences, the National
Oceanic and Atmospheric Administration Physical Sciences Laboratory, and the Alfred Wegener Institute.

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
