# Peer review of "An Overview of the Vertical Structure of the Atmospheric Boundary Layer in the Central Arctic during MOSAiC"

_EGUsphere, 2023_

## Referee Comment (RC2)

Review on

**An Overview of the Vertical Structure of the Atmospheric Boundary Layer in the Central Arctic during MOSAiC**

By Gina C. Jozef, John J. Cassano, Sandro Dahlke, Mckenzie Dice, Christopher J. Cox, and Gijs de Boer

This manuscript describes a statistical analysis for a year-round period of Arctic boundary layer observations based largely on radiosondes during the MOSAiC project accompanied by data observed by the DataHawk2 unmanned vehicle. As a central tool, a "self-organizing map" approach was applied to the data. There is no doubt that such a statistical analysis is extremely helpful in addition to all the case studies that have been and are being evaluated. Therefore, the enormous amount of work is greatly appreciated, and after major improvements to the manuscript, I also support the publication of this analysis. However, the manuscript needs a thorough revision that goes far beyond classical "major revisions". I will try to justify this in detail below.

**My main concerns are:**

The manuscript is extremely difficult to read and understand; this is largely due to the very intensive use of abbreviations. Especially when referring to the different stability regimes of the boundary layer, the abbreviations are not really intuitive and hardly anyone will be able to remember them while reading (see at line 428/29 for example when a sentence almost completely is based on abbreviations). In particular – see Tab 2- the capital "S" is sometimes used for "shallow" and sometimes for "stable" – I have no good suggestion at the moment to improve this, but please consider of a better and simpler way to categorize the different regimes and avoiding abbreviations.

Very often it is concluded in the paper that the analysis yields unsurprising results or that the results are logically and physically explainable - well, I expect that with the correlations but do you want to evaluate the tool or deliver new scientific findings?

It is a bit tiring for the reader to have each figure described in such detail (and the figures contain a large amount of detail…), and you should try to find a slightly better and more compact way of presenting and introducing the figures. I know this comment is quite generic but maybe you find a good way to describe your figures in a more comact way.

When I first read the manuscript and got to Figure 2, I was completely overwhelmed. Why do you need 30 schemes to describe the ABL? With many patterns you only see marginal differences when you look very closely. I think that the manuscript could be made much simpler and more readable if the analysis was limited to a handful of characteristic patterns.

A general note on writing style: please try to avoid repetition to strengthen the manuscript. Furthermore, the sentences are often so complicated and convoluted that a fluent reading - at least for me - was very difficult or impossible. I myself am not a native speaker, but there are enough competent co-authors who can edit the manuscript thoroughly.
General comment about most of the figures (although here I refer explicitly to Fig. 3):

The figure is based on 30 subplots which are by definition quite small but if you try to include even more information in terms of several additional numbers and vertical or horizontal lines, the plots will get really crowded. Even worth in Fig 4 where I am not able to read at all the numbers you included into the subplots – they are simply too small and too many.

I am not convinced that discussing material in the manuscript that has been moved to a supplement is the correct or formal way to do it. If you have to many figures, you should solve this problem differently.

About the analysis of temperature inversions: I am a little bit skeptical about this analysis and I wonder of how much of these results are based on self-correlation because the definition of the stability regimes is also based on temperature gradients. You mentioned this issue briefly but this needs to be discussed in more detail.

Maybe one solution for an improved structure of the entire manuscript would be a stricter separation between explaining the results in one section and discussion and interpretation in another one.

I am not convinced about the meaningful interpretation of parameters averaged over the entire MOSAiC cruise. For example, what can I learn from a statement such as *"The average ABL height during MOSAiC was 150 m, and ABL height increases with decreasing stability. "* (line 703)? You average ABL height over completely different ABL regimes and the second part of the sentence is somewhat trivial and expected – right?

**More specific comments:**

Introduction (line 35): what do you exactly mean with high temporal and spatial resolution – please specify.

Line 39ff: quite generic comment; please provide references - I think it is quite obvious that the Arctic ABL is not necessarily stably stratified in summer - right?

Line 49ff: I assume that - depending on the temperature stratification - the turbulence maybe also increased above the LLJ core because the shear could be similar - right? Furthermore, why should a LLJ weaken stability? The main preconditions for the existence of a LLJ is an almost vanishing turbulent transfer coefficient - typical for stable regimes.

Line 101: although you provide an explanation later on, I think a short introduction of what a
"self-organizing map" is should be included - simply because you mention it and I think a few words about this technique is essential already at this point and not all readers are knowing about this?!

Methods:

Line 116 ff: is this information about RV Polarstern movement is of interest for your work here – why do you mention this?

About Tab 1:

- What are the sources for the uncertainties? The Vaisala manual? Please provide a reference.

- Furthermore, I have serious concerns about the given uncertainty of wind observations: I know that this is the value given in the specifications by Vaisala but there is a lot of discussion about errors in determining the wind velocity - in particular at high latitudes where GPS comes to its end.

- Why do you mention uncertainties above 16 km here?

- I don't understand why a sonic is not enough to estimate the friction velocity?

Line 165ff: How can you expect a slope for higher altitudes just based on the lowermost 10 m? Why not simply compare the highest measurement point of the tower with the lowermost observation level of the radiosonde - I assume 12 m (helideck) and 10 m (top of the mast) should compare quite well? If not, you have a problem with the radiosonde – right? Or did I completely misunderstand your approach? – Simply double check the wording.

Line 168: interpolated => extrapolated? Please check.

Line 172: I thought a low-pass filter removed this pendulum motion? Please comment on this.

Line 185: I am somewhat surprised by the high critical Ri value which is two-times higher compared to the "classical" value. Most values published in literature are below 0.25 – do you have an explanation for this?

Line 186: I think you can shorten this part a little bit by citing your paper only one times

Line 189: just to understand it correctly: the gradient is a mean gradient from 35 m to ABL top - right?

Line 225: What do you exactly mean with "theta anomaly profile"? And in which way is one approach "better" than the other one? - please specify.

Line 230 ff: What details are "better" when using 30 patterns instead of 20 or 35? What did I learn from this detail?

Line 228ff: I understand that you want to explain details about SOM in a specific part of the paper but you mentioned SOM several time before your explanation - maybe you should at least mention at the beginning what SOM stands for and refer to this point here. I feel that many readers have never heard about SOM before and at least a brief introduction at the beginning could help – or did I have overseen this?!

Line 244ff: Maybe at this point a comment about the low-pas filtering of GRUAN data is useful and how it effects your data and evaluation?! Or why using 5 m as a grid spacing when the low-pass filtering is at 75 m or so? (see also the comment by Günther Heinemann)

About Fig 2.: Maybe I missed it but why do I need 30 patterns to describe typical ABL stratifications? For example, what is the difference between pattern 27 and 28? By eye there is no difference. A technical comment on Fig 2: the pattern number and the number of observations is in the same font and partly not well visible - maybe you could provide a color background for the two set of numbers?

Line 259ff: when reading this part, I immediately ask myself if the DH2 observations have a chance to cover all the different patterns because it didn't fly in the Polar night so it should miss the real stable conditions- right?

Line 267ff: Why is a SOM based on anomalies more visual? If you anticipate the result here, I immediately wonder why you used the gradient first and did not start with the analysis of the anomaly right away – I am confused here…

Line 279ff: I partly understand the motivation to define so many different stability regimes, but I fear that the usefulness for most readers is very limited. These 12 regimes are linked in the manuscript with 12 abbreviations that I definitely cannot remember and when these are mentioned and discussed in the text, I as a reader jump back and forth to remember the abbreviations. This disrupts the flow of reading, at least for now, whether I can do much with the information or not.

Line 285: A possible solution for a better reading flow could be to distinguish even more clearly between methods and observations - this is only one possibility but in some places these two aspects blur a bit.

line 286ff: Why Antarctica? I think MOSAiC should really be sufficient and citing a non-published paper from the other side of the world does not really help here…

line 290ff: Why is the gradient in 42.5 m representative for the AGL? I understand that this value might be representative for the surface layer (at least in summer) but the entire AGL - or do I misunderstand? Please clarify.

Line 325ff: Why are you defining possible regimes that were never observed in the data from MOSAiC? Maybe you have some good reasons but just reading this sentence confuses me.

Fig 3.: This figure (Fig. 3) contains a lot of information, and I suspect that most readers will have difficulty understanding all the lines and what they mean. Perhaps there is a way to make the diagrams a little clearer.

The colored frame lines describing the regimes should be somewhat thicker to better distinguish the different regimes

Also, at this point I wonder how the ABL height is defined! For example, in pattern 8 it is quite difficult to estimate an AGL height even by eye. I assume that the inversion and the entrainment layer are not part of the AGL height according to your definition, right? Is then the term "mixed layer" more appropriate compared to AGL height? You should at least define the phrases carefully at a prominent place.

Line 358ff: why "perhaps" - you should have the data to evaluate this "unique processes"!

Line 367: I am not convinced that the Arctic ABL is "always" stably stratified" - in particular in summer this is definitively not the case (see Tjernström et al.) So, I think to sell this as a "new finding" is going too far.

Fig 4: most of the numbers are more or less invisible, at least the numbers in the upper line. Furthermore, black labels on a dark background are quite challenging. I think you should find a much better way to illustrate your point here

Line 377: Again, I am not in favor of discussing material that is not in the manuscript but has been moved to a supplement. Regardless, I don't quite understand the following statement; DH2 couldn't fly in clouds - right? How then can DH2 observations cover all patterns when clouds affect stratification so much - or have I misunderstood something?

Line 383ff: Do I understand correctly that LLJs are to be analyzed based on mean values within one individual pattern? As you have already noticed, this must - at least partially - average out the LLJ's with having their wind maximum at different heights - right? Then I have serious doubts whether this is the right method and whether one can make statements about LLJs in this way. However, it might be a misunderstanding on my side but then please clarify and consider rephrasing. In addition, there are from my point of view quite valid remarks of Günther Heinemann about the quality and (in)accuracy of wind measurements in the low altitude and the low-pass filtering of the wind data. Please take these comments seriously.

Line 388: Physically, I don't understand this point: a LLJ is by definition linked to a more stable ABL because one precondition for a LLJ is the almost vanishing turbulent exchange coefficient - right? So how can you say that "Thus the LLJ is more closely coupled to the ABL in the weak stability cases…" Maybe I don't understand your point here but then you should clarify it.

Fig. 5: Again, too much information and details in the plots.

Line 415 ff: The is probably one of the most prominent places to say: I am lost in all the details and even more I lost track due to the huge number of abbreviations….

line 454: If a parameter x changes over the depth of the ABL it is not the same as the gradient dx/dz – right?

Line 455ff: If you cite a figure within the main text, it should be included in the main text - why has it been shifted to a supplement?

line 463: "...the fact that we see this drastic increase also supports the choice of this threshold...." can you please explain this in a little bit more detail?

line 465ff: I cannot follow this sentence at all - please double check and consider rephrasing.

line 468ff: It is quite unusual to start a sentence with a equation - I would avoid it. Furthermore: "shear" => "wind shear".

So why do you mention Ri with index "b" (for bulk?!) here? If you mention the (local?) gradients then you have the basics for the classical local Ri definition - right?

line 475ff: again, a reference outside the paper is not helpful and I suggest to avoiding this. Furthermore, I do not understand your conclusion about the physically meaningful definition of the regimes - please specify what you mean here.

line 477: what is exactly meant by "dV/dz result" - from my point of view this makes no sense

line 479ff: It is surprising that you start the discussion with the Richardson number and now you move on to the friction velocity - so why? Furthermore, you correctly mentioned that Ri describes more the tendency of turbulence development but it is not a measure of the degree of turbulence or the intensity but you concluded already that the near-surface atmosphere is always turbulent. From my point of view this is going too far. This part also needs some careful reconsiderations and not only rephrasing.

line 487: Well, turbulence itself might describe a flow but probably not the Arctic - this makes no sense. Maybe I missed it but I suggest to define u_star at the place of first occurrence (around line 145 or so). Also, I suggest to use the word „increased" turbulence.

488ff: this not really surprising and I think this statement does not need a supplementary figure - right?

492ff: I am not surprised that u_star and and wind shear do not show a clear dependency because it is a Richardson number problem - but is this a conclusion as you mentioned?

line 495ff: I cannot follow your argumentation, in particular the last part "...when stability aloft is greater." So, is there a connection between stability at higher altitudes and surface layer mixing?  Please explain what you mean here.

Fig 7 (and maybe other figures as well): you put the labels of the y-axis on the x-axis which is formally not correct and took me a time to understand the plot - please change this labeling.

line 511ff: Maybe I am wrong and I don't have the details of Banta et al in mind but how can a LLJ exists (or develop) in a well-mixed ABL? Maybe during the transition to a classical Ekman-layer like ABL this makes sense but from a theoretical point of view a LLJ and a well-mixed layer are exclusive; the turbulent exchange coefficient $K_m$ has tend to zero to decouple the LLJ region from the surface - right? In fact, that is the background for your comment in line 513ff but I think this is a precondition for the development of a LLJ....

line 515ff: maybe you mentioned it earlier but is the "LLJ speed" defined as the maximum speed in the LLJ core or the difference to the surrounding?

line 548ff: I think this is physically not really meaningful, maybe it should read as: "...it needs more wind shear in a more stable environment to create mixing ..." or similar. Furthermore, I think the phrase "hypotheses" is going a little bit too far because you never formulated a hypothesis which can now be verified or falsified...

line 555: what do you exactly mean with "excess turbulence"?

Fig 8.: See my comment on Fig 7 about the axis labels

Line 567ff: About your supplement: it just includes many further figures which are partly mentioned within the main manuscript but not explained and deeply discussed. I think this is not the right way to use a supplement because the main manuscript should be readable by itself without reading the supplement. If you have a distinct and interesting topic which might be useful for some readers but distract from the red line of the manuscript a supplement might be the right choice but if you have simply too many figures which you want to mention in the manuscript you cannot just move them into the supplement and refer to them (different to an appendix).

line 577ff: Do you consider TIs as a cause or effect of ABL development?

Line 578ff: About the TI analysis: I am a little bit skeptical about this analysis because I wonder how much of this analysis is based at least partly on self-correlation because the stability regimes are based on the temperature gradients - right? Maybe this should be discussed at least a little bit before interpreting the results.

Line 581 and general: Maybe better distinguish between explaining figures and results, than interpreting them and finally compare with other studies - the structure is often a bit confusing and you jump back and force

line 591ff: I cannot follow your argumentation about the potential for exchange of momentum when TI is well above ABL height - what do you mean? I have the feeling that here is a lot of speculation on play but a careful physical interpretation is missing.

line 603: As mentioned earlier, I have the feeling that this part is based on a big portion of self-correlation which has to be ruled out before the interpretation.

line 621: why using a decimal value for the second cloud base height? Why 6.1 km and not 6 km??

line 633ff: What do you mean with the statement that "… a regime is driven by the radiative signature of clouds..."? and further on in line 634 what are those other mechanisms? This explanation is hardly to follow and needs some careful rephrasing.

The summary will certainly need to be completely revised when the previous analyses have been appropriately re-sorted and revised - so I have refrained from detailed comments on this chapter now.

---

## Author Comment (AC1)

**Response to Community Comments 1**

Thank you for your comment on our manuscript. Below we address each of your concerns and explain any resulting changes to the manuscript.

***The fraction of LLJ profiles of 76% is very high and exceeds the values found by previous studies for sea ice in polar regions.***

There are in fact several other studies which find a similar LLJ frequency in polar sea ice regions. For example, Andreas et al. (2000) found 80% of soundings launched over the sea ice in the western Weddell Sea during fall and winter of 1992 to contain an LLJ. Note that the Andreas et al. (2000) paper defined LLJs the same way we do, without any criteria specifying that the jet core be 25% faster than the wind speed minimum above. Thus, our results are quite consistent with this previous study. Even though the Andreas et al. (2000) paper reports on LLJs over the southern ocean, the physical processes should be similar to those which occur over the Arctic sea ice and thus similar results would be expected.

An additional study by Tian et al. (2020) conducted using sounding data in the central Arctic between 31 July and 4 September 2018 found an LLJ frequency similar to that presented in the current manuscript. Again this study did not use the criteria specifying that the jet core be 25% faster than the wind speed minimum above. Tian et al. (2020) found that overall 65.48% of observations contained an LLJ, where LLJs occurred 56.62% of the time over open water regions, 65.52% of the time in the over the marginal ice zone, and 71.88% of the time over the pack ice. As much of the MOSAiC observations were over the pack ice, our finding of an LLJ in 76% of soundings is quite similar to the results of Tian at al. (2020).

Lastly, ReVelle and Nilsson (2008) report that 2/3 of soundings collected north of 80 °N in 1996 as part of the Arctic Ocean Expedition (AOE-96) contained an LLJ. They also state: "Cumulative estimates from the work of Andreas and colleagues and from those of Nilsson and colleagues suggest that LLJs occur during about 60%–80% of all of the soundings taken during these various polar oceanic field expeditions" (ReVelle and Nilsson, 2008). Thus, the authors of the current manuscript disagree with your assessment that an LLJ frequency of 76% in the central Arctic is "very high and exceeds the values found by previous studies for sea ice in polar regions." We have added a brief summary of the aforementioned discussion and citations to the manuscript to strengthen the validity of our results. We have also added explanations for why our LLJ frequency exceeds that of some other studies (see response to your next two comments).

***Lopez-Garcia et al. (2022) found about 50% of the cases using the same radiosonde data set, but only 6-hourly ascents.***

The authors understand your concern and confusion that our results find a higher frequency of LLJs than is found in the study by Lopez-Garcia et al. (2022) study which uses the same MOSAiC radiosonde dataset. However, we already mention in the manuscript (L196-200 in original manuscript) the primary reason for the discrepancy: Lopez-Garcia et al. (2022) only considers LLJs in which the jet core speed is at least 25% greater than the wind speed minimum above, while we do not include this criteria and thus we detect more LLJs. Not implementing the 25% criterion

was a conscious decision made by the authors, because this study focuses on the impacts of atmospheric forcings on boundary layer stability, and an LLJ which does or doesn't meet the 25% criteria still affects the near-surface layer below the jet core.

***Tuononen et al. (2015) found about 20% as a model-based climatology for the inner Arctic.***

The vertical resolution of the ASR-Interim data used in Tuononen et al. (2015) is 25 hPa, which equates to a resolution of $\sim 165 - 200$ m in Arctic temperatures. The radiosonde data have a resolution of $\sim$5 m. With such a discrepancy in vertical resolution, we would not expect to find the same results when it comes to identifying LLJs. To exemplify this, there are $\sim$150 cases in the MOSAiC radiosonde data which have an LLJ depth of less than 200 m, and if we remove these cases from consideration, this already decreases the LLJ frequency to 65%. Thus, results from observational data and from lower resolution data cannot be expected to give the same results.

***The method of LLJ detection needs more explanation. How do you treat multiple maxima? Do you just search for the next minimum about the LLJ height or any minimum below 1500m? Do you apply a low-pass filter on the radiosonde data to remove turbulent bursts (which was the motivation of Tuononen et al. (2015) to use a 25% criterion)? Have you made any consistency checks, if you have jumps in LLJ height between consecutive profiles? This should be tested for periods with 3-hourly radiosonde profiles.***

We have added the following information to the manuscript: When there were multiple maxima, we only considered the lowest one, and a maximum was only considered an LLJ when it was at least 2 m s$^{-1}$ greater than the next local minimum above the LLJ or the value at 1.5 km (if no local minimum above the maximum), as in Tuononen et al. (2015). We additionally mention (as was included in the first draft of the manuscript on L193) that further details on LLJ detection, including example figures, can be found in Jozef et al., 2023 (now in preprint).

We do not apply any low-pass filter on the radiosonde data to remove turbulent bursts because there is already quite some vertical filtering/smoothing applied in the level 2 sounding (used in this study) when Vaisala processes them. In fact, that the wind profiles in the level 2 product are generally less "wiggly" than in the level 3 data, suggesting that the Vaisala filtering in level 2 is even more rigorous than the filtering in the level 3 product; for reference, in the level 3 product, they apply a Gaussian kernel of 35 - 75m vertically, so the smoothing in the level 2 is likely over an even greater bin. Therefore, we can reasonably assume that any fluctuations in wind speed from turbulence bursts have already been removed, and this is not of concern.

However, to confirm this we looked at changes in LLJ height between consecutive cases when the observations occurred less than 4 hours apart, both with and without the 25% criterion. With and without the 25% criterion applied, there are jumps in LLJ altitude between consecutive cases having similar magnitude. Any large jumps can be tied to large changes in ABL height, observed at the same time. Applying the 25% criterion also removes some cases with small changes in LLJ height between consecutive cases, suggesting that this criterion is removing real LLJs. Thus, we do not find a benefit (and in some cases find a detriment) to accurately identifying LLJs in applying the 25% criterion.

*It should be proved that turbulent bursts do not influence the results. The evaluation should be repeated using the 25% criterion and/or a filtering. The differences particularly to the results of Tuononen et al. (2015) should be discussed.*

We have reproduced Supplementary Fig. S4 and Fig. 8 now using the 25% criterion (below). The annual frequency when the 25% criterion is used is in closer agreement with Lopez-Garcia et al. (2022). We also show that the trends presented in the box and whisker plot when the 25% criterion is applied do not differ from the results when the 25% criterion is not applied. The primary difference is that the mean and median LLJ speeds are slightly lower when the 25% criterion is applied (which is expected since this criterion eliminates cases of ubiquitously high wind speed LLJ events).

To address your concerns, we have added some discussion to the paper summarizing the points of this document.

[Figure]

**As in Supplementary Fig. S4, but when the 25% criterion is applied.**

[Figure]

**As in Fig. 8, but when the 25% criterion is applied.**

**References**

Andreas, E. L., Claffy, K. J., and Makshtas, A. P.: Low-Level Atmospheric Jets And Inversions Over The Western Weddell Sea, Boundary-Layer Meteorology, 97, 459-486, https://doi.org/10.1023/A:1002793831076, 2000.

Jozef, G. C., Klingel, R., Cassano, J. J., Maronga, B., de Boer, G., Dahlke, S., and Cox, C. J.: Derivation and compilation of lower atmospheric properties relating to temperature, wind, stability, moisture, and surface radiation budget over the central Arctic sea ice during MOSAiC, Earth Syst. Sci. Data Discuss. [preprint], https://doi.org/10.5194/essd-2023-141, in review, 2023.

Tian, Z., Zhang, D., Song, X., Zhao, F., Li, Z., and Zhang, L.: Characteristics of the atmospheric vertical structure with different sea ice covers over the Pacific sector of the Arctic Ocean in summer, Atmospheric Research, 245, 105074, https://doi.org/10.1016/j.atmosres.2020.105074, 2020.

ReVelle, D. O. and Nilsson, E. D.: Summertime Low-Level Jets over the High-Latitude Arctic Ocean, Journal of Applied Meteorology and Climatology, 47, 1770-1784, https://doi.org/10.1175/2007JAMC1637.1, 2008.

Lopez-Garcia, V., Neely III, R. R., Dahlke, S., and Brooks, I. M.: Low-level jets over the Arctic Ocean during MOSAiC, Elementa: Science of the Anthropocene, 10, 00063, https://doi.org/10.1525/elementa.2022.00063, 2022.

Tuononen, M., Sincalair, V. A., and Vihma, T.: A climatology of low-level jets in the mid-latitudes and polar regions of the Northern Hemisphere, Q. J. Roy. Meteor. Soc., 16, 492-499, https://doi.org/10.1002/asl.587, 2015.

---

## Author Comment (AC2)

**Response to Community Comments 2**

Thank you for your comment on our manuscript. The authors have some skepticism regarding your concerns, as you refer to published uncertainties in the level 3 data, but the level 2 data are used in the current study and thus the level 3 uncertainties may not be applicable here. For example, you state that the error in the level 3 maxima is 5-7 m/s, but the published uncertainty in the Vaisala wind speeds is only 0.15 m/s. The authors do acknowledge that the "real" uncertainty may be higher but likely are not on the order of 5-7 m/s. A comparison of radiosonde winds at 10 m to the met tower winds at 10 m from MOSAiC measurements during October – April shows an RMSE of 1.17 m/s. Additionally, the RMSE between the ship's wind measurement (29 m) to the radiosonde wind speed at 30 m is 1.27 m/s. A comparison of the DataHawk2 UAS and radiosonde measurements in Hamilton et al. (2023) reveals a difference of <1 m/s based on the 95% confidence intervals of observations from both platforms. To ultimately determine if wind uncertainties may affect the results in the current analysis, we have repeated the analysis per your suggestions (below). In the end we have determined that the original methods (using a threshold of 2 m/s and no vertical averaging) give the most accurate climatology (see Sect. 1 of response) and are relatively unaffected by the errors you refer to (see Sect 2 of response).

Hamilton, J., de Boer, G., Doddi, A., and Lawrence, D. A.: The DataHawk2 uncrewed aircraft
    system for atmospheric research, Atmos. Meas. Tech., 15, 6789–6806,
    https://doi.org/10.5194/amt-15-6789-2022, 2022.

**Section 1:** Changing the criterion for identifying a wind speed maximum as an LLJ.

First, we repeated the routines regarding LLJs and have reproduced the corresponding figures, now using a threshold of 4 m/s instead of 2 m/s. The figures are shown below. The authors found that changing the threshold this drastically yields an entirely different LLJ climatology that is less likely to be true (see reasons below). Given these results, the authors disagree that applying a 4 m/s threshold is an appropriate approach.

1. Using this threshold of 4 m/s (and not using the 25% criterion) now gives an annual LLJ frequency of 36.5%. This is now a much lower frequency than was found in by Lopez-Garcia et al. (2022). The discrepancy between our LLJ frequency and that of by Lopez-Garcia et al. (2022) in the submitted draft of this manuscript can be attributed to the fact that we did not use the 25% criterion, whereas they did. Otherwise, all other methods are consistent between the two papers, which was done intentionally through communications between the authors of the current paper and the authors of by Lopez-Garcia et al. (2022). Depending on a reader's purpose in wanting to know about LLJ frequency and characteristics, they can then draw from either the results of the current paper or those of by Lopez-Garcia et al. (2022). However, if we change our methods further by now applying a 4 m/s threshold instead of 2 m/s, this is in direct conflict with Lopez-Garcia et al. (2022) which, as both studies use the same dataset, would be even more confusing to a reader.
2. The only LLJ threshold that appears in previous literature, as far as the authors have found, is the threshold of 2 m/s. In many other studies, this threshold has been applied to radiosonde datasets which likely have uncertainties comparable to those of the level 2 data used in the current study. The authors do not think it makes sense to challenge those methods used by so

many prior studies, as this would make the results of the current study incomparable with those of prior studies.

3. An LLJ frequency of 36.5% is now very low compared to that found in previous work in polar sea ice regions. In our response to your previous comment, we shared several citations which reveal an LLJ frequency in polar sea ice regions closer to the 76% shared in the submitted manuscript for the current study. Thus, a threshold of 4 m/s produces results inconsistent with previous work, so we can be confident that it is likely missing many true LLJ events.

4. One of the primary purposes of the current study is to relate LLJ characteristics to stability regime. The trends presented in the box and whisker plot when a threshold of 4 m/s is used do not differ from the results when a threshold of 2 m/s is used. The primary difference is that the mean and median LLJ speeds are slightly higher when a threshold of 4 m/s is used. This suggests that using a threshold of 4 m/s misses a lot of LLJ events which have slower wind speeds, but are still important in their interaction with near-surface stability (as is the focus of the current study).

[Figure]

**As in Supplementary Fig. S4, but when a threshold of 4 m/s is used.**

[Figure]

**As in Fig. 8, but when a threshold of 4 m/s is used.**

**Section 2:** Applying vertical averaging.

We have also repeated the routines regarding LLJs and have reproduced the corresponding figures, using a running mean of 30 m and 60 m, which should reduce the random error by 60% and 71% respectively (see the math below). Since the claimed error in the wind speed maximum could be as high as 5-7 m/s, a reduction of 60% of the low end of this range (5 m/s) and a reduction of 71% of the high end of this range (7 m/s) would each be 2 m/s, thus bringing the threshold of 2 m/s for a wind speed maximum to be identified as an LLJ within the claimed error.

The error for one data point is:
$1/(\sqrt{1})*ERROR =1*ERROR$

Averaging over 6 data points (i.e. 30 meters vertically with 5 m resolution data), the error for one data point would then be:
$1/(\sqrt{6})*ERROR=0.4*ERROR$, thus reducing the error by 60%
$0.4 * 5$ m/s $= 2$ m/s

Averaging over 12 data points (i.e. 60 meters vertically with 5 m resolution data), the error for one data point would then be:
$1/(\sqrt{12})*ERROR=0.29*ERROR$. thus reducing the error by 71%
$0.29 * 7$ m/s $= 2$ m/s

Corresponding figures are shown below. When a 30 m running mean is used, the annual frequency of LLJs is reduced by less than 3%, to 73.4%. When significance testing is conducted, it is found that this frequency is not significantly different at the 95% confidence level from the frequency when no running mean is used. We have added a sentence to the manuscript addressing the aforementioned result. When a 60 m running mean is used, the annual frequency of LLJs is reduced by about 5%, to 71%. This time when significance testing is conducted, it is found that this frequency is significantly different at the 95% confidence level from the frequency when no running mean is used. However, an actual uncertainty of 7 m/s for the maximum in the wind speed from the level 2 data is unlikely to be true, especially given the published uncertainty in the Vaisala wind speeds of 0.15 m/s. Regardless, a reduction of LLJ frequency from 76% to 71% does not change the story that LLJs are commonly occurring in the central Arctic. Additionally, once again applying a running mean does not change the relationships between LLJ characteristics and stability regimes shown in the submitted manuscript, which is the focus of the current study.

[Figure]

**As in Supplementary Fig. S4, but when a 30 m running mean is used.**

[Figure]

**As in Fig. 8, but when a 30 m running mean is used.**

[Figure]

**As in Supplementary Fig. S4, but when a 60 m running mean is used.**

[Figure]

**As in Fig. 8, but when a 60 m running mean is used.**

---

## Author Comment (AC3)

**Response to Anonymous Referee 1 Comments**

We would like to sincerely thank Anonymous Referee 1 for taking the time to review our manuscript and for their helpful comments, which have improved the manuscript. Each referee comment is given below in ***bold italics*** followed by our response to the comment. The line numbers provided in our responses refer to line numbers in the revised manuscript, unless otherwise stated.

***This paper investigates in some detail the vertical structure of the lower atmosphere in the Arctic using MOASiC data. While the attempt as such is commendable and the methods are interesting, the paper does not harvest. Nothing new or unexpected comes to light; instead most of the conclusions either follows logically at a textbook level from cited criteria, or worse, in some cases are side effects of the sampling. The authors seem unaware of a lot of material already published on this topic, but are hung up on a few widely spread misconceptions that they belive they have proven but not, and also seem to lack insight into turbulent flows and boundary-layer dynamics.***

***These concerns are grave enough that I must recommend rejection at this point, while hoping the authors come back and do a better job of this topic because it is important.***

Thank you for your review of our paper. We appreciate the reviewer's suggestions for better placing this work in the context of previously published studies on the Arctic boundary layer. In response to this general concern we have thoroughly revised the introduction with a more comprehensive review of previously published studies relevant to our work. While we recognize that many of the results presented in the manuscript follow textbook ABL meteorology, we argue that this does not mean the findings are unimportant. MOSAiC was a unique field campaign, only the second of which (besides SHEBA) to cover an entire year in the central Arctic. Due to the difference in location from SHEBA, the widespread changes in the Arctic climate system since SHEBA, and the higher temporal resolution of radiosoundings, the MOSAiC data serves to provide additional insight into Arctic lower atmospheric dynamics beyond what has already been discovered – if this confirms what we already expect to be true, well that is still important to know! Additionally, this paper does expand upon previous studies by taking a unique approach (e.g., through the SOM analysis and detailed stability regime classification) to reveal precise quantities of important atmospheric characteristics from a new perspective. However, we agree with the reviewer that we have not properly recognized the previous literature on the subject matter, have largely failed to communicate the value of confirming textbook ABL concepts with a new dataset and new methods and have not done a good job at highlighting any new discoveries which have come from this study. Thus, we have made substantial changes to the manuscript with the hopes that we have sufficiently addressed your major concerns, such that you may reconsider your position on the status of this paper.

Below we address each of your comments, and explain how and where changes have been made to the manuscript.

*Major concerns:*

***Off the bat, the authors erroneously claim that the Arctic ABL is almost always stably stratified and then does whatever they can to make that the truth, although their own results actually shows this to (still!) **not** be the case. The most common stability in the Arctic ABL is in fact near neutral; after all, what is the stability difference near the surface between "near-neutral with stability aloft" and "shallow and well-mixed"? It is not the stability but the depth!***

We are troubled to see that one of the primary arguments of our paper was so misunderstood. Clearly, we did not do a good job at communicating our intentions in several spots throughout the manuscript. First, the purpose of mentioning that previous studies "have concluded that the Arctic ABL is typically stably stratified" (line 39 in original manuscript) was to bring up a conclusion which we would later disprove through the results of this study. However, we realize that previous studies have already disproved this notion, and so we have revised the introduction to accurately represent what is currently known:

"Previous studies have revealed that the Arctic atmosphere over sea ice is typically either stable or near-neutral (Tjernström and Graversen, 2009; Persson et al., 2002; Esau and Sorokina, 2010), while instability is rare or confined to the lowest few meters (Brooks et al. 2017; Tjernström et al., 2004; Persson et al., 2002)." (line 39)

We have also adjusted our wording throughout the text to better communicate that our findings reveal frequent near-neutral conditions in the Arctic. For example, in Fig. 6 of the original manuscript (Fig. 5 of the revised manuscript), we show that the most common stability regime annually is near neutral with strong stability aloft (NN-SSA). Considering all cases with near-surface regime of near-neutral (NN) regardless of aloft stability, this accounts for 37% of cases, and considering all cases with near-surface stability of strongly, moderately, or weakly stable (SS, MS, and WS) regardless of aloft stability, this accounts for 36% of cases (where the remaining cases are VSM, which may be either weakly stable or near-neutral). So we agree with you that the most common stability in the Arctic is near-neutral. In the revised manuscript, we have adjusted our discussion to include the above percentages and an explicit statement that near-neutral conditions occur with similar frequency (but overall slightly higher frequency) as stable conditions:

"The most frequent near-surface regime observed was NN (37% of profiles), followed by VSM (27% of profiles), MS (14% of profiles), and SS (13% of profiles) in decreasing order. WS was observed least frequently (9% of profiles). The total frequency of a stable ABL (combining SS, MS, and WS frequencies) was 36%, just slightly less than the frequency of a near-neutral ABL." (line 457)

"The SOM patterns (Fig. 2), frequency distribution of stability (Fig. 5a), and ABL height variability (Fig. 5b) highlight that near-surface stability during MOSAiC spanned from strongly stable with a shallow ABL to near-neutral with a deep ABL, with stable and near-neutral conditions occurring with similar frequencies. Stability aloft ranged from strongly to weakly stable." (line 568)

We wonder if this confusion comes from our statement that "the high frequency of regimes with either moderate or strong stability near the surface, or a well-mixed ABL with strong stability aloft, suggests that the central Arctic lower atmosphere trends towards being strongly stable, but sometimes the near-surface atmosphere can become well-mixed due to the generation of turbulence" (line 417-420 in original manuscript). What we were trying to say with this statement is simply that a $\theta_v$ inversion layer falling into the strong or moderately stable category is common somewhere in the lowest 1 km. Sometimes this $\theta_v$ inversion occurs near the surface. Sometimes this $\theta_v$ inversion is elevated, capping a weakly stable or near-natural layer below, when turbulence has been generated. Whether this moderate or stronger stability is present at the surface or just above the surface is still relevant as it limits the mixing between the surface and the deeper troposphere and is a relevant feature of the Arctic atmosphere. In the revised manuscript we have adjusted the discussion to clarify this point, and now also note that instances of an elevated $\theta_v$ inversion occurs with higher frequency:

"The most frequent stability regimes were those with strong or moderate stability either near the surface (SS and MS) or aloft (VSM-SSA, VSM-MSA, NN-SSA, and NN-MSA). Thus, we conclude that the central Arctic atmosphere over sea ice is inclined to include a stable layer somewhere below 1 km AGL; sometimes this stable layer is within the ABL and sometimes it caps a well-mixed ABL, with the latter scenario occurring with higher frequency, consistent with Tjernström and Graversen (2009)." (line 581)

Lastly, we bring attention to the following line in the abstract of the original manuscript: "The patterns identified by the SOM allowed for the derivation of criteria to categorize stability within and just above the ABL, which reveals that the Arctic ABL is stable and near-neutral with similar frequencies." (line 19-20 in original manuscript) Thus, we did clearly note this conclusion in the beginning of the document.

***These authors need to read up more literature on the vertical structure of the Arctic ABL, perhaps beginning with the Tjernstrom and Graversen paper (2009 in QJRMS), performing a similar analysis on the SHEBA data, however, without the benefit of SOM. Yes, SHEBA was a while ago, but all things old are not useless.***

Thank you for the recommendation of the Tjernstrom and Graversen (2009) paper. Indeed this paper conducted a similar analysis as the one we present, using SHEBA data, and is a very useful paper to give background context for this study. Several authors were aware of this publication, and not including it in the introduction was an oversight on the part of the authorship team. Clearly, we recognize that it is important that we properly recognize previous findings on Arctic lower atmospheric vertical structure, and also that including this reference allows us to better place the current MOSAiC findings into broader context. We have dramatically changed the introduction for this

paper, including a much more thorough discussion on previous discoveries about the Arctic ABL vertical structure, some of which is pasted below:

"Previous studies have revealed that the Arctic atmosphere over sea ice is typically either stable or near-neutral (Tjernström and Graversen, 2009; Persson et al., 2002; Esau and Sorokina, 2010), while instability is rare or confined to the lowest few meters (Brooks et al. 2017; Tjernström et al., 2004; Persson et al., 2002). In the case of a near-neutral ABL, there is almost always an elevated capping inversion, typically with base height around 200-300 m, extending up to 1-2 km (Tjernström and Graversen, 2009). Surface-based and low-level inversions have been shown to contribute to Arctic amplification (Serreze and Francis, 2006; Serreze and Barry, 2011; Bintanja et al., 2011; Lesins et al., 2012; Gilson et al., 2018; Previdi et al., 2021) by dynamically decoupling the surface from the free atmosphere, so that surface heat flux perturbations cannot easily spread through the troposphere, and warming is concentrated near the surface (Lesins et al., 2012). These inversions also impact Arctic aerosol characteristics including the destruction of boundary layer ozone at the onset of polar sunrise and the transport of Arctic haze (Kahl, 1990), and contribute to the formation of fog during Arctic summer (Gilson et al., 2018).

Stable conditions are common in Arctic winter (Tjernström and Graversen, 2009) due to persistent longwave cooling in the absence of solar radiation (Brooks et al., 2017) and extended periods of clear skies or thin high clouds (Tjernström and Graversen, 2009), attributable to the lack of open water evaporation. However, intermittent instances of low stratocumulus clouds in winter can force a shallow well-mixed ABL (Morrison et al., 2012; Tjernström and Graversen, 2009; Persson et al., 2002). Such clouds are common during stormy conditions (Brooks et al., 2017; Persson et al., 2002).

Near-neutral or weakly stable conditions are common in Arctic summer (Brooks et al., 2017; Tjernström and Graversen, 2009), often capped by persistent stratiform clouds (Intrieri et al., 2002a; Tjernstrom, 2007; Curry and Ebert, 1992; Liu and Key, 2016; Shupe et al., 2011; Tjernström, 2005, Tjernström et al., 2012; Wang and Key, 2004; Zygmuntowska et al., 2012), which form as ample moisture is advected north either into the Arctic or from the broader ice-free areas across the pan-Arctic region, during the melt season (Sotiropoulou et al., 2016; Tjernström et al., 2019). The ABL is typically decoupled from the cloud layer by a shallow stable layer, such that turbulence is not exchanged between the cloud and the surface (Curry, 1986; Sedlar and Shupe, 2014; Sedlar et al., 2012; Shupe et al., 2013; Sotiropoulou et al., 2014). However, the common advection of warm moist air into the central Arctic can also result in the formation of a shallow, stable ABL (Tjernström et al., 2019; Tjernström, 2005; Cheng-Ying et al., 2011), especially towards the beginning of an advection event, or close to the ice edge (Sotiropoulou et al., 2016; Tjernström et al., 2019). Ice and snow melt in summer may also contribute to the formation of a stable ABL (Kahl, 1990; Gilson et al., 2018)." (line 39-67)

We have also added notes throughout the discussion section when our results agree with or expand upon these previous discoveries.

"The SOM patterns (Fig. 2), frequency distribution of stability (Fig. 5a), and ABL height variability (Fig. 5b) highlight that near-surface stability during MOSAiC spanned from strongly stable with a shallow ABL to near-neutral with a deep ABL, with stable and near-neutral conditions occurring with similar frequencies. Stability aloft ranged from strongly to weakly stable. These findings are consistent with Persson et al. (2002), Tjernström and Graversen (2009), and Brooks et al. (2017)" (line 568)

"…results of the current study agree with previous findings that clouds observed in the Arctic are typically low-level clouds." (line 643)

*In the end, the most common of the 12 stability classes used in this paper comes out to be NN in all seasons except summer (Figure 6) and adding in the well-mixed shallow cases, that all has a median Rib close to zero (the very definition of near neutral), it is clear that the initial statement and the conclusions by these authors in the text is just plain wrong (cf. e.g. lines 39, 363 and 693 in the manuscript)!*

Line 39 of original manuscript: We understand that the statement on line 39 of the original manuscript is not true. This statement was included to demonstrate previous misconceptions about the Arctic boundary layer and is not one of the conclusions that comes from our analysis. We had intentionally included it so that we could show how the findings of this study disproved the statement that the Arctic ABL is almost always stable in Esau and Sorokina (2010). However, after further literature review per your recommendation, we realize that many previous studies have also shown the high frequency of a near-neutral ABL in the Arctic, and so we have adjusted the discussion of previous literature in the introduction to include these studies and their findings (see response to above comment, and line 39-67)

Line 363 of original manuscript: The text beginning on this line in the original manuscript states that the ABL in the Arctic is *often* stable. We do not claim that the ABL is always or even usually stable. Simply stating that the ABL is often stable is consistent with what our, and previous, results show. Furthermore, we follow this by stating in the next sentence that many of the SOM patterns depict a near-neutral (NN) near-surface stability, which shows that a near-neutral ABL is also very common, which disagrees with the findings of Esau and Sorokina (2010) shared in the introduction. Thus, with these discussion points, we were intending to communicate the large prevalence of a near-neutral ABL in the Arctic.

Line 693 of original manuscript: Here we stated that NN-SSA is the most frequent stability regime, and regimes with weak stability aloft (VSM-WSA, WS, and NN-WSA) were least frequent. We also stated that a near-neutral atmosphere in the Arctic is prominent. We believe that these conclusions are consistent with your statements about the prevalence of near neutral conditions but have revised the text to more clearly articulate this point.

In the revised manuscript we have adjusted the discussion to clarify that stable and near-neutral conditions occur with similar frequencies, and do not claim that this is a new finding, but rather agrees with previous results, such as those of Tjernstrom and Graversen (2009):

"The annual distribution of SOM pattern frequency is displayed in Fig. 3a. The SOM pattern with the highest frequency (pattern 15, NN-WSA) accounts for 9.4% of MOSAiC observations. The pattern with the lowest frequency (pattern 11, SS) accounts for 1.1% of MOSAiC observations. The most common SS, MS, VSM, WS, and NN patterns were 5, 12, 29, 13, and 15 respectively. There are nine SOM patterns depicting strong or moderate near-surface stability. Seven patterns are very shallow mixed. Four patterns have weak near-surface stability. Ten patterns depict near-neutral near-surface stability." (line 391)

"The most frequent near-surface regime observed was NN (37% of profiles), followed by VSM (27% of profiles), MS (14% of profiles), and SS (13% of profiles) in decreasing order. WS was observed least frequently (9% of profiles). The total frequency of a stable ABL (combining SS, MS, and WS frequencies) was 36%, just slightly less than the frequency of a near-neutral ABL." (line 457)

"The SOM patterns (Fig. 2), frequency distribution of stability (Fig. 5a), and ABL height variability (Fig. 5b) highlight that near-surface stability during MOSAiC spanned from strongly stable with a shallow ABL to near-neutral with a deep ABL, with stable and near-neutral conditions occurring with similar frequencies. Stability aloft ranged from strongly to weakly stable. These findings are consistent with Persson et al. (2002), Tjernström and Graversen (2009), and Brooks et al. (2017)." (line 568)

Overall, we agree with the reviewer that the central Arctic ABL is often near-neutral, and did not intend to communicate otherwise in the manuscript. We hope the revisions make this clear.

***The SOM approach is interesting but poorly both explained and executed. Why so very many nodes? Is there no way to objectively determine the optimal number of nodes, e.g. by minimizing the inter-node and maximizing the intra-node variances?***

The purpose of a SOM is not to find the smallest number of patterns which represent all the possible structures in the data. Rather, a SOM is meant to continuously depict the range of structures present in the training data from one SOM-identified profile to the next - in this case, we show the continuum of ABL vertical structure in the Arctic.

Here, the SOM is being used as a way to visualize a large dataset (1377 soundings) in a manageable way while also allowing subtle details in the vertical structure (such as varying heights and differing strength of stable layers) to be identified. For example, at first glance, patterns 27 and 28 in Fig. 3 of the original manuscript (Fig. 2 in the revised manuscript) appear to be very similar. However, with a closer look, we see that the ABL depth between the two patterns differs discernably (leading one to be classified as WS, and the other as VSM), and the strength of the elevated inversion is a bit stronger for pattern 27 (even though they both qualify as -SSA). Then when looking at differences in wind speed and LLJ characteristics for these two patterns in Fig. 5 of the original manuscript (Fig. 4 in the revised manuscript) indeed the characteristics differ notably between the two patterns (e.g., pattern 27 has faster wind speeds, more frequent LLJs, and a lower mean LLJ altitude) which at least partly explain the slight difference between the potential temperature structures of the two patterns.

While we understand that 30 nodes seems like a lot, it is a manageable number of patterns compared to the total sample size of 1377 radiosondes, such that we can actually visualize and understand the range of ABL vertical structures present in the data in one figure (albeit a complex figure). We tested many options for number of nodes (from 20 to 35), and found that certain ABL structures were missing when fewer than 30 nodes were used (e.g., cases with very strong stability either near the surface or aloft were merged with cases with weaker stability, and cases with varying elevated inversion height were merged into the same node).

The detailed view of the different boundary layer profiles present in the MOSAiC data afforded by the SOM analysis allowed us to define meaningful stability regimes that adequately describe the full-range and complexity of boundary layer stability profiles in the Arctic.

It is common practice when applying SOMs that the user subjectively determines the optimal number of SOM nodes, depending on the goal of their study (see Reusch et al., 2005; Sheridan and Lee, 2011; Cassano et al., 2015; Cassano et al., 2016; Nigro et al., 2017; Dice and Cassano, 2022). Additionally, many other SOM studies have used 30 or more nodes (Sheridan and Lee, 2011; Cassano et al., 2016; Nigro et al., 2017), so we do not stray from common practice in doing the same.

[revised manuscript text omitted]

***In fact, by reducing everything to 12 classes in Table 2, the authors themselves pretty much abandons the SOM, and does something else that does not require it; the SOM part becomes underutilized and, in the end, doesn't add much to the final results, mostly based on the 12 classes and not on the 30 nodes.***

In order to determine the 12 stability classes used for much of the study, the SOM was required. Without the SOM, it would have been nearly impossible to determine the range of stability structures (considering both near-surface and aloft) present in the current MOSAiC-based analysis, such that we could develop criteria and thresholds to encompass all of them with a relatively concise set of stability regimes. Therefore, to justify the 12 stability regimes and their thresholds, it is necessary to share the SOM analysis. The SOM also has important utility on its own. It

allows us to compactly visualize the range of ABL structures in the Arctic, from nearly 1400 soundings, with a great level of detail, which provides context that the reader can refer to when any stability regimes and associated analysis are discussed throughout the paper. Additionally, the SOM provides insight beyond what the 12 stability regimes can tell us - for example, the SOM shows that for a profile with NN-MSA stability, the depth of the ABL and the strength of the elevated inversion can differ quite a bit from case to case (or SOM node to SOM node). In doing this work we found that if we simply used the 12 stability classes and created composite profiles based on all soundings that correspond to each stability class to depict the different vertical structures present in these 12 stability classes, many details were smoothed out or lost due to different heights of stable layers and varying intensity of the stable layers. As such, the SOM is critical in allowing us to clearly show the range of vertical profile types present in a year of soundings from the central Arctic.

Not only is this important for the content presented throughout the current manuscript, but is also relevant for future studies, such as determining if a weather or climate model can simulate the full range of vertical structures revealed by the SOM (this work is currently in process for evaluation of the NOAA Coupled Arctic Forecast System model and plans are being made for a multi-model evaluation with an ensemble of Arctic CORDEX models).

We have added some text throughout the manuscript to state more clearly these points:

"A self-organizing map (SOM) analysis (which objectively identifies a user-selected number of patterns present in a training data set) was conducted with the radiosonde profiles to reveal the range of vertical structures observed during MOSAiC (differentiated by stability within the ABL and the height and strength of a capping inversion), and their relative frequencies during the MOSAiC year. The SOM results were used to develop criteria to define stability regimes characterized by stability both within and above the ABL, such that their relative frequencies and relationships to ABL, LLJ, and clouds characteristics could be analyzed." (line 126)

"Results from the SOM analysis will focus on the frequency of occurrence of each pattern and the variability in the vertical structure depending on time of year (e.g., which SOM patterns largely occur in certain seasons). Seasonal analysis in this paper is carried out by grouping observations during September, October, and November as fall; December, January, and February as winter; March, April, and May as spring; and June, July, and August as summer. Additionally, profiles of wind speed (produced by interpolating the zonal and meridional components to the 5 m grid and then calculating total wind speed profiles) and LLJ characteristics in the context of the SOM patterns will be analyzed. Lastly, once the full range of vertical structures was revealed by the SOM, this information was used to develop a set of criteria for classifying stability of any given observation that distills the detail of the SOM to the most critical factors of stability within and above the ABL." (line 301)

"These stability regime definitions are based on the range of profiles seen in the SOM" (line 313)

*I would also like to know much more about how the SOM analysis handles structure versus geometry. Will, for example, profiles profiles with similar structure but very different geometry end up in the same node? Consider, for example, two inversion capped but well-mixed ABLs, one with an inversion base at 200 m and the other at 1 km, end up in the same node?*

The SOM will not group two inversion-capped but well-mixed ABLs with vastly differing inversion base heights into the same node. This is precisely the benefit of a SOM over other methods of grouping data (such as the more simplified stability regime categorization). The SOM training identifies the user specified number of profiles that minimize the squared difference between all samples in the training data and the user specified number of SOM patterns. For the current SOM trained with the potential temperature gradient profiles, the SOM separates observations into nodes depending on the strength, base height, and depth of the inversion, with the end result of the 30 nodes showing the continuum between these characteristics. Observations with similar strength stable layers but at vastly different heights will have a large squared difference despite having similar vertical structure that differs only in depth of the various layers. The SOM training will recognize this difference and create multiple nodes to depict the range of depths for these layers (e.g., patterns 1, 2, 3, 7, 14, 20, and 21 are all NN-MSA, but the height of the moderately stable elevated inversion layer varies between ~250 m in the case of node 21 and ~750 m in the case of node 3)

We have added more details to the methods to described how a SOM is trained, and what features it picks out in the current study, highlighting that our SOM specifically differentiates the height, strength, and depth of the inversion.

"A SOM is created by randomly initializing patterns from the input data space and comparing the training data to these patterns. Each sample in the input data is presented to the SOM and compared to all patterns in the initial map. The pattern to which the input data sample is most similar is known as the "winning" pattern, and this pattern, and adjacent neighboring patterns, are modified to reduce the squared difference between it and the input data sample. This process continues for all samples in the training data (Liu and Weisburg, 2011; Cassano et al., 2006) and is repeated thousands of times for the entire training data set until the squared differences between the SOM identified patterns and the training data have been minimized." (line 255)

"Each sample in the training data is mapped to the resulting SOM pattern with which it has the smallest squared difference resulting in a list of samples for each SOM-identified pattern. This list of data samples can then be used to calculate the frequency of each SOM pattern and for additional analyses. (Dice and Cassano, 2022)." (line 272)

"Training the SOM with $d\theta_v/dz$ profiles resulted in an array of patterns differentiated by the strength and height of the $\theta_v$ inversion. As such, observations with similar strength $\theta_v$ inversions which occurred at different heights, and observations with similar heights of the $\theta_v$ inversion but different strengths, were separated into different SOM-identified patterns." (line 294)

***A lot is confusing in the way the SOMs are described, for example there seems to be way more than a 100% of soundings, adding the numbers in each node for a total. BTW, many of the numbers printed in the large node boxes in the figures are so small I neede a magnifying glass to read them. If you insist that the SOM analysis is fruitful, and I'm not saying it isn't, then do it properly and explain it well; that could very well be a useful paper all by itself!***

Perhaps we did not do a good job at describing what is shown in the SOM figures. It is the number in the top center of the node in Fig. 3 of the original manuscript (Fig. 2 of the revised manuscript) which indicates the number of radiosonde observations mapped to that node. If you add these numbers up for all nodes, it equals 1377, which is the total number of radiosonde observations used in the study (as specified on lines 183, 223, 231, and 251 of the original manuscript and lines 212 and 277 of the revised manuscript).

In order to improve the readability and understandability of the SOM figures, we have increased the font size for the numbers and letters in each node, and now include a "subplot key" which demonstrates largely and clearly what is presented in each SOM node subplot (see Fig. 2 and Fig. 4 of the revised manuscript).We hope that these revisions to the SOM figures are helpful.

***Then make a separate paper out of the 12 stability classes, but do a proper analysis and try and find something new.***

We agree with the perspective presented in the above comment. It is for that reason that we have already submitted a separate paper focusing on the 12 stability classes, which looks in more detail at the relationships between thermodynamic and kinematic turbulent processes and the stability regimes, as well as how these relationships may differ by season. We had realized that it would too much to include all of that information in a single paper which also contains the SOM analysis. Thus, it was decided to use the separate paper to focus on more nuanced processes and results, and use the current paper to simply describe the annual characteristics of the vertical structure of the Arctic atmospheric boundary layer, from both a SOM and stability regime perspective, and some important atmospheric characteristics.

This other paper (which is currently under review) is mentioned on line 731 of the original manuscript, but we have added some more discussion to make clear that such analysis has been completed and submitted in a complementary paper:

"A complementary paper (Jozef et al., 2023b) explores the role of kinematic (e.g., wind characteristics forced by synoptic setting) and thermodynamic (e.g., surface radiation budget forced by clouds) processes that contribute to, and are modified by, vertical structure and stability conditions, so such details are not heavily discussed in the current paper. " (line 115)

"A complementary paper (Jozef et al., 2023b) delves deeper into the impact of atmospheric radiative and mechanical forcings on ABL stability, and how these relationships vary by season, with a focus on the peculiarities of summer processes, through additional analysis of the synoptic setting, surface radiation budget, near-surface mixing ratio, and fog observations." (line 660)

*Work more with the crieria; ask yourself, for example, what would be necessary to characterize a decoupled stratocumulus, with a high inversion and most of the turbulence is generated in the cloud layer, or an inversion-and-stratocumulus-capped relatively deep but coupled ABL where the buoyancy from the cloud top generates a much deeper total ABL than motivated by the surface fluxes (e.g. Brooks et al. 2018 in JGR).*

These different conditions likely contribute to differences in vertical structure presented by different SOM nodes which were then used to develop the 12 stability regime definitions. For example, it was through the SOM that we realized it was important to separate a well-mixed ABL into the cases with a shallower ABL (the VSM regime) and cases with a deeper ABL (the WS and NN regimes), due to the notion that different conditions/processes are at play when we have a VSM case versus a deeper WS or NN regime, such as what you describe above. We have added some discussion to the text to include your example as possible reasoning for differences between depth of the well-mixed layer:

"One explanation for differing depths of a well-mixed layer is whether the ABL is coupled to a stratocumulus cloud layer: a coupled cloud supports a deeper ABL that is well-mixed up to cloud base whereas a decoupled cloud is separated from a shallower ABL by a $\theta_v$ inversion below cloud base (Brooks et al., 2017)." (line 575)

"Low clouds, correlated with greater LWP, were observed with greater frequency for cases with weaker stability both within the ABL and aloft, highlighting the ability of low clouds and enhanced moisture content to support turbulent mixing both near the surface through enhanced downwelling longwave radiation, and below cloud base though cloud top radiative cooling. In such cases, a well-mixed ABL can be coupled to the cloud layer and extend through the depth of the cloud to cloud top, though a shallow stable layer may decouple a well-mixed ABL from a low cloud. Conversely, mid-level and high clouds were observed with greater frequency for cases with stronger stability, highlighting that in such cases, the cloud is likely to be decoupled from the surface, allowing the strong stability to persist." (line 644)

*There is a lot to be gained here but it does need an insightful analysis. A very stable boundary layer is, as we all know, shallow with large stability, large wind shear and Rib but low u\*; everything in Figure 7 is just a confirmation of textbook ABL meteorology. If you select your criteria this way, cases will have all these other characteristics automatically; no use even looking at the result and the question has become the answer, while no one learned anything new.*

While we do recognize that some of the results shown in Fig. 7 ($d\theta_v/dz$, Rib) of the original manuscript automatically fall out from how the regimes were defined, many do not. We argue that the fact that MOSAiC data behave as suggested in textbooks is valuable, even if unsurprising. The Arctic boundary layer is unique in many ways and it is valuable to demonstrate that despite this uniqueness many features commonly observed elsewhere are also observed in the Arctic. Thus, for the sake of only retaining the most interesting information, we have excluded the panels for $d\theta_v/dz$ and $Ri_b$ from the revised manuscript, as they are both a direct reflection of how we define the stability regimes. We retain ABL height, $dV/dz$, and $u_*$ because these results, while consistent with textbook ABL meteorology, sill have the added value of providing specific quantities in the context of a more detailed classification of stability regime than has been presented in previous literature (Fig. 5 of revised manuscript).

*In Figure 8, nothing except panel d, is significantly different from anything else and most of the conclusions are hand waiving. WS & NN are the deepest and hence expected to have the largest jet wind speeds simply by their*

*distance from the surface; only surface friction can reduce wind speed; ABL turbulence just mixes things around.*

The reviewer brings up a valid point that many of the conclusions in Fig. 8 of the original manuscript are not significant, and the discussion on them was lacking substance. Thus, we have removed panels a (LLJ height) and c (LLJ depth) for the revised manuscript (see Fig. 6 of the revised manuscript). We retain the LLJ core height minus ABL height and the LLJ speed because they provide the most interesting conclusions. We do however disagree that WS and NN would have larger LLJ speeds simply because of their distance from the surface, because these LLJs are not actually farther from the surface. We showed in Fig. 5 of the original manuscript (Fig. 4 of the revised manuscript) that LLJ height is relatively consistent regardless of stability regime (the LLJ is just farther from the ABL height for stronger stability regime). So this suggests turbulence produced from shear below an LLJ core likely contributed to the development of the ABL into WS or NN stability, which is the topic of a paper recently posted to preprint by Egerer et al. (2023). We address this more clearly in the revised manuscript:

"Results regarding LLJ height, specifically its relationship to ABL height, support the notion that both baroclinicity and inertial oscillations contribute to LLJ formation in the Arctic. For the SS, MS, and the VSM regimes (represented by patterns on the right half of the SOM), the LLJ core was situated a greater distance above the ABL than for the WS and NN regimes (represented by patterns on the left half of the SOM). This greater distance suggests decoupling between the relatively stable ABL and the LLJ, which is consistent with inertial oscillations as an LLJ formation mechanism. The smaller distance between the ABL and LLJ core for the weaker stability regimes suggests greater coupling between the well-mixed ABL and the LLJ, so inertial oscillations are unlikely to be the formation mechanism, and rather baroclinicity is the more probable cause. The results show that such LLJs have faster speeds, in agreement with Jakobson et al. (2013). The similarity in LLJ core height despite varying stability occurs because of these two different formation mechanisms. Thus, an LLJ can be both a cause and an effect of stability. For a well-mixed or weakly stable ABL, LLJs contribute to the creation of the mechanical turbulence that mixes the ABL. For more strongly stable ABLs, an LLJ can be an effect of the strong stability if the above atmosphere becomes decoupled from the surface." (line 616-628)

Egerer, U., Siebert, H., Hellmuth, O., and Sorensen, L. L.: The role of a low-level jet for stirring the stable atmospheric surface layer in the Arctic, EGUsphere [preprint], https://doi.org/10.5194/egusphere-2023-567, 2023.

*Temperature inversions (Figure 9) are common – everywhere! You find them also in the mid-latitude convective ABL. What I think is special for the Arctic is that: i) the ABL is so often quite shallow; ii) when not shallow, it is often capped by stratocumulus providing the extra energy explaining the depth, and iii) when it is stable (which is mostly in winter), it can be so very stronly stable for a long time, since there is no diurnal cycle that resets the stability once per day. If you base a category on having weak stability aloft, is it surprising that this class stands out when you analyze that inversion, and what does a median inversion-base height at 2 km has to do with the ABL? And how can the WS alone class have a larger "inversion intensity" than the WS-MSA and WS-SSA classes; ins't that counterintuitive?*

We agree with the reviewer's critique of this part of the manuscript. Since the temperature inversion results are largely a function of how we classify stability regime, and do not provide very interesting results, we have removed the temperature inversion results and discussion from the revised manuscript.

*Going on, the WS ABL is formed by surface cooling, so it stands to reason that it has no clouds or at least a high cloud base; clouds higher than a few km in winter has very little effect on the surface energy budget. The lowest clouds seem to be found in well mixed or near neutral cases; one may wonder if that is because these low clouds force that particular stability? Or is it the other way around?*

The secondary paper which we discussed in response to a previous comment (and which is mentioned on line 115 and 660), currently under review for publication, addresses these questions, and thus we have opted not to go into that level of detail in the discussion in the current manuscript.

*Finally, drop the AUV data! For two reasons: i) I can't see that it adds anything useful, and ii) the risk that it contaminates the statistics; radiosounding where done every day regardless of weather but UAVs flew only occasionally during a part of the year.*

We appreciate this comment from the reviewer. We had gone back and forth about whether or not it was valuable to include the UAS data in the current paper and had decided that it may be of interest to some readers to see a comparison of UAS and radiosonde profiles. However, given your feedback, we are happily willing to remove this from the manuscript, and have done so for the revised copy.

*I could go on for many more pages, and list many detailed complaints and things I don't understand, and I would have if I could find it in me to recommend - at least - major revision. It makes me somewhat sad to have to be so negative; obviously this looks like a failure in supervision. But I wouldn't be doing my part in upholding the quality of the journal if I let this through – I'm so sorry.*

We greatly appreciate the thoughtful comments you have provided and we have taken your comments very seriously. Addressing your comments has helped produce a revised manuscript that we believe is greatly improved. We hope you agree, but if there are points that remain to be of concern, or that still cause confusion, we would be happy to continue working to address them.

**Anonymous referee #2**

*This manuscript describes a statistical analysis for a year-round period of Arctic boundary layer observations based largely on radiosondes during the MOSAiC project accompanied by data observed by the DataHawk2 unmanned vehicle. As a central tool, a "self-organizing map" approach was applied to the data. There is no doubt that such a statistical analysis is extremely helpful in addition to all the case studies that have been and are being evaluated. Therefore, the enormous amount of work is greatly appreciated, and after major improvements to the manuscript, I also support the publication of this analysis. However, the manuscript needs a thorough revision that goes far beyond classical "major revisions". I will try to justify this in detail below.*

*My main concerns are:*

*The manuscript is extremely difficult to read and understand; this is largely due to the very intensive use of abbreviations. Especially when referring to the different stability regimes of the boundary layer, the abbreviations are not really intuitive and hardly anyone will be able to remember them while reading (see at line 428/29 for example when a sentence almost completely is based on abbreviations). In particular – see Tab 2- the capital "S" is sometimes used for "shallow" and sometimes for "stable" – I have no good suggestion at the moment to improve this, but please consider of a better and simpler way to categorize the different regimes and avoiding abbreviations.*

We understand and appreciate your concern. To make the manuscript easier to follow, we have made the following adjustments throughout the paper.
- Reorganized the flow of the paper such that there are separate "Results" and "Discussion and conclusions" sections.
- Cleaned up the SOM figures and provided detailed keys
- Removed some panels from the box and whisker plots that did not add much to the discussion
- Removed all figures and text related to temperature inversions and the UAS data to provide a more concise manuscript

Additionally, where appropriate, we have done our best to rather refer to the general near-surface and aloft stability categories as they relate to the results, instead of listing off the abbreviations.

However in large part, we believe that use of abbreviations is still the best option, and hope that the other changes we have made throughout the manuscript make it easier to read and understand. There are a few reasons for this decision: 1) We believe if a reader takes a few extra moments to review Table 2, they will be able to follow the abbreviations thoughtful the paper. In the end, there are only 5 near-surface regimes and only 3 aloft regimes (which are the same as the near-surface regimes, but only adding an A to indicate "aloft") to remember; 2) A complementary paper containing the same stability regimes as included in the current paper is also under review for publication, and no reviewers have expressed concerns with being able to keep track of them; and 3) If we were to entirely avoid abbreviations, we would instead need to write out the regimes when they are mentioned, but this would perhaps make the paper even more long and confusing, as several of the regime names are quite lengthy.

***Very often it is concluded in the paper that the analysis yields unsurprising results or that the results are logically and physically explainable - well, I expect that with the correlations but do you want to evaluate the tool or deliver new scientific findings?***

To address this concern, we have made a few adjustments. First, we have now more clearly communicated one of the goals of the paper in the introduction – to use the MOSAiC dataset to see if "textbook" ABL meteorology holds true in the central Arctic, while providing new insight into the quantitative values for the characteristics we show as they relate to stability.

"The results of such a study are firstly valuable to reveal whether current observations agree with past observations and well-known ABL meteorological processes. Additionally, through the use of new methods (i.e., the SOM analysis and detailed stability regime classification), the results also provide further constraints on the vertical structure and features of the Arctic lower atmosphere that may be helpful to improve parameterizations of the central Arctic in weather and climate models." (line 132)

Thus, when we later claim that aspects of the analysis yield unsurprising results, the reader will better understand that this is still an important finding as it relates to the goals of the paper. Next, we have adjusted the results and discussion to focus more heavily on new results which were discovered through this analysis, and have removed the panels from the box and whisker plots that show results which are largely a function of how we define stability.

***It is a bit tiring for the reader to have each figure described in such detail (and the figures contain a large amount of detail...), and you should try to find a slightly better and more compact way of presenting and introducing the figures. I know this comment is quite generic but maybe you find a good way to describe your figures in a more comact way.***

We have worked to make the description of the figures more concise throughout the paper. One way that we do this is by making the figures themselves easier to read (through larger font and removing unnecessary information) and understand as stand-alone entities (through the addition of detailed keys in the SOM figures which explain what we see in each subplot), and thus less description is necessary in the text. Additionally, the separation of the content of the paper into distinct "Results" and "Discussion and conclusions" sections, allows us to spend less time describing each figure when it comes up in the paper.

***When I first read the manuscript and got to Figure 2, I was completely overwhelmed. Why do you need 30 schemes to describe the ABL? With many patterns you only see marginal differences when you look very closely. I think that the manuscript could be made much simpler and more readable if the analysis was limited to a handful of characteristic patterns.***

We completely understand being overwhelmed with Fig. 2 in the original manuscript. Thus, as this figure is not instrumental for understanding the analysis (and is rather a demonstration of the methods), we have simplified the figure and moved it to the supplement (see Supplementary Fig. S1 of the revised manuscript). This way, if a reader is very interested to know more about what the 30 SOM patterns look like, and the spread in the observations around the SOM pattern in a given node, they can refer to it, but it is not required to understand the rest of the paper.

Additionally, we understand your questioning as to why we need 30 patterns to describe the ABL, and realize that we need to better describe the purpose of a SOM in the Methods section in order to convince you (and other readers) that 30 patterns are necessary.

The purpose of a SOM is not to find the smallest number of patterns which represent all the possible structures in the data. Rather, a SOM is meant to continuously depict the range of structures present in the training data from one SOM-identified profile to the next - in this case, we show the continuum of ABL vertical structure in the Arctic. Here, the SOM is being used as a way to visualize a large dataset (1377 soundings) in a manageable way while also allowing subtle details in the vertical structure (such as varying heights of stable layers, differing strength of stable layers) to be identified. For example, at first glance, patterns 27 and 28 in Fig. 3 of the original manuscript (Fig. 2 in the revised manuscript) appear to be very similar. However, with a deeper look, we see that the ABL depth between the two patterns differs discernably (leading one to be classified as WS, and the other as VSM), and the strength of the elevated inversion is a bit stronger for pattern 27 (even though they both qualify as -SSA). Then when looking at differences in wind speed and LLJ characteristics for these two patterns in Fig. 5 of the original manuscript (Fig. 4 in the revised manuscript) indeed the characteristics differ notable between the two patterns (e.g., pattern 27 has faster wind speeds, more frequent LLJs, and a lower mean LLJ altitude) which at least partly explain the slight difference between the potential temperature structures of the two patterns.

While we understand that 30 nodes seems like a lot, it is a manageable number of patterns compared to the total sample size of 1377 radiosondes, such that we can actually visualize and understand the range of ABL vertical structures present in the data in one figure (albeit a complex figure). We tested many options for number of nodes (from 20 to 35), and found that certain ABL structures were missing when fewer than 30 nodes were used (e.g., cases with very strong stability either near the surface or aloft were merged with cases with weaker stability, and cases with varying elevated inversion height were merged into the same node).

Additionally, many other SOM studies have used 30 or more nodes (Sheridan and Lee, 2011; Cassano et al., 2016; Nigro et al., 2017), so we do not stray from common practice in doing the same.

Cassano, J. J., Nigro, M., and Lazzara, M.: Characteristics of the near surface atmosphere over the Ross ice shelf, Antarctica,  J. Geophys. Res.-Atmos., 121, 3339-3362, https://doi.org/10.1002/2015JD024383, 2016.
Nigro, M. A., Cassano, J. J., Willi, J., Bromwich, D. H., and Lazzara, M. A.: A Self-Organizing-Map-Based Evaluation of the Antarctic Mesoscale Prediction System Using Observations from a 30-m Instrumented Tower on the Ross Ice Shelf, Antarctica, Weather Forecast, 32, 223-242, https://doi.org/10.1175/WAF-D-16-0084.1, 2017.
Sheridan, S. C. and Lee, C, C.: The self-organizing map in synoptic climatology research, Prog. Phys. Geog., 35, 109-119, https://doi.org/10.1177/0309133310397582, 2011.

In order to address all of these points, we have:

1) Added more description in the 'self-organizing map analysis' section explaining how a SOM works, including how it is trained, what the goals of such an analysis are, what the resulting product is:

[revised manuscript text omitted]

Lastly, while we use the 30 pattern SOM as a starting point for understanding the range of stability profiles present in the MOSAiC soundings, the remainder of the paper does in fact use a reduced number of patterns (stability regimes) to perform further analysis. We view the SOM and the stability regimes as complementary, with the SOM showing details of how the vertical profile of stability varies, while the stability regimes distill these details to the

most critical factors of near surface and aloft stability. We have added some text to more explicitly state these things:

"… a more tangible demonstration of the range of vertical structures present during MOSAiC is shown in Fig. 2 (Sect. 3.1) with the mean profiles of $d\theta_v/dz$ and $\theta_v$ anomaly for all radiosondes mapped to a given pattern. Results from the SOM analysis will focus on the frequency of occurrence of each pattern and the variability in the vertical structure depending on time of year (e.g., which SOM patterns largely occur in certain seasons). Seasonal analysis in this paper is carried out by grouping observations during September, October, and November as fall; December, January, and February as winter; March, April, and May as spring; and June, July, and August as summer. Additionally, profiles of wind speed (produced by interpolating the zonal and meridional components to the 5 m grid and then calculating total wind speed profiles) and LLJ characteristics in the context of the SOM patterns will be analyzed. Lastly, once the full range of vertical structures was revealed by the SOM, this information was used to develop a set of criteria for classifying stability of any given observation that distills the detail of the SOM to the most critical factors of stability within and above the ABL." (line 299-309)

***A general note on writing style: please try to avoid repetition to strengthen the manuscript. Furthermore, the sentences are often so complicated and convoluted that a fluent reading - at least for me - was very difficult or impossible. I myself am not a native speaker, but there are enough competent co-authors who can edit the manuscript thoroughly.***

We have worked throughout the manuscript to avoid repetition and simplify the sentences. Again we believe that following your suggestion to separate the results and discussion into two separate sections helps with this.

***General comment about most of the figures (although here I refer explicitly to Fig. 3): The figure is based on 30 subplots which are by definition quite small but if you try to include even more information in terms of several additional numbers and vertical or horizontal lines, the plots will get really crowded. Even worth in Fig 4 where I am not able to read at all the numbers you included into the subplots – they are simply too small and too many.***

We have made several adjustments to make the complex SOM figures easier to read and understand:

Fig. 3 in the original manuscript (Fig. 2 in the revised manuscript):
- Removed the SOM profile – this is shown in what is now supplementary Fig. S1, and it is not necessary for the discussion of this figure
- Made the font of the numbers and letters in each subplot bigger
- Made the axes fonts bigger
- Added a "subplot key" which demonstrates clearly what is shown in each subplot, so that a reader can refer to it, rather than digging through the figure caption to figure out what they are looking at.
- We have opted to retain the horizonal and vertical lines because we think that they importantly demonstrate how each SOM pattern fits into the various stability regime classifications. However, if the referee still feels that these lines should be removed, we are willing to consider removing the lines.

Fig. 4 in the original manuscript (Fig. 3 in the revised manuscript):
- Removed the number in the upper center of each box which indicated the number of radiosonde profiles mapped to that SOM pattern. This information is included in Fig. 3 in the original manuscript (Fig. 2 in the revised manuscript), so is not needed here.
- Removed the number in the center of each box which indicated the number of cases in that pattern/season. We only retain the percentage, as this is the more valuable number for understanding the results
- Made the font of the numbers and letters in each box bigger.
- Added an opaque white box behind the percentage written in each box, so it can be better seen regardless of the shading in each box
- Made the seasonal subplots bigger
- We also include now the annual frequency subplot in this same figure, in response to your next comment (we discuss this figure in the text so you suggest this not be in the supplement).

Fig. 5 in the original manuscript (Fig. 4 in the revised manuscript):
- Made the font of the numbers and letters in each subplot bigger
- Made the axes fonts bigger
- Removed the number in the upper center of each box which indicated the number of radiosonde profiles mapped to that SOM pattern. This information is included in Fig. 3 in the original manuscript (Fig. 2 in the revised manuscript), so is not needed here.
- Added a "subplot key" which demonstrates clearly what is shown in each subplot, so that a reader can refer to it, rather than digging through the figure description to figure out what they are looking at.
- Due to the above bullet point, we have removed "LLJ:" and "2m" from each subplot, as the "subplot key" now indicates what these numbers mean.

***I am not convinced that discussing material in the manuscript that has been moved to a supplement is the correct or formal way to do it. If you have to many figures, you should solve this problem differently.***

We have reorganized the content of the paper to include all figures which are heavily discussed in the main text of the manuscript, rather than in the supplement. Specifically:
- The annual SOM pattern frequencies (Supplementary Fig. S1 in the original manuscript) has been moved to the main text, and combined into the figure with the seasonal frequencies (Fig. 3 in the revised manuscript)
- The DH2 figure (Supplementary Fig. S2 in the original manuscript) has been removed entirely, as we have chosen to remove the UAS analysis from the current paper (in response to concerns from the other reviewer)
- LLJ frequency (Supplementary Fig. S4 in the original manuscript) has been moved to the main text, and combined into the figure with the LLJ characteristics (Fig. 6 in the revised manuscript)
- TI frequency (Supplementary Fig. S6 in the original manuscript) has been removed entirely, as we have chosen to remove presentation of TI characteristics from the current paper. See response to your next comment for reasoning.

The only figures which remain in the supplement are the figure showing SOM pattern profiles and percentiles of observations (this is adapted from what was Fig. 2 in the original manuscript, and is now Supplementary Fig. S1 in the revised version) and the grid plots showing statistical significance. These figures are all minimally discussed, and not instrumental to understanding the results presented in the paper but may be of interest to some readers seeking additional details not contained in the main text and figures.

***About the analysis of temperature inversions: I am a little bit skeptical about this analysis and I wonder of how much of these results are based on self-correlation because the definition of the stability regimes is also based on temperature gradients. You mentioned this issue briefly but this needs to be discussed in more detail.***

We agree with the reviewer's critique of this part of the manuscript. Since the temperature inversion results are largely a function of how we classify stability regime (as you note), and do not provide very interesting results, we have removed the temperature inversion results and discussion from the revised manuscript.

***Maybe one solution for an improved structure of the entire manuscript would be a stricter separation between explaining the results in one section and discussion and interpretation in another one.***

In the revised manuscript, we now separate the content such that the results and corresponding figures are presented in one section ("Results"), and the discussion and interpretation is presented in another section ("Discussion and conclusions"). We believe this helps create better flow of the paper, such that it is easy to understand, and repetition is avoided.

***I am not convinced about the meaningful interpretation of parameters averaged over the entire MOSAiC cruise. For example, what can I learn from a statement such as "The average ABL height during MOSAiC was 150 m, and ABL height increases with decreasing stability. "***

*(line 703)? You average ABL height over completely different ABL regimes and the second part of the sentence is somewhat trivial and expected – right?*

This is a good point. In the revised manuscript, we only share parameters averaged over the entire MOSAiC cruise when that parameter is largely unchanged between stability regimes (e.g., mean LLJ height is fairly consistent regardless of stability regime). Otherwise, we instead focus on quantities for individual stability regimes (or mean values for groupings of regimes with the same near-surface regime, but different aloft regimes, when appropriate), and how these quantities vary between regimes.

*More specific comments:*

*Introduction (line 35): what do you exactly mean with high temporal and spatial resolution – please specify.*

We have largely rewritten the introduction, and so this comment is no longer applicable. We now introduce MOSAiC by saying:

"Thus, there is much to be gained by analysis of more recent data, such as that from the Multidisciplinary drifting Observatory for the Study of Arctic Climate (MOSAiC; Shupe et al. 2020), which observed the central Arctic following one ice floe for a full year from September 2019 to October 2020." (line 110)

*Line 39ff: quite generic comment; please provide references - I think it is quite obvious that the Arctic ABL is not necessarily stably stratified in summer - right?*

As mentioned above, we have largely rewritten the introduction. Now, we more correctly state that the "Arctic atmosphere over sea ice is typically either stable or near-neutral" (line 39), and then go on to describe in more detail the different mechanisms which contribute to a stable or near-neutral ABL in winter vs. summer:

"Stable conditions are common in Arctic winter (Tjernström and Graversen, 2009) due to persistent longwave cooling in the absence of solar radiation (Brooks et al., 2017) and extended periods of clear skies or thin high clouds (Tjernström and Graversen, 2009), attributable to the lack of open water evaporation. However, intermittent instances of low stratocumulus clouds in winter can force a shallow well-mixed ABL (Morrison et al., 2012; Tjernström and Graversen, 2009; Persson et al., 2002). Such clouds are common during stormy conditions (Brooks et al., 2017; Persson et al., 2002).

Near-neutral or weakly stable conditions are common in Arctic summer (Brooks et al., 2017; Tjernström and Graversen, 2009), often capped by persistent stratiform clouds (Intrieri et al., 2002a; Tjernstrom, 2007; Curry and Ebert, 1992; Liu and Key, 2016; Shupe et al., 2011; Tjernström, 2005, Tjernström et al., 2012; Wang and Key, 2004; Zygmuntowska et al., 2012), which form as ample moisture is advected north either into the Arctic or from the broader ice-free areas across the pan-Arctic region, during the melt season (Sotiropoulou et al., 2016; Tjernström et al., 2019). The ABL is typically decoupled from the cloud layer by a shallow stable layer, such that turbulence is not exchanged between the cloud and the surface (Curry, 1986; Sedlar and Shupe, 2014; Sedlar et al., 2012; Shupe et al., 2013; Sotiropoulou et al., 2014). However, the common advection of warm moist air into the central Arctic can also result in the formation of a shallow, stable ABL (Tjernström et al., 2019; Tjernström, 2005; Cheng-Ying et al., 2011), especially towards the beginning of an advection event, or close to the ice edge (Sotiropoulou et al., 2016; Tjernström et al., 2019). Ice and snow melt in summer may also contribute to the formation of a stable ABL (Kahl, 1990; Gilson et al., 2018)." (line 50-67)

*Line 49ff: I assume that - depending on the temperature stratification - the turbulence maybe also increased above the LLJ core because the shear could be similar - right? Furthermore, why should a LLJ weaken stability? The main preconditions for the existence of a LLJ is an almost vanishing turbulent transfer coefficient - typical for stable regimes.*

Shear above the LLJ core is typically much less than that below the LLJ core since wind speed below the jet core goes to zero at the surface while that is almost never the case above the LLJ core. The criteria for an LLJ is that the

core speed is just 2 m/s greater than the minimum in the wind speed above the jet core. The mean jet core speed during MOSAiC was 11.5 m/s, which is much greater shear when compared to 0 m/s wind speeds at the surface, versus the diminishment of wind speeds above the jet core. However, the wind shear that does exist above the jet core may partly explain diminishment of the capping inversion, and we note that now:

"For both LLJs forced by baroclinicity and inertial oscillations, enhanced wind shear above the jet core may also contribute to turbulent mixing above the LLJ." (line 91)

There are two primary situations in which an LLJ will occur in the Arctic. One such situation is the one you describe: the LLJ forms due to inertial oscillations and is decoupled from the (typically shallow and stable) ABL, such that wind shear from the LLJ does not weaken stability. The other situation is when an LLJ forms due to a baroclinic environment, where wind speeds decrease with altitude according to the thermal wind relationship, and surface friction reduces wind speeds to 0 at the surface, this resulting in an LLJ some distance above the surface. In this case the LLJ is not decoupled from the ABL, and can contribute to weakening of stability in the boundary layer through the enhanced shear from the LLJ.

We have added more discussion in the introduction to clarify these two situations under which an LLJ may form, and the subsequent relationships with ABL stability.

"There are two primary forcing mechanisms for LLJs in the Arctic: baroclinicity and inertial oscillations. Baroclinicity in the Arctic most often occurs near the ice edge (Brümmer & Thiemann, 2002) or due to the passing of a transient cyclone (Jakobson et al., 2013) which creates regions of enhanced temperature contrasts (Koyama et al., 2017). Depending on the wind direction, the horizontal temperature gradient causes the geostrophic wind speed to decrease with height according to the thermal wind relationship (Stull, 1988). This, paired with diminishment of wind speeds at the surface due to friction (Stull, 1988), contributes to the formation of an LLJ at some distance above the surface, typically just above the ABL (Brümmer & Thiemann, 2002). Thus, an LLJ forced by baroclinicity is typically coupled to the surface, and can cause weakening of stability within the ABL due to enhanced shear below the jet core (Banta, 2008; Egerer et al., 2023).

Inertial oscillations in the Arctic can be induced after well-mixed conditions are replaced by increased near-surface stability, for example, after the passing of a storm (Andreas et al., 2010a; Jakobson et al., 2013). In such cases,  air aloft becomes decoupled from the surface, ceasing frictional drag, which, along with the impact of the Coriolis force, allows the winds aloft to accelerate to supergeostrophic speeds (Blackadar, 1957; Stull, 1988; Jakobson et al., 2013)." (line 78-90)

***Line 101: although you provide an explanation later on, I think a short introduction of what a***
***"self-organizing map" is should be included - simply because you mention it and I think a few words about this technique is essential already at this point and not all readers are knowing about this?!***

We have added a short statement in parentheses when the SOM is first introduced, to give a brief summary of what it is:

"A self-organizing map (SOM) analysis (which objectively identifies a user-selected number of patterns present in a training data set) was conducted with the radiosonde profiles to reveal the range of vertical structures observed during MOSAiC…" (line 126)

***Methods:***

***Line 116 ff: is this information about RV Polarstern movement is of interest for your work here – why do you mention this?***

You bring up a good point that this statement about when the *Polarstern* travelled under its own power is not relevant to the current work. Thus, the statement has been removed.

***About Tab 1:***

• ***What are the sources for the uncertainties? The Vaisala manual? Please provide a reference.***

We have added a reference for the Vaisala RS41-SGP uncertainties.

"Instrument specifications and uncertainties for the radiosonde variables are available at: https://www.vaisala.com/sites/default/files/documents/WEA-MET-RS41SGP-Datasheet-B211444EN.pdf (Vaisala Radiosonde RS41-SGP, 2017), and are summarized in Table 1." (line 155)

We have added the citation for the ceilometer uncertainty.

"CBH derivation and uncertainty is discussed in Morris (2016)." (line 178)

Uncertainties for the meteorological tower variables were provided by one of the co-authors, but are also included in Cox et al., submitted, which we hope will be published soon. We already provide a source for the uncertainty in the microwave radiometer data.

• ***Furthermore, I have serious concerns about the given uncertainty of wind observations: I know that this is the value given in the specifications by Vaisala but there is a lot of discussion about errors in determining the wind velocity - in particular at high latitudes where GPS comes to its end.***

We have added a statement to note that the true uncertainty in the winds is likely higher than that given in the datasheet, but this is probably not due to the GPS issues. It is our understanding that GPS issues at high latitudes are largely in vertical accuracy, as all the satellites are down at the horizon. However, because there is visibility on many satellites at any given time even in the Arctic, the horizontal position accuracy (which is used to determine the wind speeds) should be pretty good. Given this, and some other reasons (see text below), we find the winds to be sufficiently reliable for this study.

"It is recognized that the true uncertainty in the winds is likely to be greater than that provided in the data sheet, however after determining that our results changed minimally when additional vertical averaging was applied to the winds (beyond the filtering already applied by Vaisala during their data processing), we find the original winds provided in Maturilli et al. (2021) to be sufficiently reliable for the current study." (line 157)

• ***Why do you mention uncertainties above 16 km here?***

We had mentioned uncertainties above 16 km (for the temperature) and at < 100 hPa (for pressure) simply because they are also included in the data sheet for the Vaisala RS41-SGP. However, as you brought to our attention, uncertainties at those altitudes are not necessary to include in the current manuscript, since the study only uses data at lower levels. Thus, we have removed these high altitude uncertainties from the table.

• ***I don't understand why a sonic is not enough to estimate the friction velocity?***

This is because we use bulk friction velocity, rather than the standard eddy-covariance (EC) value. We have added some text which states more explicitly the difference between the bulk friction velocity and the EC value, as well as justification for our choice.

"Bulk $u_*$ was chosen, as opposed to the standard eddy-covariance value, as the bulk parameterization considers both wind fluctuations and latent heat fluxes (developed using guidance from eddy-covariance data collected during SHEBA; Andreas et al., 2010b) which is more comparable to $u_*$ used in models (e.g., Fairall et al., 2003)." (line 172)

***Line 165ff: How can you expect a slope for higher altitudes just based on the lowermost 10 m? Why not simply compare the highest measurement point of the tower with the lowermost observation level of the radiosonde - I assume 12 m (helideck) and 10 m***

*(top of the mast) should compare quite well? If not, you have a problem with the*
*radiosonde – right? Or did I completely misunderstand your approach? – Simply*
*double check the wording.*

Since the lowest measurement in the radiosonde is indeed at ~12 m, and the highest tower measurement is at 10 m, we would expect them to line up well, with a similar slope, if there is no issue with the radiosonde measurements (as you state). However, there IS often an issue with the radiosonde measurements in these lowest levels due to the local "heat island" resulting from the presence of the Polarstern (which we already describe in the original manuscript). Thus, we use the tower measurements to identify where the radiosonde measurements are likely incorrect and remove this data. We have adjusted the wording to hopefully make this method more clear:

"Thus, if this "convective layer" was present, then the lowest radiosonde measurements were visually compared to measurements from the met tower to confirm whether the radiosonde measurements were indeed incorrect (e.g., if the lowest few radiosonde measurements were notably warmer then the tower measurement at 10 m). The first credible value of the radiosonde measurements was then taken to be the point at which the tower measurements extrapolated upward would line up with the observed radiosonde measurement, or in the case of a temperature offset between the tower and radiosonde, would have approximately the same slope. All data at the altitudes below this first credible value were removed." (line 194)

*Line 168: interpolated => extrapolated? Please check.*

Extrapolated is probably the better word. Thank you for this suggestion. The change has been made:

"The first credible value of the radiosonde measurements was then taken to be the point at which the tower measurements extrapolated upward would line up with the observed radiosonde measurement…" (line 196)

*Line 172: I thought a low-pass filter removed this pendulum motion? Please comment*
*on this.*

It is probably true that the effect of the pendulum motion was already removed during Vaisala's filtering of the data, as we don't really see evidence of this in the published processed data. Thus, we have removed this statement from the revised manuscript.

*Line 185: I am somewhat surprised by the high critical Ri value which is two-times*
*higher compared to the "classical" value. Most values published in literature are below*
*0.25 – do you have an explanation for this?*

An explanation for this is described in detail in the Jozef et al. (2022) paper (the subject of which is testing the efficacy of various ABL height detection methods), but in summary this higher critical value is necessary with higher resolution data than is typically used when the "classical" value of 0.25 applies well, and may also be attributed to the methods of calculating $Ri_b$ over a running bin of 30 m throughout the profile (rather than always calculating $Ri_b$ with respect to the surface). Jozef et al. (2022) found that a critical value of 0.25 always identifies an ABL height that is much too shallow for the MOSAiC radiosonde data. We have added some brief text to clarify that this information can be found in the Jozef et al. (2022) paper, as we believe it would take too much space to sufficiently justify in the current manuscript:

"The methodology for calculating the $Ri_b$ profile used to identify ABL height, as well as justification for the use of 0.5 as a critical value (rather than the more traditional value of 0.25) is described in Jozef et al. (2022)." (line 218)

*Line 186: I think you can shorten this part a little bit by citing your paper only one times*

We have removed the second citation.

*Line 189: just to understand it correctly: the gradient is a mean gradient from 35 m to*
*ABL top - right?*

The gradient is the overall gradient between 35 m and ABL top: (value at ABL top – value at lowest measurement)/(ABL height – height of lowest measurement). However, we have removed the inclusion of $d\theta v/dz$ over ABL depth from the revised manuscript, and decided it was not necessary to mention the calculation of $dV/dz$ at this point in the manuscript, and thus the sentence has been removed.

*Line 225: What do you exactly mean with "theta anomaly profile"? And in which way is one approach "better" than the other one? - please specify.*

We have clarified this discussion and the text now reads:

"We also tested the SOM trained with the $\theta_v$ profiles rather than the gradient (in the form of the $\theta_v$ anomaly compared to 1 km, to remove seasonal temperature dependence), but found that the range in height and strength of the $\theta_v$ inversion, as well as the differentiation between a weakly stable or near-neutral layer below a $\theta_v$ inversion, were not distinguished." (line 284)

*Line 230 ff: What details are "better" when using 30 patterns instead of 20 or 35? What did I learn from this detail?*

We have added some discussion to convince the reader of the choice for 30 patterns versus 20 or 35:

"Before settling on the 6x5 (30 pattern) SOM, we tested SOMs with size and orientation of 5x4 (20 patterns) to 7x5 (35 patterns). When using 20 patterns, the range in strength of near-surface stability and the varying depths of a weakly stable or near-neutral layer were not fully evident. To fully understand the range of vertical structures in the Arctic, highlighting these differences is important, so the inclusion of additional SOM patterns was necessary. However, with 35 patterns, we found that no additional details were introduced beyond what was shown with 30 patterns. Thus, we determined that 30 patterns is the smallest number to sufficiently describe the range lower atmospheric stability during MOSAiC, retaining fundamental features of vertical structure (e.g., varying height and strength of the $\theta_v$ inversion)." (line 277)

*Line 228ff: I understand that you want to explain details about SOM in a specific part of the paper but you mentioned SOM several time before your explanation - maybe you should at least mention at the beginning what SOM stands for and refer to this point here. I feel that many readers have never heard about SOM before and at least a brief introduction at the beginning could help – or did I have overseen this?!*

We now mention very briefly in the introduction that a SOM is:

"A self-organizing map (SOM) analysis (which objectively identifies a user-selected number of patterns present in a training data set) was conducted with the radiosonde profiles to reveal the range of vertical structures observed during MOSAiC…" (line 126)

Additionally, we have added much more detail about how a SOM works, before going on to explaining how the SOM technique was applied in the current study (line 242-275).

*Line 244ff: Maybe at this point a comment about the low-pas filtering of GRUAN data is useful and how it effects your data and evaluation?! Or why using 5 m as a grid spacing when the low-pass filtering is at 75 m or so? (see also the comment by Günther Heinemann)*

Wind speed was interpolated to a 5 m grid spacing to match the resolution of the interpolated $\theta_v$ profiles, such that an average wind speed per SOM pattern could be calculated and visualized in conjunction with the $\theta_v$ profiles (see Fig 4 of the revised manuscript). It would not make sense to reduce the resolution of the wind speeds, simply because of the low-pass filtering resolution. However, we have chosen to mention the wind speed interpolation a

little later on in the section, as this is not actually relevant for training the SOM, and only confuses the description at this point:

"Additionally, profiles of wind speed (produced by interpolating the zonal and meridional components to the 5 m grid and then calculating total wind speed profiles) and LLJ characteristics in the context of the SOM patterns will be analyzed." (line 305)

As a side note, we use the level 2 radiosonde data, which is not GRUAN processed, but is rather Vaisala processed, because the level 2 data are more reliable at low altitudes. We have added a note earlier in the paper which states that we find the winds provided in the level 2 dataset (processed by Vaisala) to be reliable without additional filtering:

"…after determining that our results changed minimally when additional vertical averaging was applied to the winds (beyond the filtering already applied by Vaisala during their data processing), we find the original winds provided in Maturilli et al. (2021) to be sufficiently reliable for the current study." (line 158)

***About Fig 2.: Maybe I missed it but why do I need 30 patterns to describe typical ABL stratifications? For example, what is the difference between pattern 27 and 28? By eye there is no difference. A technical comment on Fig 2: the pattern number and the number of observations is in the same font and partly not well visible - maybe you could provide a color background for the two set of numbers?***

We have moved Fig. 2 in the original manuscript to the supplementary figures of the revised manuscript, as it is a rather technical figure, and does not intuitively demonstrate the nuanced differences between SOM patterns, which can all be better seen in Fig. 3 of the original manuscript (Fig. 2 of the revised manuscript) which highlights the differences in stability regime between seemingly similar patterns (e.g., pattern 27 is classified as WS-SSA, and pattern 28 is VSM-SSA due to varying ABL depths). We also hope that our added discussion on the utility of a SOM (line 242-275), and how this this applies to the current study (line 276-309) clarifies why 30 patterns are necessary.

For clarity of the figure (now Supplementary Fig. S1), we have:
- Removed the lines for all cases mapping to each pattern (this information is summarized well with the percentile profiles which we have retained)
- Removed the mean and median profiles
- Changed the colors to be more appealing and visible
- Put an opaque background behind the pattern numbers so they are more visible

***Line 259ff: when reading this part, I immediately ask myself if the DH2 observations have a chance to cover all the different patterns because it didn't fly in the Polar night so it should miss the real stable conditions- right?***

The fact that the DH2 did capture strongly stable conditions despite not flying in polar night shows that strongly stable conditions can happen outside of polar night (this is also evident by the seasonal SOM pattern and stability regime frequencies observed by the radiosonde, Fig. 3 and Fig. 5 of the revised manuscript). However, due to suggestions from the other reviewer, we have removed the DH2 analysis from the current paper, as it does not add much to the overall takeaways that can already be learned from the radiosonde data.

***Line 267ff: Why is a SOM based on anomalies more visual? If you anticipate the result here, I immediately wonder why you used the gradient first and did not start with the analysis of the anomaly right away – I am confused here…***

We actually did first try training the SOM with the anomaly profiles, but found that the SOM trained with the anomaly was largely failing to produce any patterns with a distinct near-neutral layer, or to highlight the distinct elevated inversion which is often present. So then we tried training the SOM with $d\theta_v/dz$ profiles, and found a much better representation of the range in height and strength of the $\theta_v$ inversion, as well as the differentiation between a

weakly stable or near-neutral layer below a $\theta_v$ inversion. We have added some text to explain this, and have also adjusted the text to state that the anomaly profile is simply another way to visualize the data, but don't claim that it is the most intuitive way to visualize the data (as what is most intuitive may differ depending on the reader).

"We also tested the SOM trained with the $\theta_v$ profiles rather than the gradient (in the form of the $\theta_v$ anomaly compared to 1 km, to remove seasonal temperature dependence), but found that the range in height and strength of the $\theta_v$ inversion, as well as the differentiation between a weakly stable or near-neutral layer below a $\theta_v$ inversion, were not distinguished." (line 284)

"… a more tangible demonstration of the range of vertical structures present during MOSAiC is shown in Fig. 2 (Sect. 3.1) with the mean profiles of $d\theta_v/dz$ and $\theta_v$ anomaly for all radiosondes mapped to a given pattern." (line 299)

***Line 279ff: I partly understand the motivation to define so many different stability regimes, but I fear that the usefulness for most readers is very limited. These 12 regimes are linked in the manuscript with 12 abbreviations that I definitely cannot remember and when these are mentioned and discussed in the text, I as a reader jump back and forth to remember the abbreviations. This disrupts the flow of reading, at least for now, whether I can do much with the information or not.***

We feel strongly that the use of all 12 regimes is important. In the Results section (e.g., with the histograms and box/whisker plots), we show discernable differences in frequencies and atmospheric characteristics both between the five different near-surface regimes (SS, MS, VSM, WS, and NN), as well as between the different aloft regimes within a certain near-surface regime (e.g., VSM-SSA, VSM-MSA, VSM-WSA). Thus, we feel is it justified to retain all of these regimes, in order to reveal important nuances in the results.

***Line 285: A possible solution for a better reading flow could be to distinguish even more clearly between methods and observations - this is only one possibility but in some places these two aspects blur a bit.***

We believe we have already well separated the description of the observations (Sect. 2.1-2.2) and the description of the methods (2.3-2.4). Whenever methods of deriving a quantity from the observations is included outside of Sect. 2.2 (e.g., the example referenced here, deriving a specific $d\theta_v/dz$ profile for stability regime identification) this has been done intentionally, because we think it would be confusing for a reader to follow what we did and for what purpose, if it were placed elsewhere. Thus, we choose not to reorganize the separation of observations and methods. However, if we have misunderstood your comment, please let us know.

***line 286ff: Why Antarctica? I think MOSAiC should really be sufficient and citing a nonpublished paper from the other side of the world does not really help here…***

The point is to demonstrate that the methods are robust across multiple polar locations (i.e., in Antarctica as well as the Arctic), to support the stability regime criteria which are new to this study. We clarify this by now saying:

"The stability regime definitions were developed alongside a similar SOM-based analysis of ABL profiles in Antarctica (Dice et al., submitted), which supports the robustness of these methods for classifying stability in polar regions." (line 317)

We hope by the time of publication of the current manuscript, the Antarctica paper will be in pre-print, but if it is not, we understand that we will need to remove the citation.

***line 290ff: Why is the gradient in 42.5 m representative for the AGL? I understand that this value might be representative for the surface layer (at least in summer) but the entire AGL - or do I misunderstand? Please clarify.***

Sometimes stability in the lowest ~10 m can differ from that in the rest of the ABL (e.g., there might be a very shallow well-mixed layer below 10 m due to enhanced mixing from the interaction between wind and surface

roughness, but above 10 m the atmosphere is stable) but we find that the stability at 42.5 m is high enough to be representative of stability throughout the majority of the ABL. We have added clarifying text in a few places:

"Twelve stability regimes have been defined based on stability within the ABL (hereafter referred to as "near-surface" stability)…" (line 311)

"Since the stability criteria in part depend on stability within the ABL and some observations have an ABL height as low as 50 m, we first include a measurement of $d\theta_v/dz$ at 42.5 m (this determines the near-surface stability), calculated across a 15 m interval between 35 m (lowest point of the profile) and 50 m." (line 320)

***Line 325ff: Why are you defining possible regimes that were never observed in the data from MOSAiC? Maybe you have some good reasons but just reading this sentence confuses me.***

We have clarified the reasoning:

"VSM-WSA and WS are not represented by a SOM pattern, but do occur rarely in individual profiles, and thus are still defined in Table 2 (see Sect. 3.2 onward). While NN was never observed in an individual MOSAiC profile, we include its definition in Table 2 to support the use of these criteria for observations from other campaigns." (line 361)

***Fig 3.: This figure (Fig. 3) contains a lot of information, and I suspect that most readers will have difficulty understanding all the lines and what they mean. Perhaps there is a way to make the diagrams a little clearer.***

As described in a repones to a previous comment, Fig. 3 in the original manuscript (Fig. 2 in the revised manuscript) has been revised to improve clarity through the following steps:
- Removed the SOM profile – this is shown in what is now Supplementary Fig. S1, and it is not necessary for the discussion of this figure
- Made the font of the numbers and letters in each subplot bigger
- Made the axes fonts bigger
- Added a "subplot key" which demonstrates clearly what is shown in each subplot, so that a reader can refer to it, rather than digging through the figure caption to figure out what they are looking at.

***The colored frame lines describing the regimes should be somewhat thicker to better distinguish the different regimes***

We have made the colored frame lines thicker (see Fig. 2 in the revised manuscript)

***Also, at this point I wonder how the ABL height is defined! For example, in pattern 8 it is quite difficult to estimate an AGL height even by eye. I assume that the inversion and the entrainment layer are not part of the AGL height according to your definition, right? Is then the term "mixed layer" more appropriate compared to AGL height? You should at least define the phrases carefully at a prominent place.***

In addition to a description of how ABL height is calculated (as was already included in the original manuscript), at the same point in the text we have also added a description of what that means physically, in terms of what is included in the ABL per our method:

"ABL height from each radiosonde profile was determined using a bulk Richardson number ($Ri_b$) based approach in which the top of the ABL was identified as the first altitude in which $Ri_b$ exceeds a critical value of 0.5 and remains above the critical value for at least 20 consecutive meters (Jozef et al., 2022). These criteria typically identify the ABL height as the bottom of the elevated virtual potential temperature ($\theta_v$) inversion (or the bottom of the layer of enhanced $\theta_v$ inversion strength) for moderately stable to near-neutral conditions, and at the top of the most stable layer for conditions with a strong surface-based $\theta_v$ inversion. The methodology for calculating the $Ri_b$ profile used to identify

ABL height, as well as justification for the use of 0.5 as a critical value (rather than the more traditional value of 0.25) is described in Jozef et al. (2022)." (line 213)

*Line 358ff: why "perhaps" - you should have the data to evaluate this "unique processes"!*

We have added some solid evidence for our hypothesis:

"Pattern 4 is particularly interesting, as there is strong near-surface stability and an elevated region of enhanced stability around 600 m AGL, which may be explained by unique processes occurring primarily in summer. Reported visibility and ceilometer observations suggest a possible low fog layer and additional elevated cloud layer." (line 407).

*Line 367: I am not convinced that the Arctic ABL is "always" stably stratified" - in particular in summer this is definitively not the case (see Tjernström et al.) So, I think to sell this as a "new finding" is going too far.*

We agree that the Arctic ABL is not "always stably stratified," and were attempting to argue the opposite point – that near-neutral is also common. However, we recognize through further literature review that this is already a notion demonstrated in prior work. Thus, we no longer present this as a new finding, and also have moved this conclusion from the results section to a separate discussion/conclusions section, per another one of your recommendations.

"The SOM patterns (Fig. 2), frequency distribution of stability (Fig. 5a), and ABL height variability (Fig. 5b) highlight that near-surface stability during MOSAiC spanned from strongly stable with a shallow ABL to near-neutral with a deep ABL, with stable and near-neutral conditions occurring with similar frequencies. Stability aloft ranged from strongly to weakly stable. These findings are consistent with Persson et al. (2002), Tjernström and Graversen (2009), and Brooks et al. (2017)." (line 568).

*Fig 4: most of the numbers are more or less invisible, at least the numbers in the upper line. Furthermore, black labels on a dark background are quite challenging. I think you should find a much better way to illustrate your point here*

We have made many adjustments to improve Fig. 4 in the original manuscript (Fig. 3 in the revised manuscript):
- Removed the number in the upper center of each box which indicated the number of radiosonde profiles mapped to that SOM pattern. This information is included in Fig. 3 in the original manuscript (Fig. 2 in the revised manuscript), so is not needed here.
- Removed the number in the center of each box which indicated the number of cases in that pattern/season. We only retain the percentage, as this is the more valuable number for understanding the results
- Made the font of the numbers and letters in each box bigger.
- Added an opaque white box behind the percentage written in each box, so it can be better seen regardless of the shading in each box
- Made the seasonal subplots bigger
- We also include now the annual frequency subplot in this same figure, in response to your next comment (we discuss this figure in the text so you suggest this not be in the supplement).

*Line 377: Again, I am not in favor of discussing material that is not in the manuscript but has been moved to a supplement. Regardless, I don't quite understand the following statement; DH2 couldn't fly in clouds - right? How then can DH2 observations cover all patterns when clouds affect stratification so much - or have I misunderstood something?*

The DH2 can fly when there are clouds, but it is not supposed to fly into the actual cloud (it can go all the way to cloud base). This allowed the DH2 to observe the cloud-forced stratification scenarios. However, due to suggestions from the other reviewer, we have removed the DH2 analysis from the current paper.

*Line 383ff: Do I understand correctly that LLJs are to be analyzed based on mean values within one individual pattern? As you have already noticed, this must - at least partially - average out the LLJ's with having their wind maximum at different heights - right? Then I have serious doubts whether this is the right method and whether one can make statements about LLJs in this way. However, it might be a misunderstanding on my side but then please clarify and consider rephrasing. In addition, there are from my point of view quite valid remarks of Günther Heinemann about the quality and (in)accuracy of wind measurements in the low altitude and the low-pass filtering of the wind data. Please take these comments seriously.*

There may perhaps be a misunderstanding here. While we show average wind speed profiles, the LLJ characteristics (height and speed) are taken as the average of these LLJ characteristics identified from each individual profile in a given SOM pattern. Thus, while the LLJ is not necessarily visually evident in the average wind speed profiles due to slight differences in the heights of the LLJs in individual cases, it is still reasonable to analyze the LLJ characteristics as a mean of those from the individual cases. Additionally, we include the interquartile ranges (IQRs) to show the spread in the LLJ characteristics throughout all of the cases in each SOM pattern. We have revised the discussion to more clearly explain our methods, while more heavily noting what we learn when considering the IQRs of the LLJ characteristics. For example, on the right half of the SOM, the IQRs for LLJ height largely don't overlap with the IQRs for ABL height, whereas on the left half of the SOM, they do, which demonstrates our conclusion that for patterns on the right half of the SOM, the LLJ is largely decoupled from the ABL and thus does not weaken the near-surface stability (and vice versa for the left half of the SOM).

"To understand the influence of mechanical mixing on the stability structures presented by the SOM, we visualize average wind speed profiles for each SOM pattern; additionally, we analyze the LLJ characteristics for each pattern, as the average across all individual cases in each pattern (Fig. 4). As LLJ core height and speed varies across the cases in each pattern, the LLJ is often smoothed out in the average wind speed profile. Interestingly, the average LLJ height was found to be similar across all SOM patterns (roughly 400 m AGL). The higher ABL heights of the weaker stability patterns (WS and NN; on the left side of the SOM) place the LLJ closer to the ABL top than for the stronger stability patterns with lower ABL heights (SS, MS and VSM; on the right side of the SOM). Additionally, the interquartile ranges (IQR) of ABL height and LLJ height overlap for all patterns on the left half of the SOM, and for many patterns, the IQR of LLJ height extends below the average ABL height. Conversely, on the right half of the SOM, the IQR of ABL height and LLJ height only overlap for pattern 23. The LLJ speeds, 2 m wind speeds, and overall wind speed profiles have greater values for the patterns on the left half of the SOM (mean LLJ speed of 12.3 m s$^{-1}$ and mean 2 m wind speed of 5.3 m s$^{-1}$), compared to the right half (mean LLJ speed of 9.7 m s$^{-1}$ and mean 2 m wind speed of 3.3 m s$^{-1}$). The LLJ frequency for all SOM patterns is similar, showing that an LLJ was present for 67% – 84% of all observations mapped to any given pattern, with a median LLJ frequency of 76%." (line 418-431)

"Results regarding LLJ height, specifically its relationship to ABL height, support the notion that both baroclinicity and inertial oscillations contribute to LLJ formation in the Arctic. For the SS, MS, and the VSM regimes (represented by patterns on the right half of the SOM), the LLJ core was situated a greater distance above the ABL than for the WS and NN regimes (represented by patterns on the left half of the SOM). This greater distance suggests decoupling between the relatively stable ABL and the LLJ, which is consistent with inertial oscillations as an LLJ formation mechanism. The smaller distance between the ABL and LLJ core for the weaker stability regimes suggests greater coupling between the well-mixed ABL and the LLJ, so inertial oscillations are unlikely to be the formation mechanism, and rather baroclinicity is the more probable cause. The results show that such LLJs have faster speeds, in agreement with Jakobson et al. (2013). The similarity in LLJ core height despite varying stability occurs because of these two different formation mechanisms. Thus, an LLJ can be both a cause and an effect of stability. For a well-mixed or weakly stable ABL, LLJs contribute to the creation of the mechanical turbulence that mixes the ABL. For more strongly stable ABLs, an LLJ can be an effect of the strong stability if the above atmosphere becomes decoupled from the surface." (line 616-628)

Additionally, we have thoroughly considered Prof. Heinemann's remarks, and have shown through various testing metrics that the wind speeds we use are reliable. Specifically relating to your concern, we applied a few different vertical averaging ranges to the wind speed data (we tried a 30 m and 60 m running average), and reproduced the LLJ frequency and box/whisker plots, to see if the results differed from when the original wind profiles were used.

We found the results to be largely unchanged, and thus conclude that we are sufficiently confident in the original wind speeds given in the level 2 soundings (used in this study), which had already undergone quite some vertical filtering/smoothing when Vaisala processed them, prior to their publication (see our responses to Prof. Heinemann's comments for more details).

***Line 388: Physically, I don't understand this point: a LLJ is by definition linked to a more stable ABL because one precondition for a LLJ is the almost vanishing turbulent exchange coefficient - right? So how can you say that "Thus the LLJ is more closely coupled to the ABL in the weak stability cases…" Maybe I don't understand your point here but then you should clarify it.***

Our response to a previous comment of yours is again relevant for this comment:

There are two primary situations in which an LLJ will occur in the Arctic. One such situation is the one you describe: the LLJ forms due to inertial oscillations and is decoupled from the (typically shallow and stable) ABL, such that wind shear from the LLJ does not weaken stability. The other situation is when an LLJ forms due to a baroclinic environment, where wind speeds decrease with altitude according to the thermal wind relationship, and surface friction reduces wind speeds to 0 at the surface, this resulting in an LLJ some distance above the surface. In this case the LLJ is not decoupled from the ABL, and can contribute to weakening of stability.

We have added more discussion in the introduction to clarify these two situations under which an LLJ may form, and the subsequent relationships with ABL stability.

[revised manuscript text omitted]

We have revised the Fig. 5 in the original manuscript (Fig. 4 in the revised manuscript) to be make it more clear to understand:
- Made the font of the numbers and letters in each subplot bigger
- Made the axes fonts bigger
- Removed the number in the upper center of each box which indicated the number of radiosonde profiles mapped to that SOM pattern. This information is included in Fig. 3 in the original manuscript (Fig. 2 in the revised manuscript), so is not needed here.
- Added a "subplot key" which demonstrates clearly what is shown in each subplot, so that a reader can refer to it, rather than digging through the figure description to figure out what they are looking at.
- Due to the above bullet point, we have removed "LLJ:" and "2m" from each subplot, as the "subplot key" now indicates what these numbers mean.

***Line 415 ff: The is probably one of the most prominent places to say: I am lost in all the details and even more I lost track due to the huge number of abbreviations….***

We have restructured how we share the stability regime frequency, focusing on the individual near-surface regimes first, and then talking about aloft regimes more broadly. We hope this is a more intuitive and interesting way for a reader to digest these results:

"The most frequent near-surface regime observed was NN (37% of profiles), followed by VSM (27% of profiles), MS (14% of profiles), and SS (13% of profiles) in decreasing order. WS was observed least frequently (9% of profiles). The total frequency of a stable ABL (combining SS, MS, and WS frequencies) was 36%, just slightly less than the frequency of a near-neutral ABL. The most frequent regime observed aloft was -SSA (66% of VSM cases, 54% of WS cases, and 60% of NN cases had strong stability aloft) followed by -MSA (31% of VSM cases, 39% of WS cases, and 35% of NN cases had moderate stability aloft). Weak stability aloft was infrequently observed (3% of VSM cases, 7% of WS cases, and 5% of NN cases had weak stability aloft). The overall most common regime was NN-SSA, followed by VSM-SSA." (line 457)

***line 454: If a parameter x changes over the depth of the ABL it is not the same as the gradient dx/dz – right?***

We have adjusted the wording to more clearly state what we mean. Also, we now only include dV/dz in the revised manuscript, as the $d\theta_v/dz$ result was largely a function of how we defined the stability regimes.

"…change in horizontal wind speed between the surface and top of the ABL (dV/dz)…" (line 471)

***Line 455ff: If you cite a figure within the main text, it should be included in the main text - why has it been shifted to a supplement?***

It is relatively common practice to include some figures in a supplement which are mentioned in the main text but are not crucial to the understanding of the paper, and rather supply additional details to an interested reader. We have moved most of the figures originally included in the supplement in the original submission to the main text of the revised manuscript, per your suggestion, but we choose to keep the statistical significance figures in the supplement. As the current paper is already quite long and complex, we do not think that adding the statistical

significance figures to the main text will help with improve the readability of the paper. The main takeaways of the study can largely be discerned without them, though we do still provide a brief discussion of them in the main text.

*line 463: "...the fact that we see this drastic increase also supports the choice of this threshold...." can you please explain this in a little bit more detail?*

We have modified the sentence to better state our point:

"The jump in ABL height between the VSM and WS regimes is in part a product of how we define the VSM regime (which requires an ABL height of 125 m or less). However, the magnitude of the increase in ABL height between the VSM regime (mean of 85 m) and WS regimes (mean of 221 m) demonstrates that this threshold was meaningful." (line 478)

*line 465ff: I cannot follow this sentence at all - please double check and consider rephrasing.*

We have modified the sentence to clarify our point while not including unnecessary information:

"Additionally, we find that ABL height increases as stability aloft decreases (e.g., the mean ABL height for WS-MSA is greater than the mean ABL height for WS-SSA)." (line 481)

*line 468ff: It is quite unusual to start a sentence with a equation - I would avoid it. Furthermore: "shear" => "wind shear".*

We have made sure to not start a sentence with an equation, and have changed "shear" to "wind shear":

"SS and MS had the greatest (largely above average) wind shear (dV/dz) within the ABL (Fig. 5c)." (line 483)

*So why do you mention Ri with index "b" (for bulk?!) here? If you mention the (local?) gradients then you have the basics for the classical local Ri definition - right?*

This is a good point. We have, however, removed the discussion of Richardson number as part of our adjustment of the content to only include the most interesting and new results.

*line 475ff: again, a reference outside the paper is not helpful and I suggest to avoiding this. Furthermore, I do not understand your conclusion about the physically meaningful definition of the regimes - please specify what you mean here.*

As mentioned in a response to a previous comment, we choose to keep the statistical significance figures in the Supplement, as the statement of what can be gleaned from these figures is sufficiently meaningful, while an interested reader may find additional details in the supplementary figure if they would like.

Next, we have tried to clarify what we mean about the physically meaningful definitions of the regimes, which we have merged with the similar discussion with regards to u*:

"Significant differences in dV/dz and u* between most pairs of stability regimes (Fig. S2b) highlights that turbulence properties are distinct for each regime. While perhaps an intuitive statement, it is important to confirm that physically meaningful differences in stability regimes classified largely based on thermal gradient are found for mechanical processes, as well as for turbulence measured by the met tower (a separate platform than the radiosondes used to classify stability regime). This confirmation supports the validity of the stability regime criteria defined in Sect. 2.4." (line 487)

*line 477: what is exactly meant by "dV/dz result" - from my point of view this makes no sense*

We have clarified this sentence:

"For the weaker stability regimes (WS and NN), winds vary less with height due to greater mixing, which is a common behavior of winds within a weakly stable or near-neutral ABL (Wallace and Hobbs, 2006)." (line 483)

*line 479ff: It is surprising that you start the discussion with the Richardson number and now you move on to the friction velocity - so why? Furthermore, you correctly mentioned that Ri describes more the tendency of turbulence development but it is not a measure of the degree of turbulence or the intensity but you concluded already that the near-surface atmosphere is always turbulent. From my point of view this is going too far. This part also needs some careful reconsiderations and not only rephrasing.*

As mentioned previously, we have removed discussion of the Richardson number from the manuscript, and thus do not discuss the implication of the Richardson number for turbulence. We instead focus the discussion on $u_*$.

*line 487: Well, turbulence itself might describe a flow but probably not the Arctic - this makes no sense. Maybe I missed it but I suggest to define u_star at the place of first occurrence (around line 145 or so). Also, I suggest to use the word „increased" turbulence.*

We have removed the statement that the Arctic environment is characterized by turbulence. Also, we have added a more in-depth description of bulk $u_*$ used in this study when it was first introduced:

"Atmospheric observations of … bulk friction velocity (a theoretical wind speed that expresses the magnitude of stress exerted by wind flowing over the Earth's surface, indicating the magnitude of turbulence; $u_*$), come from a 10 m meteorological tower… and provide information about near-surface turbulence at the time of each radiosonde launch. Bulk $u_*$ was chosen, as opposed to the standard eddy-covariance value, as the bulk parameterization considers both wind fluctuations and latent heat fluxes (developed using guidance from eddy-covariance data collected during SHEBA; Andreas et al., 2010b) which is more comparable to $u_*$ used in models (e.g., Fairall et al., 2003)." (line 167-175)

Lastly, we have removed the statement that increased $u_*$ indicates increased turbulence here, as we have included a more thorough description of $u_*$ when it is first introduced.

*488ff: this not really surprising and I think this statement does not need a supplementary figure - right?*

We have revised the discussion to highlight the importance of sharing this unsurprising result:

"Significant differences in dV/dz and $u_*$ between most pairs of stability regimes (Fig. S2b) highlights that turbulence properties are distinct for each regime. While perhaps an intuitive statement, it is important to confirm that physically meaningful differences in stability regimes classified largely based on thermal gradient are found for mechanical processes, as well as for turbulence measured by the met tower (a separate platform than the radiosondes used to classify stability regime). This confirmation supports the validity of the stability regime criteria defined in Sect. 2.4." (line 487)

*492ff: I am not surprised that u_star and and wind shear do not show a clear dependency because it is a Richardson number problem - but is this a conclusion as you mentioned?*

We have removed the discussion on the dependence (or lack thereof) of $u_*$ and dV/dz from this section, as the revised manuscript instead includes a separate discussion section. With the separation of the "Results" and the "Discussion and conclusions" into different sections, and the removal of Richardson number results from the paper, we hope the discussion on these points is more interesting and the primary takeaways are better highlighted:

"Despite slower wind speeds and lesser $u_*$ for stronger near-surface stability, wind shear (dV/dz) over the depth of the ABL increases with increasing stability, revealing that in strong stability cases, static stability suppresses mechanically generated turbulence, promoting continued ABL stability despite high amounts of wind shear."  (line 607)

*line 495ff: I cannot follow your argumentation, in particular the last part "...when stability aloft is greater." So, is there a connection between stability at higher altitudes and surface layer mixing? Please explain what you mean here.*

This conclusion has been moved to the new, separate "Discussion and conclusions" section, as several results throughout the paper highlight the same conclusion. Additionally, we had revised the description of this conclusion to be more clear:

"While LLJ speed and $u_*$ increase with decreasing near-surface stability, the opposite relationship is seen for stability aloft: LLJ speed and $u_*$ values are greatest when stability aloft is greatest. These results suggest that when the atmosphere is inclined to be strongly stable (e.g., in the absence of clouds during winter), more mechanically generated turbulence is required to fully mix out the near-surface layer than if the atmosphere is inclined to be weakly stable (e.g., in the presence of clouds)." (line 611)

*Fig 7 (and maybe other figures as well): you put the labels of the y-axis on the x-axis which is formally not correct and took me a time to understand the plot - please change this labeling.*

This change has been made.

*line 511ff: Maybe I am wrong and I don't have the details of Banta et al in mind but how can a LLJ exists (or develop) in a well-mixed ABL? Maybe during the transition to a classical Ekman-layer like ABL this makes sense but from a theoretical point of view a LLJ and a well-mixed layer are exclusive; the turbulent exchange coefficient K_m has tend to zero to decouple the LLJ region from the surface - right? In fact, that is the background for your comment in line 513ff but I think this is a precondition for the development of a LLJ....*

We have added some text in the introduction to explain how an LLJ can exist when there is a well-mixed ABL. Thus, we believe now that the proper context to support the results presented in the LLJ section are provided.

"Another common feature of the Arctic lower atmosphere is a low-level jet (LLJ), which is a local maximum in the wind speed profile below 1.5 km (Tuononen et al., 2015) that is at least 2 m s$^{-1}$ greater than wind speed minima above and below (Stull, 1988). There are two primary forcing mechanisms for LLJs in the Arctic: baroclinicity and inertial oscillations. Baroclinicity in the Arctic most often occurs near the ice edge (Brümmer & Thiemann, 2002) or due to the passing of a transient cyclone (Jakobson et al., 2013) which creates regions of enhanced temperature contrasts (Koyama et al., 2017). Depending on the wind direction, the horizontal temperature gradient causes the geostrophic wind speed to decrease with height according to the thermal wind relationship (Stull, 1988). This, paired with diminishment of wind speeds at the surface due to friction (Stull, 1988), contributes to the formation of an LLJ at some distance above the surface, typically just above the ABL (Brümmer & Thiemann, 2002). Thus, an LLJ forced by baroclinicity is typically coupled to the surface, and can cause weakening of stability within the ABL due to enhanced shear below the jet core (Banta, 2008; Egerer et al., 2023)." (line 76-86)

*line 515ff: maybe you mentioned it earlier but is the "LLJ speed" defined as the maximum speed in the LLJ core or the difference to the surrounding?*

The LLJ speed is the maximum speed in the LLJ core. This information was stated on line 194 – 195 of the original manuscript. It can be found in the revised manuscript on line 226-228:

"If an LLJ was found, we identified the LLJ core altitude as the altitude of the maximum in the wind speed, and the LLJ speed as the wind speed at that altitude (Jakobson et al., 2013)."

*line 548ff: I think this is physically not really meaningful, maybe it should read as: "...it needs more wind shear in a more stable environment to create mixing ..." or similar. Furthermore, I think the phrase "hypotheses" is going a little bit too far because you never formulated a hypothesis which can now be verified or falsified...*

We revised this description following your suggestion to be more physically meaningful, and now include it in the separate "Discussion and conclusions" section. The "hypothesis" we were referring to was previously stated in association with the dV/dz and $u_*$ results, where more turbulence was necessary to mix out an environment with greater stability aloft. By separating the "Results" and "Discussion and conclusions", we can now make this conclusion one time, with support from the various results:

"While LLJ speed and $u_*$ increase with decreasing near-surface stability, the opposite relationship is seen for stability aloft: LLJ speed and $u_*$ values are greatest when stability aloft is greatest. These results suggest that when the atmosphere is inclined to be strongly stable (e.g., in the absence of clouds during winter), more mechanically generated turbulence is required to fully mix out the near-surface layer than if the atmosphere is inclined to be weakly stable (e.g., in the presence of clouds)." (line 611)

*line 555: what do you exactly mean with "excess turbulence"?*

The intent was to say that, even in ubiquitously high wind speed environments, a LLJ would contribute to more turbulence production than if there was not an LLJ, as the LLJ speed exceeds that of the rest of the winds throughout the column. We have clarified this wording:

"However, such LLJs can still be important because even if the wind speeds are fast throughout the entire profile up to 1.5 km (for example, during a storm), the slightly greater speed of the LLJ beyond that of the ubiquitously high winds throughout the column supports the production of increased turbulence in the ABL compared to without an LLJ." (line 634)

*Fig 8.: See my comment on Fig 7 about the axis labels*

This issue has been fixed for all figures.

*Line 567ff: About your supplement: it just includes many further figures which are partly mentioned within the main manuscript but not explained and deeply discussed. I think this is not the right way to use a supplement because the main manuscript should be readable by itself without reading the supplement. If you have a distinct and interesting topic which might be useful for some readers but distract from the red line of the manuscript a supplement might be the right choice but if you have simply too many figures which you want to mention in the manuscript you cannot just move them into the supplement and refer to them (different to an appendix).*

The reviewer makes a good point. To address this comment, we have reconsidered which figures to include in the paper, and retain only those which reveal the most interesting and/or new results, and have remove the other figures/panels.

*line 577ff: Do you consider TIs as a cause or effect of ABL development?*

As noted in previous responses, we recognize that the TI results are largely a function of how stability regime is defined, and thus the results are not the most pertinent. Thus, we have removed all figures and text related to temperature inversions to provide a more concise manuscript.

*Line 578ff: About the TI analysis: I am a little bit skeptical about this analysis because I wonder how much of this analysis is based at least partly on self-correlation because the stability regimes are based on the temperature gradients - right? Maybe this should be discussed at least a little bit before interpreting the results.*

See our response to your above comment.

*Line 581 and general: Maybe better distinguish between explaining figures and results, than interpreting them and finally compare with other studies - the structure is often a bit confusing and you jump back and force*

We have restructured the paper to address this and other comments. We now purely share results in the "Results" section, and include a separate "Discussion and conclusions" section where we discuss interpretations of the results, and comparisons to other studies.

*line 591ff: I cannot follow your argumentation about the potential for exchange of momentum when TI is well above ABL height - what do you mean? I have the feeling that here is a lot of speculation on play but a careful physical interpretation is missing.*

See our response to your previous comments regarding TIs.

*line 603: As mentioned earlier, I have the feeling that this part is based on a big portion of self-correlation which has to be ruled out before the interpretation.*

See our response to your previous comments regarding TIs.

*line 621: why using a decimal value for the second cloud base height? Why 6.1 km and not 6 km??*

The original threshold of 6.1 km was chosen, as this threshold was given by an online source. However, the thresholds for low, mid-level, and high clouds vary between sources, so in the end, choosing a threshold is a bit arbitrary. Thus, for simplicity, we have changed the mid-level cloud base height threshold to 6 km. The results are essentially unchanged.

*line 633ff: What do you mean with the statement that "… a regime is driven by the radiative signature of clouds..."? and further on in line 634 what are those other mechanisms? This explanation is hardly to follow and needs some careful rephrasing.*

We intended to mean that the VSM and NN regimes are driven by the enhanced downwelling radiation produced by clouds versus clear skies. The other mechanisms we refer to are primarily longwave cooling and wind speeds. We now clarify all these things when the interpretation of the cloud-related results are discussed in the "Discussion and conclusions" section.

"Low clouds, correlated with greater LWP, were observed with greater frequency for cases with weaker stability both within the ABL and aloft, highlighting the ability of low clouds and enhanced moisture content to support turbulent mixing both near the surface through enhanced downwelling longwave radiation, and below cloud base though cloud top radiative cooling. In such cases, a well-mixed ABL can be coupled to the cloud layer and extend through the depth of the cloud to cloud top, though a shallow stable layer may decouple a well-mixed ABL from a low cloud. Conversely, mid-level and high clouds were observed with greater frequency for cases with stronger stability, highlighting that in such cases, the cloud is likely to be decoupled from the surface, allowing the strong stability to persist." (line 644)

*The summary will certainly need to be completely revised when the previous analyses have been appropriately re-sorted and revised - so I have refrained from detailed comments on this chapter now.*

We have indeed revised the summary greatly based on the reorganization of the paper, and our original "Summary and Conclusions" section is now a "Discussion and Conclusions" section. We are open to hearing your feedback on this new section during the next round of reviews.

---

## Author Response (AR2)

**Response to Anonymous Referee Comments**

We would like to sincerely thank the two anonymous referees for taking the time to review our manuscript and for their helpful comments, which have improved the manuscript. Each referee comment is given below in **bold italics** followed by our response to the comment. The line numbers provided in our responses refer to line numbers in the revised manuscript, unless otherwise stated.

**Anonymous Referee 1**

***Comments about the revised manuscript***

***An Overview of the Vertical Structure of the Atmospheric Boundary Layer in the Central Arctic during MOSAiC***

***By***
***Gina C. Jozef, John J. Cassano, Sandro Dahlke, Mckenzie Dice, Christopher J. Cox and Gijs de Boer***

***First of all, I must say that my comments on the first version of the manuscript were addressed quite thoroughly and the revised version is much clearer and easier to read. A lot of effort has been put into the revision, but it has benefited the quality of the manuscript and is therefore worth it from my point of view. Besides a few minor points - mainly of a technical nature - I still have some concerns about the LLJ analysis which I will describe in detail below and which, in my view, need further discussion.***

***As already written in my first review: I am not a native speaker but for me some sentences and phrases still sound a bit bumpy but that is certainly a matter of taste should be judged by the native speaker.***

We have made an effort to clean up the sentences for increased clarity throughout the paper. If there are certain sentenced you still find to be bumpy, please let us know.

***I refer to the lines in the pdf of the revised manuscript version without track changes***

***General note on wording: in some places the manuscript still contains some very imprecise wording that should be better avoided, just a few examples:***

***i) A LLJ cannot be fast or slow - only the wind speed (in its core) can be fast (after line 505). In the same paragraph, sometime you just write speed, then LLJ speed - be consistent, in particular when comparing typical wind speeds of the LLJ you could introduce a parameter such as $U_{LLJ}$ or so –***

We have removed the section after line 505 based on the comments from the other reviewer. This reviewer argued that the paper was too long, and the information in the LLJs and cloud sections at the end did not add great value for the results. Thus, we removed these sections and rather focus on presenting the important results relating to LLJs and clouds in relation to the SOM. This includes adding a SOM figure which presents moisture and cloud properties. We have still, however, checked throughout the paper to ensure we always refer to the speed of the LLJ (rather than the LLJ) as fast or slow, as well as revised the text to be consistent between saying 'LLJ speed' when referring to the speed of the LLJ. Whenever we refer to simply 'speed' this is because we are not specifically talking about the LLJ in that case.

***or***

***ii) If you mean the "wind speed" than do not just write "speed" (see line 22)***

This has been fixed here and throughout.

***Line 46: what do you mean with flux perturbations? Please define!***

The original phrasing was taken from Lesins et al. (2012), but we have clarified this, and now say:

"… by dynamically decoupling the surface from the free atmosphere, so that lower atmospheric warming related to increased surface to air (or decreased air to surface) heat fluxes cannot easily spread through the troposphere, and warming is concentrated near the surface (Lesins et al., 2012)." (line 46)

*Line 58: the argument about ozone destruction is probably not wrong but a little out of the context – I would skip it*

We have removed this sentence.

*Line 61: I think decoupling is frequently observed but "typically" is probably going too far as there are have been also coupled ABLs observed*

We now say "The ABL is often decoupled from the cloud layer by a shallow stable layer…" (line 71)

*Line 86: If I interpret Egerer et al 2023 correctly, they argued that turbulence below the LLJ can be created by shear only during the collapsing phase of the LLJ and decoupling is one of the prerequisites of a LLJ (turbulent exchange coefficient tends to zero).*

We do not fully agree with your interpretation of Egerer et al. (2023). Egerer et al. (2023) shows that there is a greater production of turbulence closer to the jet core, but this increased turbulence does not extend to the surface. That does not mean that no turbulence reaches the surface. Thus, wind shear associated with an LLJ can still contribute to turbulence near the surface, just not with as great a magnitude as closer to the jet core. Several other studies (Mahrt, 2002; Mäkirantaetal., 2011) rather argue that the wind shear below the core of an LLJ may be the main source of turbulence for a stable boundary layer (and are one of multiple sources of turbulence for a neutral or unstable boundary layer). Whether the LLJ is coupled or decoupled to the surface does not necessarily dictate whether the wind shear associated with the LLJ can contribute to turbulence in the ABL. To avoid confusion, we have removed the Egerer et al. (2023) citation and added the other references instead:

"Due to enhanced wind shear and subsequent turbulent kinetic energy production, an LLJ can contribute to mechanically generated turbulence below the jet core (Banta, 2008; Mahrt, 2002; Mäkiranta et al., 2011)." (line 79)

*Line 109ff: why are the conclusions of the two references so different? (Although I think there should be better references than a textbook contribution). Are they both observations at the same time of year? I think if you are going to make such a strong comment, you should elaborate on this a bit more.*

We have added some additional information which may explain the difference between the two results:

"For example, Esau and Sorokina (2010) claims that the central Arctic ABL is stable 70-90% of the time based on lower resolution observational and reanalysis data, while Tjernström and Graversen (2009) found stable and near-neutral conditions to occur with similar frequencies based on higher resolution observations from SHEBA in the Beaufort gyre." (line 96).

*Line 132ff: It is hard for me to follow your argumentation: If you want to test if the results of MOSAiC agree with previous observations you should use the same method (SOM) with older data sets – right? There is probably nothing "wrong" with the past observations and only the conclusions drawn from the observations can agree or not. So, I am a little bit confused about your argumentation here – maybe it is only the wording.*

We have decided to remove this sentence, as the real argument for the benefit of this study is that it adds to the body of literature on ABL vertical structure in the Arctic and includes measurements to fill gaps in our knowledge, through the use of new methodologies. Thus, we now jump right into these points:

"A self-organizing map (SOM) analysis (which objectively identifies a user-selected number of patterns present in a training data set) was conducted with the radiosonde profiles to reveal the range of vertical structures observed during MOSAiC (differentiated by stability within the ABL and the height and strength of a capping inversion), their relative frequencies, and their correlation to wind and moisture features during the MOSAiC year. The SOM results also were used to develop criteria to define stability regimes characterized by stability both within and above the ABL, such that

features related to stability can be analyzed both in the context of the SOM patterns, as well as a more simplified grouping of observations by stability. Through the use of these new methods (i.e., the SOM analysis and detailed stability regime classification), the results provide further constraints on the vertical structure and features of the Arctic lower atmosphere that may be helpful to improve parameterizations of the central Arctic in weather and climate models." (line 116)

***Line 156: such a reference should be placed in the bibliography and not inside the main text, maybe a matter of taste.***

We no longer include the link in the text, but only in the full reference in the bibliography, while simply providing the in-text citation at this point:

"Instrument specifications and uncertainties for the radiosonde variables are provided in the manufacturer data sheet for the Vaisala Radiosonde RS41-SGP (2017), and are summarized in Table 1." (line 143)

***Line 159: additional averaging can reduce noise but not uncertainties – please consider rephrasing and I think you should change "winds" to "wind speeds" or similar. I still have mixed feelings about this issue. All the problems related to wind speed determination, especially in the lowest 100 m or so, cannot be solved simply by vertical averaging.***

We have changed 'winds' to 'wind speed and direction', clarified that vertical averaging only reduces noise but not uncertainties, and have added some more details as to why we are confident in the radiosonde wind speeds, even below 100 m:

"It is recognized that the true uncertainties in the wind speed and direction are likely to be greater than those provided in the data sheet, however for the following reasons, we find the original winds provided in Maturilli et al. (2021) to be sufficiently reliable for the current study. First, we determined that our results changed minimally when additional vertical averaging was applied to the winds (beyond the filtering already applied by Vaisala during their data processing), and thus noise in the observations does not bias the results. Second, when comparing radiosonde wind speeds to those measured by the DataHawk2 uncrewed aircraft system which observed the atmosphere during MOSAiC between 5 m and 1 km, Hamilton et al. (2022) found a difference of less than 1 m s$^{-1}$ based on the 95% confidence intervals of observations from both platforms." (line 145).

***What exactly convinces you that wind measurements above 35 m are trustworthy? I have my doubts and have heard several comments that radiosondes should not be trusted too much for wind measurements below 100 meters. I would at least make a clear comment that this problem exists and one should be somewhat careful with the interpretation.***

In response to your above comment, we share a reason why we are confident in the wind measurements above 35 m:

"Second, when comparing radiosonde wind speeds to those measured by the DataHawk2 uncrewed aircraft system which observed the atmosphere during MOSAiC between 5 m and 1 km, Hamilton et al. (2022) found a difference of less than 1 m s$^{-1}$ based on the 95% confidence intervals of observations from both platforms." (line 150).

Nonetheless, we have added a note that:

"Nonetheless, caution should be taken with interpretation of radiosonde wind speeds in the lowest 100 m." (line 152).

***Line 168ff: The explanation of the bulk velocity is a bit misleading and needs a more precise scientific description including the equation to make sure that we all mean the same thing. You have the information from the ultrasonic anemometer to calculate u\* based on co-variances how it is developed from theory so what's the problem with this calculation and why do you use a bulk estimate and how does the latent heat flux contribute to the friction velocity? Maybe I have overseen it in Andreas et al., 2010b but he uses in Eq. 2.1a the drag***

*coefficient and a mean wind velocity to estimate the friction velocity. For me it is not clear what you are exactly doing here and why? The argument that another method is frequently used by modelers is not really convincing.*

We have decided to switch from using the bulk parameterization of $u_*$ to the standard eddy-covariance value, calculated using sonic anemometer measurements at 10 m. Therefore, we have removed discussion of bulk $u_*$ from the paper.

"Atmospheric observations of wind speed at 2 m above the surface, as well as friction velocity (a measure of the vertical fluxes of zonal and meridional horizontal momentum, suggesting the magnitude of mechanically generated turbulence; $u_*$) measured at 10 m, come from a 10 m meteorological tower (hereafter called the met tower) located on the sea ice near the *Polarstern* (Cox et al., 2023a,b). These measurements provide information about near-surface turbulence at the time of each radiosonde launch. Derivation of $u_*$ through standard eddy-covariance methodology, and corresponding uncertainties, follow Persson et al. (2002)." (line 159)

*Line 170: maybe you could combine bot Cox citations (Cox, et al., 2023 a,b) or so ?!*

As the Cox et al., accepted paper has now been published, we have changed this to Cox et al., 2023b, and have changed the other Cox et al., 2023 citation to Cox et al., 2023a. We have also revised the sentence to read:

"… come from a 10 m meteorological tower (hereafter called the met tower) located on the sea ice near the *Polarstern* (Cox et al., 2023a,b)" (line 162)

*Line 176: I think the ceilometer derives cloud base height from backscatter – right? From your wording it sounds like it measures both quantities (backscatter and cloud base height).*

You are correct, and thank you for pointing this out. We have clarified the sentence to read:

"… which derives cloud base height (CBH) from measured atmospheric backscatter…" (line 166)

*Line 185: please include in the table caption that these numbers are provided by the manufacturer (which might differ from the reality…) to make this clear.*

The caption now reads:

"Instrument name and uncertainty for each variable used in this study, as provided by the manufacturer (real uncertainties may differ from those listed)." (line 175)

*Line 338: "We make this distinction because we there are different processes that would lead to a shallow versus deep well-mixed layer." - The sentence does not make sense, maybe just delete "we" ?*

This was a typo, so thank you for pointing this out. The sentence now reads:

"We make this distinction because there are different processes that would lead to a shallow versus deep mixed layer, which would be better highlighted by differentiating such categories." (line 344)

*Figure 2: Still, the labels and numbers are partly quite small; in particular the numbers for the altitude in Fig 2 should be enlarged; same for the labels for the vertical lines in the subplot key of Fig 2. However, it is really appreciated that a subplot key is included to explain the complex figures and the figure quality has improved a lot.*

We are glad to hear that adding the subplot key is useful. We have enlarged the vertical and horizontal axes fonts, as well as the font for the labels of the vertical and horizontal lines in the subplot key.

*And the same for Fig 3: the upper line in the subplot describing the "Annual" is hardly readable but for the four seasonal cases it is still very tiny.*

We have enlarged all of the fonts in Figure 3, and hope the figure is now readable.

*Line 536: It is obvious what you want to express here but your wording could be more precise: What cloud property can lead to de-coupling of the sub-cloud layer (better than "below-cloud" layer) from the atmosphere above? This statement is quite vague…*

We have changed "below-cloud layer" to "sub-cloud layer" (line 458). We have also added some additional information about cloud properties leading to decoupling:

"Properties of clouds and moisture can impact vertical $\theta_v$ structure and stability due to their radiative effect and ability to decouple sub-cloud layers from the atmosphere above (e.g., clouds which form from long-range moisture transport are often separated from the ABL by a stable layer, such that turbulence is not continuous between the ABL and the cloud)." (line 457)

*I'm still having some trouble following this line of reasoning: What kind of relationships between mechanical mixing and atmospheric stability have you uncovered? It is probably true that around a LLJ - if the temperature gradient is not too strong - there is shear-induced turbulent mixing, but you have not measured it, so all very speculative. I am not convinced that friction velocity is generally an appropriate measure of potential mixing around the LLJ, or have I misunderstood the reasoning?*

Perhaps a line number was missing from your comment, and we are not entirely sure which section you are referring to with your comment about the relationship between mechanical mixing and atmospheric stability. Were you referring to lines 599-601 in the previously-submitted manuscript? We will address this comment given this assumption. First, we have changed the word "reveal" to "suggest" as much of the results are inference, based on the fact that wind shear contributes to mechanical mixing. Additionally, we were not intending to say that friction velocity tells us about potential mixing around the LLJ. Friction velocity and the LLJ are two separate characteristics of the atmosphere which both separately suggest the potential for mechanical generation of turbulence, where friction velocity tells us about mechanical turbulence near the surface, and the presence of an LLJ tells us about mechanical turbulence at higher altitudes. The point of this sentence was to list all of the variables we looked at for inferring the mechanical generation of turbulence. We have attempted to clarify this:

"In the following discussion, we first summarize the relationships between wind speed features and atmospheric stability. Average wind speed and LLJ characteristics for each SOM pattern (Fig. 4), and wind shear and $u_*$ within the ABL (Fig. 6c-d) suggest important relationships between mechanical mixing and atmospheric stability and vertical structure. Wind shear and $u_*$ within the ABL quantify mechanical turbulence near the surface, while the presence of an LLJ can enhance mechanical generation of turbulence aloft." (line 593)

*About the LLJ discussion:*

*In the revised version, you explain the two different ways an LLJ can form, but their characteristics and causes are quite different, and you have no way to analyze exactly what kind of LLJ you observed. Also, for the SOM analysis it is not possible to distinguish between the two LLJ variants although they have very different characteristics or are actually two completely different phenomena. On the other hand, this could be important for the interpretation of a possible correlation between the friction velocity and the properties of the LLJ.*

*However, from my point of view the friction velocity and the LLJ have not necessarily to do with each other - especially not for the LLJ which can be described by an inertial oscillation. The friction velocity only describes the momentum transfer from the surface to the lower atmosphere, and probably a low friction velocity is a prerequisite for the occurrence of one LLJ variant to minimize the turbulent vertical energy exchange (for the "inertial oscillation LLJ"). Therefore, I also have some concerns about the described situation and interpretation with stronger LLJs (faster wind speeds within the LLJ combined with higher friction speed) in a near neutral ABL, as you mention in line 603.*

*If a LLJ is caused by baroclinicity and is directly above the ABL according to your reasoning, how can this LLJ be coupled to the surface since most ABLs are covered by at least a weak inversion? By the way, to which section in Brümmer and Thiemann are you referring exactly?*

***The detailed description of the two LLJ types helps in the text, but what consequences it has for the analysis is just not clear to me yet. I think that this point of the manuscript needs some deeper discussion and especially the differences of the two LLJ types and in particular their consequence for the analysis needs to be better discussed.***

You make a good point that with the data we use, we really have no way of definitively knowing whether an LLJ was formed by baroclinicity or inertial oscillations, so our discussion of such things is only speculative. Therefore, based on your comments here, as well as comments from the other reviewer, we have chosen to eliminate discussion of these two LLJ formation mechanisms from the introduction and results. Ultimately, the main conclusions we want a reader to draw from this paper are independent of the LLJ formation mechanism. With such edits, we have removed the Brümmer and Thiemann citation. Next, it seems one of your greatest concerns has to do with our discussion of the relationship between LLJ features and friction velocity. However, we did not intend to argue that the LLJ and friction velocity have anything to do with each other. We were simply trying to share that LLJ speed and stability have a similar relationship to each other as do friction velocity and stability. We hope by adding the following sentences, this is now clear:

"In the following discussion, we first summarize the relationships between wind speed features and atmospheric stability. Average wind speed and LLJ characteristics for each SOM pattern (Fig. 4), and wind shear and $u_*$ within the ABL (Fig. 6c-d) suggest important relationships between mechanical mixing and atmospheric stability and vertical structure. Wind shear and $u_*$ within the ABL quantify mechanical turbulence near the surface, while the presence of an LLJ can enhance mechanical generation of turbulence aloft." (line 593)"

***Line 612: "These results suggest…" is this a surprising result? By the way, the argumentation is somewhat misleading: in a stable environment you need more wind shear to develop turbulence and finally mix the ABL – right? The amount of turbulence to mix the ABL is the same – independent of stability. It is a classical Richardson problem.***

We have revised the sentence to better clarify our point:

"One possible explanation is that when the atmosphere is initially strongly stable (e.g., in the absence of clouds during winter), more wind shear is required to produce enough mechanically generated turbulence to fully mix out the near-surface layer than if the atmosphere is initially weakly stable (e.g., in the presence of clouds). Then the stable layer becomes elevated, separated from the surface by the mixed layer." (line 609)

***Line 652: It sounds to me like you consider a "storm" and an LLJ to be equivalent? Maybe it is more about the wording but please clarify!***

Based on comments from the other reviewer, we have removed the figure which shows LLJ frequency depending on the season, and thus have removed this paragraph from the Discussion and Conclusions section.

**Anonymous Referee 2**

***This is a revised version of a paper dealing with an extensive analysis of the vertical structure of the Arctic lower troposphere, and especially the boundary layer (BL), from the MOSAiC year-long field campaign. Let me start by saying that the revised version is a clear improvement to the original manuscript and there is a lot of interesting new information.***

***That being said it is still not a great paper and this is sad because it could be a great paper. So, at this juncture the choice is whether to accept an extensive paper that has several flaws, in which case some revisions are still necessary, or if yet another major revision is required to make this the really great paper it could become. I will recommend major revision, because I want to maintain a high standard, especially when it is possible, but ultimately this is an editorial decision.***

***Major concerns***

***The first thing that becomes obvious is that the extensive scope of the paper is a problem. For one it is quite long;***

*close to 40 pages including figures and references (although the review manuscript is longer than necessary depending on how the figures are set). There are three reasons for this: 1) A quite long introduction and methods section; 2) that the manuscript is actually two studies merged together, and; 3) the inclusion of some other related data to the analysis of vertical structure (low-level jets (LLJs) & clouds) at the end.*

We have made an effort to significantly reduce the length of the paper based on your comments. 1) In the originally submitted version of this manuscript, the introduction and methods sections were a bit shorter, but previous reviewer comments were asking for more details from prior literature on Arctic ABL stability and structure, as well as about the SOM methodology. Thus, in response to those comments, we added additional content to both the introduction and methods sections. Therefore, we do not want to remove too many details at this point, to remain consistent with the previous reviewer comments. However, we have shortened the introduction by about half a page and have worked to reduce the length of the methods section throughout, where possible. 2-3) We have removed the LLJ and clouds sections at the end, and instead have added some additional content/discussion regarding the SOM, so that the paper comes across as more of just one study, and is not including unnecessary content at the end. The amount of content added is less than that removed, so ultimately this reduces the length of the paper.

*The introduction (~3 pages) reviews almost anything that one could think of being done previously on the topic and is written quite long. It also cover topics that are not at the center of this paper, such as decoupling, and I'm, sure it could be shortened by 30%. There is a 12-line paragraph on page 2 that deals with decoupling, which is important – indeed suggested even to be "typical" (I would have settled for common)"- however, the methods used in this paper makes it impossible to separate out decoupled BL clouds in this study. So even if it is indicated to be important in the introduction, there is no feed-back to this from the results in the study, nor are there any comments on this. By using a bulk-Richardson approach to determine the BL depth only the lower coupled layer will be detected. When this is later used as a criterium in the profile analysis, that seals the deal, although several of the upper-left SOM nodes clearly indicate what this reviewer interprets as a clear case of decoupling; a layer of high static stability – presumably a capping inversion – much higher than the indicated BL depth. At the very least this should be discussed. Also, while focusing on thermal stability, very little is discussed about the moisture structure. For example, what about moisture inversions; another feature where the Arctic seems to be special.*

It was in response to a previous reviewer comment on the originally submitted manuscript that so much information had been included in the introduction. Therefore, we have worked to reduce it some, but did not want to remove too much, as we want to honor the original reviewer comment which called for a more detailed introduction. We have addressed your specific comments about the introduction and discussion. First, we now say that "the ABL is often decoupled from the cloud layer" (line 71) instead of "typically decoupled" as originally stated. Second, we have reduced the amount of text spent explaining coupled vs. decoupled clouds in the introduction, and have added some results and discussion which delve deeper into whether cloud may be coupled or decoupled for the various SOM patterns, which now better supports the inclusion of this information in the introduction . This is largely done by adding a SOM figure which looks at mixing ratio profiles, cloud, and liquid water path characteristics for each SOM pattern.

[revised manuscript text omitted]

***This brings me to the duality of the study; the SOM part and the criterium-based analysis. It seems to this reviewer that the former is not at all necessary for latter and that instead this mix causes problems with the narrative of the paper. That doesn't at all mean the SOM analysis is pointless; in fact, I think that the SOM analysis, with a more in-depth discussion of the results, would make a very nice paper all by itself. The text argues that the SOM analysis is a prerequisite for the formulation of the criteria later used, but I see very little of that.***

To address this comment, we have removed most of the criterium-based analysis, and have added an additional figure related to the SOM analysis (Figure 5 in the revised manuscript). We have also added additional discussion to explain why the SOM was a prerequisite for the formulation of the stability regime criteria.

"The SOM made the development of these stability regime criteria possible, as it revealed a manageable number of physically meaningful patterns representative of the entire training dataset, such that important variations between profile $\theta_v$ structures could be discerned. Based on these SOM results, stability regime criteria could be developed that were applicable to any $\theta_v$ profile, and thus were applied to each SOM pattern (using the average of all radiosonde profiles mapped to a given SOM pattern) as well as to individual radiosonde profiles." (line 313)

The SOM was crucial for revealing the range of stability types ultimately defined in Table 2, as if we were to look individually at nearly 1400 radiosonde observations to try to understand the range in stability, it would have been very difficult to make sense of all of it. The SOM, however, makes this possible by distilling the information in all of the observations into 30 patterns which represent the range in features present in the entire dataset.

***This also leads to another peculiarity. First in the methods section (section 2.4), where the SOM analysis is referred to in a "hand-waiving" manner long before any of the SOM results are even presented, let alone discussed. Second, in Section 3.1 where the SOM results are now described already having defined the SOM nodes in terms if the stability classes, that here have not yet been presented and discussed. Either the stability classes a re dependent on the SOM analysis, and then you need to present those, followed by how they inform the***

*stability classes and then discuss those. Or you define the stability classes first, then do the SOM analysis and then identify where they fit. Now you're trying to do both and the result is confusing.*

In order to help with the flow of the paper, we now include Fig. 2 in Section 2.3 when it is first introduced, and we explain that:

"The full range of vertical structures revealed by the SOM was used to develop a set of criteria for classifying stability of any given observation that distills the details of the SOM to the most critical factors of stability within and above the ABL, which will be discussed in Sect. 2.4." (line 296).

There are, however, many details in Fig. 2 which are not discussed until later in the manuscript (e.g., how we determined the stability regime of each SOM pattern). Therefore, we have also added that:

"Additional details included in Fig. 2 will be discussed later in the manuscript." (line 299).

Ultimately, it is up to the typesetter where this figure is inserted in the manuscript, but we hope by moving it to the methods section right after the SOM methods are discussed, as well as adding some additional discussion about how we got from the SOM to the stability regime classifications (see our response to your previous comment, and our response to your following comment) help to make this all more clear.

*In the end, very little quantitative information from the SOM makes it into the criterium selection. If the authors actually did use quantitative results from the SOM directly informing the criterium selection in the vertical-structure statistics analysis, that needs to be much discussed in detail and how this is handled in the narrative needs to be clearer. That this doesn't seem to be the case is borne out by the results. There are very few WS cases; for some seasons there are only a hand-full or less and even annually there's one class with only 9 cases. Is there a point in having a specific criteria that hardly ever happens? To me this seems to be proof that the criteria was not determined objectively.*

We have added some more details about how the SOM was used for the criteria selection:

"The stability regime definitions were developed alongside a similar SOM-based analysis of ABL profiles in Antarctica (Dice et al., 2023). An iterative process was conducted by visually inspecting the MOSAiC and Antarctic SOMs to identify groupings within each SOM which appeared to be substantially different from other groupings in that SOM, based on the near-surface and aloft stability. Then, thresholds (based on prior literature where possible) were determined to differentiate each grouping that made sense for the MOSAiC SOM and all the Antarctic SOMs. This process was completed considering both Arctic and Antarctic SOMs to support the robustness of these methods for classifying stability in either polar region, and to reduce subjectiveness. Further details about the determination of thresholds are provided below." (line 317)

Though the process cannot be completely objective, we attempted to make it as objective as possible by considering both Arctic and Antarctic SOMs, as well as by applying the same thresholds for categorizing near-surface stability as for categorizing stability aloft. This is why we end up with a class such as WS which has very few cases. In our opinion, it would be more subjective to decide that this WS category should be grouped together with another category, than to let it stand alone based on the thresholds for near-surface and aloft stability. Therefore, we stand by our decision for the stability regime criteria even though there are some classes that, especially when separated out seasonally, are rare. To find that these regimes are rare is in itself an important result. We have added some additional text in Sect. 2.4 to explain how we end up with the WS regime, for clarity:

"For the regimes listed as WS and NN, this means that the stability aloft does not fall into a category with greater stability than near the surface." (line 355)

*At the end, the inclusion of the LLJ & clouds into the analysis is also not uninteresting, but somewhat superficial and doesn't add much to the results in general. Almost a page of the introduction (page 3) discusses LLJs and how they can form and LLJs are also taking up almost half a page in the methods section. Also, the cloud information if very superficial; how much clouds (a few octas, scattered or overcast) and how are multi-layer*

*clouds treated? Together the LLJ and cloud section adds 3 pages and two complex figures to an already long paper without adding too much new information on the physics, aside from the frequencies of occurrence.*

We have removed the LLJ and cloud sections included at the end of the paper in the previously submitted version of the manuscript. We have also removed details about LLJs and clouds in the introduction and methods which are no longer relevant based on the removal of these final sections. Instead, we use the SOM to show the important results about how LLJs and clouds correspond to vertical structure and stability, and therefore some of the discussion on these topics is kept in the Discussion and Conclusions section.

*In summary, the content of this paper holds great potential. The novelty of the SOM analysis is however underutilized and the SOM results could have been discussed in much more detail. The criterium-based vertical structure analysis, which I think is the core of this paper, does not really build on the SOM analysis, although some of the results refer back to it. Criterium-based studies are not really very novel but is here more detailed and extensive. But having both in the same paper in this way is at best confusing and is causing the paper to be very long. The addition of the LLJ and cloud analysis at the end makes it even longer without adding very much; both these aspects deserve better. So even if I'm typically not a fan of 2-part-papers, this is a case where I would advocate that method: Part I with the SOM analysis and a Part II with the vertical structure analysis, extended with the moisture profiles and a deeper analysis of the LLJs and cloud information.*

We would like to first note that this work has already been separated into 2 papers: the one which you have reviewed, and another paper recently published in ACP which focuses on the thermodynamic and kinematic turbulent forcings which influence stability regime (i.e., delving deeper in the stability regime-based analysis introduced in the current paper). This paper (Jozef et al., 2023b) is referenced in multiple places (lines 115 and 660 of the previously submitted manuscript). Therefore, we do not wish to further separate the current paper into two papers. However, a lot of the major takeaways from the final LLJ and clouds sections in the current paper which you suggest that we remove are reiterated and explored deeper in Jozef et al., 2023b. Therefore, we are ok with removing these sections from the current manuscript, and have thus done so.

To address your other concern that the SOM is underutilized and that moisture profiles should be considered as well, we have added an additional figure and additional discussion to the SOM analysis which looks at moisture profiles and cloud characteristics in the context of the SOM. This is all discussed in more detail in response to one of your previous comments. We do choose to keep the figure and associated discussion which shows stability regime frequency distribution and ABL characteristics when stability regime identification is applied to individual radiosonde observations (Fig. 6 of revised manuscript), as we think these results are important to highlight some key takeaways from this study in a more straightforward, concise, and accurate way than can be understood from the SOM.

*And possibly as a side issue, and not being a native English speaker, the wind speed can be "large" or "small". The wind may be "strong" or "weak". Bu neither are "fast" or "slow".*

This has been fixed throughout.

*Detailed issues:*
*Lines 45-47: If this is an explanation of the so-called lase-rate feedback, it needs another attempt. The surface heat fluxes have very little to do with Arctic warming; that is determined at the top of the atmosphere. The key issue here is the almost-total absence of deep convection.*

The original phrasing was taken from Lesins et al. (2012) which discusses that Arctic lower atmospheric warming is disproportionately felt right near the surface. We have clarified this, and now say:

"… by dynamically decoupling the surface from the free atmosphere, so that lower atmospheric warming related to increased surface to air (or decreased air to surface) heat fluxes cannot easily spread through the troposphere, and warming is concentrated near the surface (Lesins et al., 2012)." (line 46)

*Lines 56-60: This was a really looong sentence; I'm sure it can be broken up in two or three.*

This sentence has been broken up, with some additional information added, and now reads:

"Near-neutral or weakly stable conditions can occur in the presence of stratiform clouds (Intrieri et al., 2002a; Tjernstrom, 2007; Curry and Ebert, 1992; Liu and Key, 2016; Shupe et al., 2011; Tjernström, 2005, Tjernström et al., 2012; Wang and Key, 2004; Zygmuntowska et al., 2012). Increased near-surface temperatures associated with enhanced downwelling longwave radiation caused by cloud cover can erode the surface inversion (Tjernström et al., 2019), which is sometimes supplemented by downward mixing from the cloud itself (forced by cloud-top radiative cooling) (Morrison et al., 2012). This is common in Arctic summer (Walden et al., 2017) when ample moisture is advected north either into the Arctic or from the broader ice-free areas across the pan-Arctic region. (Sotiropoulou et al., 2016; Tjernström et al., 2019)." (line 60)

*Line 61: To me, "typical" means "very often", almost always. Studies indicate maybe 30% of the time, so I would say "often" instead.*

This now reads:

"The ABL is often decoupled from the cloud layer by a shallow stable layer…" (line 71)

*Lines 66-67: And how would this work? To me its not evident so a little more here would be nice*

In order to shorten the introduction, this sentence has been removed.

*Lines 68-71: Maye also a bit long; shorten or divide if you can.*

In order to shorten the introduction, this sentence has been removed.

*Lines 92-93: The very few direct studies of turbulence on top of a low-level jet that this reviewer has seen doesn't seem to indicate very much mixing.*

Information given about LLJs in the introduction has been reduced per your other comments, so this sentence has been removed.

*Line 97: Does "current paper" mean this manuscript? That begs the question, similar how?*

To clarify, the sentence now reads:

"A study conducted using some of the same measurements as this manuscript found LLJs to be present more than 40% of the time in the central Arctic (Lopez-Garcia et al., 2022)." (line 84)

*Line 94: Is the fact that models have a lower frequency surprise? Is there an explanation for this; comment please!*

The explanation for this is provided in the Discussion and Conclusions when comparing our results to those from previous literature:

"Lastly, the difference in frequency from Tuononen et al. (2015) is likely because the much lower vertical resolution of the Arctic System Reanalysis (ASR-Interim) data used in Tuononen et al. (2015) would miss shallow LLJ cases." (line 626)

*Lines 133-134: Strange mix of "firstly" and "additionally". If you use "first" there has to be a "second". If you use "additional", "first" is not necessary*

Based on the comments from another reviewer, we have removed this sentence from the manuscript.

*Lines 144-145: Change "with the result being being" to "resulting in".*

This has been changed:

"During the MOSAiC year, many measurements were taken to observe the atmosphere (Shupe et al. 2022), sea ice (Nicolaus et al. 2022), and ocean (Rabe et al. 2022), resulting in the most comprehensive set of observations of the central Arctic climate system to date." (line 131)

***Line 151: Discussion about Level 2 and 3 sounding data is meaningless without a description.***

We have added a further descriptor for each product:

"We use the level 2 radiosonde product (Maturilli et al., 2021) for this analysis, as the level 2 Vaisala-processed product is found to be more reliable in the lower troposphere than the level 3 GRUAN-processed product (Maturilli et al., 2022)." (line 138)

***Lines 157-161: This discussion is meaningless without more information. What data sheet is that?***

This discussion was included in response to previous concerns about the validity of the radiosonde winds. We have now added some more information:

"Instrument specifications and uncertainties for the radiosonde variables are provided in the manufacturer data sheet for the Vaisala Radiosonde RS41-SGP (2017), and are summarized in Table 1. It is recognized that the true uncertainties in the wind speed and direction are likely to be greater than those provided in the data sheet, however for the following reasons, we find the original winds provided in Maturilli et al. (2021) to be sufficiently reliable for the current study. First, we determined that our results changed minimally when additional vertical averaging was applied to the winds (beyond the filtering already applied by Vaisala during their data processing), and thus noise in the observations does not bias the results. Second, when comparing radiosonde wind speeds to those measured by the DataHawk2 uncrewed aircraft system which observed the atmosphere during MOSAiC between 5 m and 1 km, Hamilton et al. (2022) found a difference of less than 1 m s$^{-1}$ based on the 95% confidence intervals of observations from both platforms. Nonetheless, caution should be taken with interpretation of radiosonde wind speeds in the lowest 100 m." (line 143)

***Figure 1: The makers are not seen with this resolution, so maybe use a line instead. Also, some of the yellow parts of the legs fade away into the white paper.***

We have adjusted the symbol size and color for better readability.

***Lines 168-170: This was a weird explanation of the "friction velocity". It is not a "theoretical wind speed"; it is a velocity scale derived directly from the momentum flux. It does not express the "magnitude of turbulence"; it is perfectly possible to have high turbulence and friction velocity, for example of turbulence is dominated by buoyancy.***

We have revised the description of friction velocity to now read:

"… a measure of the vertical fluxes of zonal and meridional horizontal momentum, suggesting the magnitude of mechanically generated turbulence… " (line 160)

***Lines 172-175: Using the eddy-covariance momentum flux gives exactly the momentum flux and nothing else; this has nothing to do with latent heat flux! There may be other reasons for using the bulk flux formulas, but this is not one of them!***

We have decided to switch from using the bulk parameterization of u$_*$ to the standard eddy-covariance value, calculated using sonic anemometer measurements at 10 m. Therefore, we have removed discussion of bulk u$_*$ from the paper.

"Atmospheric observations of wind speed at 2 m above the surface, as well as friction velocity (a measure of the vertical fluxes of zonal and meridional horizontal momentum, suggesting the magnitude of mechanically generated

turbulence; $u_*$) measured at 10 m, come from a 10 m meteorological tower (hereafter called the met tower) located on the sea ice near the *Polarstern* (Cox et al., 2023a,b). These measurements provide information about near-surface turbulence at the time of each radiosonde launch. Derivation of $u_*$ through standard eddy-covariance methodology, and corresponding uncertainties, follow Persson et al. (2002)." (line 159)

**Lines 211-212: Would be useful to know how much data was lost when retaining 1377 soundings. Does this number include any inetsive period with more than 6-hourly? 1377 divided by 4 gives a little over 344; very close to the 352 of a full year.**

We have added some description about when the most data were lost:

"The 132 profiles which were removed from analysis are dispersed throughout the year, but many of them were observations from early October, mid-April (an intensive period when radiosondes were being launched every 3 hours), or mid-September." (line 202)

**Lines 213-215: First, a bit more detail would be appreciated; the term "bulk" here can refer either to a finite difference across two layers (in contrast o a real derivative) or between a layer and the surface. Second, if the latter (which I tend to believe) this means, as far as I can see, that the BL top detected will be that of the lower surface-based layer and will not capture a decoupled but turbulent layer aloft, unless the decoupled layer is less than 20 meters on top. You need to be clear about this. The text in lines 218-220 is not enough, since this is such a central issue.**

The bulk Richardson number profile for ABL height detection is calculated over a rolling altitude range of 30 m, calculated every 5 m. This way, a decoupled, but turbulent layer aloft should still be captured and considered as within the ABL. It is only when the layer truly is consistently having a $Ri_b$ value above 0.5 that the ABL height is found. This is explained in detail in Jozef et al., (2022), but additional information has been added to the text:

"$Ri_b$ profiles were created by calculating $Ri_b$ across 30 m intervals in steps of 5 m, rather than using the ground as the reference level, in order to isolate local likelihood of turbulence rather than that over the full depth from the surface (Jozef et al., 2022)." (line 207)

**Line 222: What do you mean by "below"? The wind at the surface below is zero, right?**

Most previous literature phrases the description of an LLJ as a peak in the wind speed that is 2 m/s faster than the minima above and below, which is why we too phrased it like this. However, you are right that the wind speed minimum below is always zero at the surface, so how we phrased it may be confusing. We have changed it to say:

"LLJs were identified from each radiosonde, where there was a maximum in the wind speed that was at least 2 m s$^{-1}$ greater than the wind speed minimum above (Stull, 1988)." (line 215)

**Lines 231-234: This doesn't make sense to me. If you include cases where the LLJ is less than 25% larger than then the next minimum, how does that make you include high-wind environments? Or should it be "low" on line 234.**

We did mean to say "high wind speed environments" on this line. For example, if the wind speeds are high throughout the entire profile, we may have a wind speed maximum of 20 m/s, and the wind speed minimum above is 17 m/s. This meets the criteria of the wind speed maximum being 2 m/s faster than the minimum above, though 20 m/s is not at least 25% greater than 17 m/s. However, if wind speeds are lower throughout the entire profile, we may have a wind speed maximum of 10 m/s, and the wind speed minimum above is 7 m/s. This meets the criteria of the wind speed maximum being 2 m/s faster than the minimum above, AND being least 25% greater than 7 m/s. We have added this example to the text:

"Our analysis differs from that by Lopez-Garcia et al. (2022) as they only considered LLJs in which the jet core speed was at least 25% greater than the wind speed minimum above the jet core, whereas we do not include this criterion, and thus our analysis also includes LLJs which occur in ubiquitously high wind speed environments (e.g., a wind

speed maximum of 20 m s$^{-1}$ would be 2 m s$^{-1}$ faster, but not 25% faster, than a wind speed minimum of 17 m s$^{-1}$ above)." (line 225)

*Line 254: Is this correct? The figure seems to indicate the SOM is applied to the gradient of θv (see Lines 284-285).*

You are correct, thank you for pointing out this typo. This has been fixed:

"Here, the SOM analysis is applied to radiosonde profiles of $\theta_v$ gradient to identify vertical structure and stability in the lowest 1 km of the atmosphere over the Arctic ice pack during MOSAiC." (line 248)

*Line 277: "in the 1377"*

This has been fixed:

"In this study, a 30 pattern SOM was used to describe the range of lower atmospheric stability profiles, defined by $\theta_v$ gradient ($d\theta_v/dz$), present in the 1377 MOSAiC radiosonde profiles." (line 271)

*Line 285: Subtracting the value at 1 km does not make the result an "anomaly".*

We have changed this to say:

"… in the form of the $\theta_v$ difference with respect to that at 1 km, to remove seasonal temperature dependence…" (line 280)

*Line 287: What do you mean by "distinguished"? Maybe the wrong word?*

We have changed this to say:

"… but found that the range in height and strength of the $\theta_v$ inversion, as well as the differentiation between a weakly stable or near-neutral layer below a $\theta_v$ inversion, were not as evident." (line 281)

*Line 289-230: Wouldn't it be easier to calculate the specific humidity, then θv and if necessary linearly interpolate that directly, rather than calculating the pressure separately with the hypsometric equation?*

In order for the value of $\theta_v$ to be most accurate, the individual variables should first be interpolated properly before calculating $\theta_v$. Though perhaps easier to calculate $\theta_v$ first and then interpolate that linearly, this is not the most accurate method.

*Line 300: Again, subtracting the value at 1 km from a profile doesn't make the result an "anomaly". Just a difference.*

We chose to use the word 'anomaly' to represent that the $\theta_v$ profiles are not the raw $\theta_v$, but rather the value of $\theta_v$ with respect to that at a standard altitude of 1 km. We understand that 'anomaly' is commonly used to represent a difference from a climatological mean, but there are other uses of this word in atmospheric sciences. For example, in Cassano et al. (2015), they use 'anomaly' to refer to the value of SLP at a given grid cell minus that averaged across the domain on that day. Previous studies similar to the current study (Cassano et al., 2016; Dice and Cassano, 2022) use 'anomaly' to represent the value at a given altitude minus the value at a standard altitude, similar to what we do. To avoid confusion, we have included additional description of our definition of 'anomaly' in this work:

"… with the mean profiles of $d\theta_v/dz$ and $\theta_v$ anomaly (where 'anomaly' refers to the value at a given altitude minus the value at 1 km) for all radiosondes mapped to a given pattern."(line 295)

*Section 2.3 starts out by discussing relationships with a SOM analysis from which no results have yet been presented and discussed.*

We now include Fig. 2 in Section 2.3, so that the SOM results have been shown by the time we begin the discussion of stability regimes, and how they relate to the SOM patterns, in Section 2.4.

***Lines 326-335: Maybe just language, but in the definition of the criteria, how come "mixed" is used for something more stable than "weakly stable"? In my book "mixed" is a synonym to "near neutral".***

We did not intend to communicate that something more stable than "weakly stable" is "mixed." We have revised the text for improved clarity:

"The second step for stability regime identification is only applied to cases with a near-surface regime of WS or NN and is carried out to differentiate such mixed ABLs (where NN is well-mixed, and WS is almost well-mixed) that are very shallow, from those that are deeper." (line 342)

***Line 338: "… we there …" – drop "we".***

This was a typo, so thank you for catching this. We have revied the sentence to read:

"We make this distinction because there are different processes that would lead to a shallow versus deep mixed layer, which would be better highlighted by differentiating such categories." (line 344)

***Lines 364-365: I don't understand "… was never observed in an individual MOASiC profile …".. There seems to be several NN-like profiles in the upper left of the SOM.***

We were intending to say that NN with no enhanced stability aloft (i.e., talking about the last row in Table 2 labeled as NN, not including NN-SSA, NN-MSA, or NN-WSA) was not observed in any individual profile, nor in any SOM patterns. We have revised the text to make this more clear:

"While NN with no enhanced stability aloft (last row of Table 2) was never observed in an individual MOSAiC profile (in the case of near-surface stability of NN, stability aloft was always weakly to strongly stable), we include its definition in Table 2 to support the use of these criteria for observations from other campaigns." (line 370)

***Line 394-396: Isn't the number of SOM-patterns representing a certain stability a bit beside the point, as they are not equally populated?***

We have added a sentence to explain why this is an interesting/valuable result:

"We note this, as a greater number of patterns of a given stability regime highlights greater variation in vertical structure within that stability regime category." (line 393)

***Line 458: "Drop "in descending order" – pretty obvious if you read.***

This has been removed:

"The most frequent near-surface regime observed was NN (37% of profiles), followed by VSM (27% of profiles), MS (14% of profiles), and SS (13% of profiles)." (line 507)

***Line 475: What is df?***

We explain in that same sentence that df is degrees of freedom. We have attempted to clarify this:

"The determination uses a two-tailed t-test when degrees of freedom (abbreviated as 'df') $\leq$ 100 and a two-tailed z-test when df > 100." (line 524)

***Figure 5 and corresponding discussion: Well, now when you look at the results, compared to the other classes, there are very few WS cases, correct? For some seasons there are only a hand-full or less and even annually***

***there's one class with 9 cases. Does that tell you anything about the validity of the stability criteria? Is there a point in having a criterion that is so specific that it hardly ever happens?***

As explained in a response to one of your previous comments, we included the WS regime for the sake of consistency in separating out weak to strong stability both near the surface, and aloft, following the same thresholds. To then discover that one of the regimes (WS) is very uncommon is an interesting result because it tells us that first of all, weak stability near the surface is the most rare of all near-surface regimes, and when we do have weak stability near the surface, the stability is usually enhanced aloft. We argue that to learn that a certain regime is very uncommon is just as important as learning which regimes are most common, as it adds to our overall understanding of the Arctic ABL. We have added some text to highlight the importance of this finding.

"Weak stability either near the surface or aloft is the rarest condition (demonstrated by few WS or -WSA SOM pattern and low frequencies of the WS and -WSA regimes). Thus, a near-surface regime of WS may represent a transition state between the stronger (SS, MS, and VSM) and weaker (NN) stability regimes, and there are rarely conditions to support weak stability aloft (-WSA). Discovering both the most common and the least common stability regimes are equally as important for our overall understanding of the Arctic lower atmosphere." (line 575)

***Line 577: in a coupled system, the cloud is mixed all the way to the cloud top; not the cloud base.***

We now clarify this:

"… a coupled cloud supports a deeper ABL that is well-mixed up to cloud base (with the mixed layer extending to cloud top)…" (line 641)

***Line 582-584: This statement is so weak that it is entirely useless. Essentially all boundary layers are capped by an inversion, except for the stable boundary layer which is inside an inversion. Hence, there is always a stable layer in the lower troposphere regardless of what the boundary layer looks like. The distinction here is not the stable layer; it is the height. In the tropics you may have to go to several km to find the capping inversion, over the extratropical land maybe 1-3 km on a sunny summer day. The stability of the boundary layer is determined by the stability in the boundary layer. And that is not dominated by stable conditions. How hard is that to say?***

Our main point here is that the stable layer, either within the ABL or capping the ABL, is usually strongly or moderately stable, but not often weakly stable. And this stable layer (capping inversion in the case of a mixed ABL) is located within the lowest 1 km of the atmosphere in the Arctic, which is in contrast to much of the planet where the capping inversion is located at much higher altitudes (as you point out). We have revised the sentence to make our points more clear:

"The most frequent stability regimes were those with strong or moderate stability either near the surface (SS and MS) or aloft (VSM-SSA, VSM-MSA, NN-SSA, and NN-MSA). Thus, we conclude that the central Arctic atmosphere over sea ice is inclined to include a strongly or moderately stable layer somewhere below 1 km AGL, and usually below 400 m (this contrasts with the mid-latitudes and tropical regions where the capping inversion is often as high as 1 to 2 km). Sometimes this strongly to moderately stable layer is within the ABL and sometimes it caps a well-mixed ABL, with the latter scenario occurring with higher frequency than the former, consistent with Tjernström and Graversen (2009). In the latter scenario, the depth of the well-mixed layer is highly variable, ranging from 38 m (minimum ABL height of a VSM case) to 914 m (maximum ABL height of an NN case). Weak stability either near the surface or aloft is the rarest condition (demonstrated by few WS or -WSA SOM pattern and low frequencies of the WS and -WSA regimes). Thus, a near-surface regime of WS may represent a transition state between the stronger (SS, MS, and VSM) and weaker (NN) stability regimes, and there are rarely conditions to support weak stability aloft (-WSA). Discovering both the most common and the least common stability regimes are equally as important for our overall understanding of the Arctic lower atmosphere." (line 568)

***Lines 616-628: This paragraph is very speculative, to the point that I think it should be dropped. First, the discussion about how a LLJ is formed in relation to the LLJ core height and BL depth is very hand-waiving to say the least. Second, it is also dependent on the definition of the BL-height which here is not the same as the capping inversion depth. In all cases where there is a decoupling, one may find the LLJ at the top of the turbulent layer associated with the decoupled cloud and that would not necessarily be a sign of baroclinicity.***

We have removed any discussion here and throughout the paper on the LLJ formation mechanisms, as you make a good point that this is all speculative, and we lack the analysis needed to really answer this question in the current study.

***Line 638: If you mention the ASR, you will have to tell the reader what that is and why its resolution is lower.***

We now include the full name of ASR, which reveals this is reanalysis, making it obvious as to why the resolution would be lower (reanalysis data usually have lower resolution than current observations):

"Lastly, the difference in frequency from Tuononen et al. (2015) is likely because the much lower vertical resolution of the Arctic System Reanalysis (ASR-Interim) data used in Tuononen et al. (2015) would miss shallow LLJ cases." (line 626)

---

## Author Response (AR3)

**Response to Anonymous Referee Comments**

We would like to sincerely thank the anonymous referee for once again taking the time to review our manuscript and for their helpful comments. Each referee comment is given below in ***bold italics*** followed by our response to the comment. The line numbers provided in our responses refer to line numbers in the revised manuscript.

***Review of the second revised version on***

***An overview of the vertical structure of the atmospheric boundary layer in the central Arctic during MOSAiC***

***By***
***Gina C. Jozef, John J. Cassano, Sandro Dahlke, Mckenzie Dice, Christopher J. Cox and Gijs de Boer***

***In my view, most of my criticisms of the 2nd version have been satisfactorily addressed and the manuscript is now much improved and much easier to read. Above all, the quality of the illustrations is now quite convincing, especially in comparison with the first version. I still have a few minor points to consider and then the paper can be accepted.***

***In Table 1: How can the inaccuracy of the friction velocity in the METEK-Sonic be less than the velocity measurement itself? This doesn't make sense to me at first glance, but it may still be correct, especially since the inaccuracies seem to be defined differently.***

$u*$ (friction velocity) is an expression of the Reynold's stress normalized by the fluid density. While the units of $u*$ are m/s, the metric is not comparable to the horizontal wind velocities. $u*$ is a fictitious velocity whose absolute value is even generally smaller than the uncertainty in wind speed. It is derived from the covariances of the components of the 3-dimensional wind field. While these vector components have the larger uncertainty you refer to, they are themselves derived from a common acoustic measurement such that the error is shared, reducing the uncertainty in a relative calculation of their co-variability, e.g., $u*$.

***In Figure 6, the figures at the upper edge could be enlarged a little; I think there should be enough space and if it is not too much effort, I would do it.***

Did you mean to say that "the numbers at the upper edge could be enlarged a little" in Figure 6? Assuming this is what you had intended to say (as I cannot determine what "figures at the upper edge" would mean), we have made this change (line 544).

***About the LLJ and the interpretation of Egerer et al, 2023:***

***Probably this was a misunderstanding as Egerer et al. provide one of the most comprehensive sets of observational data including local turbulence, so I would not suggest deleting this citation, but I agree with your revised sentences and think it would be fair to include Egerer et al. like the other references.***

We have added the Egerer et al., 2023 citation back into the manuscript at this point (line 81).